# Whole genome phylogenies reflect the distributions of recombination rates for many bacterial species

**Thomas Sakoparnig, Chris Field†, Erik van Nimwegen***

Biozentrum, University of Basel, and Swiss Institute of Bioinformatics, Basel, Switzerland

**Abstract** Although recombination is accepted to be common in bacteria, for many species robust phylogenies with well-resolved branches can be reconstructed from whole genome alignments of strains, and these are generally interpreted to reflect clonal relationships. Using new methods based on the statistics of single-nucleotide polymorphism (SNP) splits, we show that this interpretation is incorrect. For many species, each locus has recombined many times along its line of descent, and instead of many loci supporting a common phylogeny, the phylogeny changes many thousands of times along the genome alignment. Analysis of the patterns of allele sharing among strains shows that bacterial populations cannot be approximated as either clonal or freely recombining but are structured such that recombination rates between lineages vary over several orders of magnitude, with a unique pattern of rates for each lineage. Thus, rather than reflecting clonal ancestry, whole genome phylogenies reflect distributions of recombination rates.

***For correspondence:**
erik.vannimwegen@unibas.ch

**Present address:** †Institute of Microbiology, ETH-Zurich, Zurich, Switzerland

**Competing interests:** The authors declare that no competing interests exist.

## Introduction

The only illustration that appears in Darwin's Origin of Species (*Darwin, 1859*) is of a phylogenetic tree. Indeed, the tree has become the archetypical concept representing biological evolution. Since every biological cell that has ever lived was the result of a cell division, all cells are connected through cell divisions in a giant tree that stretches all the way back to the earliest cells that existed on earth. Thus, the study of biological evolution in some sense corresponds to the study of the structure of this giant cell-division tree.

It is therefore natural that the first step in the analysis of a set of related biological sequences is to reconstruct the phylogenetic tree that reflects the cell division history of the sequences, that is their 'clonal phylogeny'. Once the ancestral relationships between the sequences are known, the evolution of the sequences can then be modeled along the branches of this tree. Indeed, virtually all models of evolutionary dynamics are formulated as occurring along the branches of a tree, and many mathematical and computational methods have been developed for their inference, see for example (*Felsenstein, 1981*; *Page and Holmes, 1998*). This strategy has been employed from the earliest days of sequence analysis (*Zuckerkandl and Pauling, 1965*) and is almost invariably applied in the analysis of microbial genome sequences, which is the main topic of this work.

A second key concept in models of evolutionary dynamics is the idea of a 'population' of organisms that are mutually competing for resources and that, for purposes of mathematical modeling, can be considered exchangeable in the sense that they are subjected to the same environment. Indeed, populations of exchangeable individuals form the basis of almost all mathematical population genetics models (see e.g. [*Hartl and Clark, 2006*]), including coalescent models for phylogenies (*Wakeley, 2008*). Although it is of course well recognized that, in the real world, populations are structured into sub-populations with varying degrees of interaction between them, population

genetics models of evolutionary dynamics almost by definition assume that at some level there are sub-populations of exchangeable individuals sharing a common environment.

In this paper, we present evidence that we believe challenges the usefulness of applying these two concepts for describing genome evolution in prokaryotes. First, we find that for most bacterial species recombination is so frequent that, within an alignment of strains, each genomic locus has been overwritten by recombination many times and the phylogeny typically changes tens of thousands of times along the genome. Moreover, for most pairs of strains, none of the loci in their pairwise alignment derives from their ancestor in the clonal phylogeny, and the vast majority of genomic differences result from recombination events, even for very close pairs. Consequently, apart from a few groups of very closely related strains, clonal ancestry cannot be reconstructed from the genome sequences using currently available methods and, more generally, the strategy of modeling microbial genome evolution as occurring along the branches of a single phylogeny breaks down.

Second, we show that recombination among bacterial strains is not random but subject to strong and complex *population structure*, that is recombination occurs at very different rates between different lineages. Although almost every short segment in the whole genome alignment of a set of strains follows a different phylogeny, these phylogenies are not uniformly randomly sampled from all possible phylogenies, but sampled from highly biased distributions that reflect this population structure. In particular, while a large diversity of phylogenetic trees occurs along the whole genome alignment, some subsets of strains share alleles much more frequently than others. In particular, the frequencies with which different subsets of strains share alleles follow approximately power-law distributions. In addition, almost every individual strain has a distinct distribution of frequencies with which it shares alleles with other strains. This suggests that the rates at which the genetic ancestors of each strain have recombined with the genetic ancestors of other strains are unique for each strain, so that the assumption that at some level the strains can be considered as exchangeable members of a population may also fundamentally break down.

The structure of the paper is as follows. To present our analyses, we will focus on a collection of 91 wild *Escherichia coli* strains that were isolated over a short period from a common habitat (*Ishii et al., 2007*). Using these strains, we introduce the main puzzle of bacterial whole genome phylogeny: although the phylogenies of individual genomic loci are all distinct, the phylogeny inferred from any large collection of genomic loci converges to a common structure, for example (*Wolf et al., 2002*; *Lerat et al., 2003*; *Touchon et al., 2009*). This convergence has led many researchers to assume that the phylogeny reconstructed from a whole genome alignment must represent the clonal phylogeny, and that recombination can be detected and quantified by measuring deviations from this whole genome phylogeny, for example (*Bobay et al., 2015*; *Didelot and Wilson, 2015*; *Yahara et al., 2016*; *Mostowy et al., 2017*; *Garud et al., 2019*). However, as far as we are aware, there are no rigorous justifications for assuming that this whole genome phylogeny corresponds to the clonal phylogeny or that deviations from this global phylogeny can be used to quantify recombination. Here, we introduce methods for quantifying the role of recombination that do not rely on such assumptions and show that the whole genome phylogeny in fact does not correspond to the clonal phylogeny for many bacterial species, but instead reflects population structure.

We first study recombination by studying pairs of strains, extending a recent approach by *Dixit et al., 2015*, to model each pairwise alignment as a mixture of clonally inherited and recombined regions. We show that, as the distance to the pair's clonal ancestor increases, the fraction of the genome covered by recombined segments increases, and at some pairwise distance all clonally inherited DNA disappears. Importantly, this distance is far below the typical divergence of pairs of strains such that for the vast majority of pairs, none of the DNA in their genome alignment stems from their clonal ancestor.

Much of the new analysis methodology that we introduce is based on bi-allelic single-nucleotide polymorphisms (SNPs; which constitute almost all SNPs in the core alignment). Although bi-allelic SNPs have been studied to estimate the number of recombinations along alignments of sexually reproducing species (*Hudson and Kaplan, 1985*), they have received relatively little attention in the study of prokaryotic genomes. We show that almost every bi-allelic SNP corresponds to a single-nucleotide substitution in the history of the strains at the corresponding genomic position, so that the subset of strains sharing a common nucleotide must form a clade in the phylogeny at that position. We show various ways in which these bi-allelic SNPs can be used to investigate which SNPs are consistent with given phylogenies, or each other, and use them to quantify the amount of

phylogenetic variation along the alignment. Using such analysis, we show that the phylogeny must change every few SNPs along the genome alignment, and derive a lower bound on the ratio of recombination to mutation events in the genome alignment.

To obtain a separate validation of our methods, we not only apply our methods to the real data from the *E. coli* strains but also to data from simulations of a simple evolutionary dynamics in which genomes of a well-mixed population of fixed size $N$ evolve under reproduction, (neutral) mutation at a fixed rate $\mu$ per base per generation, and homologous recombination at a rate $\rho$ per base per generation. By comparing the known ground truth for the simulated data with the results of our methods, we confirm the accuracy of our methods on the simulated data. We also use results of our methods on simulated data to clarify the precise meaning of several statistics that we calculate.

Applying the methods and statistics that we developed for *E. coli* to a set of other bacterial species, that is *Bacillus subtilis*, *Helicobacter pylori*, *Mycobacterium tuberculosis*, *Salmonella enterica*, and *Staphylococcus aureus*, we show that, with the exception of *M. tuberculosis* where all strains are very closely related and most if not all DNA has been clonally inherited, all other species follow the same general behavior as *E. coli*.

To explain how a robust core genome phylogeny can emerge in spite of the fact that a very large number of different phylogenies occurs along the genome, we use data from human genomes as an illustration. We show that phylogenies reconstructed from large numbers of loci from human genomes also converge to a robust phylogeny. Moreover, this phylogeny reflects the known human population structure, and we propose that bacterial whole genome phylogenies similarly reflect population structure, that is the rates with which different lineages have recombined. To support this interpretation, we show how bi-allelic SNPs can also be used to quantify the relative frequencies with which different subsets of strains share alleles. We find that these frequencies follow approximately scale-free distributions, indicating that there is population structure at every scale. Finally, we define entropy profiles of the phylogenetic variability of each strain and show that these entropy profiles provide a unique phylogenetic fingerprint of almost every strain. In addition, we show that simple evolutionary models of fully mixed populations cannot reproduce the statistics we observe for real genomic data, further supporting that these phylogenetic patterns reflect complex population structure. We conclude that our observations necessitate a new way of thinking about how to model genome evolution in prokaryotes.

## Results

To illustrate our methods, we focus on the SC1 collection of wild *E. coli* isolates that were collected in 2003–2004 near the shore of the St. Louis river in Duluth, Minnesota (*Ishii et al., 2006*; *Ishii et al., 2007*). We sequenced 91 strains from this collection together with the K12 MG1655 lab strain as a reference. In a companion paper (Field, 2020, in preparation), we discuss this collection in more detail and extensively analyze the evolution of gene content and phenotypes of this collection. Here, we focus on sequence evolution in the core genome of these strains, that is the genomic regions that are shared by all strains. Although the SC1 strains were collected from a common habitat over a short period of time, they show a remarkable diversity, with no two identical strains, all known major phylogroups of *E. coli* represented, as well as an 'out group' of 9 strains that are more than 6% diverged at the nucleotide level from other *E. coli* strains (see *Figure 1—figure supplement 1* for a phylogenetic tree constructed using maximum likelihood on the joint core genome of the SC1 strains and 189 reference strains [Field, 2020, in preparation]).

### Phylogenies of individual loci disagree with the phylogeny of the core genome

To construct a core genome alignment of the SC1 strains and K12 MG1655, that is the genomic regions that occur in all strains, we used the Realphy software (*Bertels et al., 2014*; see Materials and methods), resulting in a multiple alignment across all 92 strains of 2′756′541 base pairs long, with 299′077 (10.8%) of the positions exhibiting polymorphism. Note that the core genome length corresponds to about 56% of the median genome length of the strains. Although all strains are unique when their full genomes are considered, there are some groups of strains that are so close that they are identical in their core genomes, so that there are only 82 unique core genomes in total. Realphy

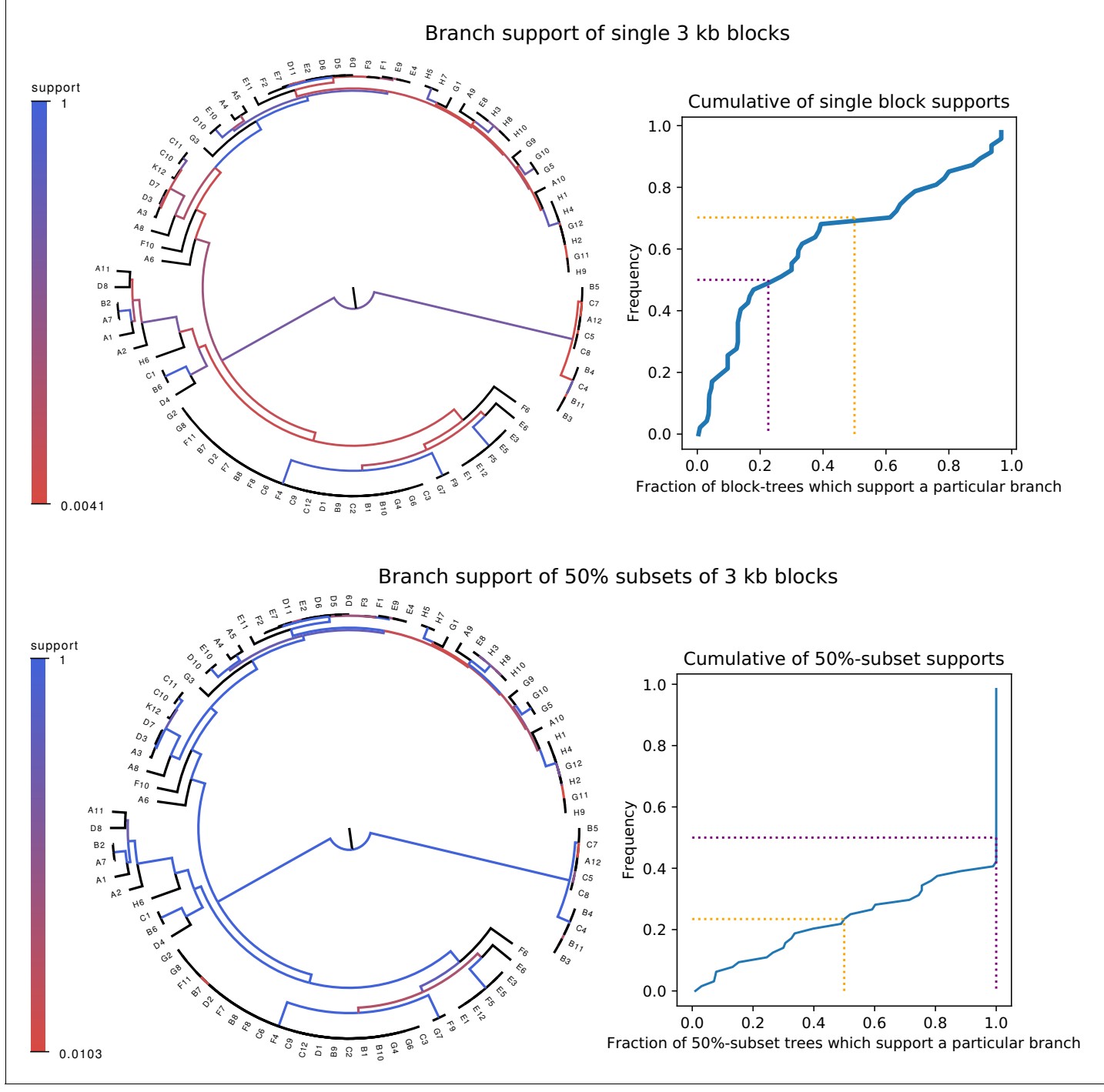

**Figure 1.** Whereas phylogenies of individual alignment blocks differ substantially from the core tree, phylogenies reconstructed from a large number of blocks are highly similar to the core tree. Top left: For each split (i.e. branch) in the core tree, the color indicates what fraction of the phylogenies of 3 kb blocks support that bi-partition of the strains. Top right: Cumulative distribution of branch support, that is fraction of 3 kb blocks supporting each branch. The dotted lines indicated show the fraction of branches that have less than 50% support (yellow) and the median support per branch (purple). Bottom left and bottom right : As in the top row, but now based on phylogenies reconstructed from random subsets of 50% of all 3 kb blocks as opposed to individual blocks.

The online version of this article includes the following source data and figure supplement(s) for figure 1:

**Source data 1.** List of database accessions of the genomes used.

**Figure supplement 1.** Joint maximum likelihood phylogeny of our strains and 189 *E.coli* reference strains.

*Figure 1 continued on next page*

*Figure 1 continued*

**Figure supplement 2.** Each 3 kb alignment block rejects the core tree topology as well as the topologies of the phylogenies reconstructed from all other blocks.

**Figure supplement 3.** Fractions of positions that are clonally inherited (not affected by recombination) along each branch of the clonal phylogeny for the simulated datasets.

used PhyML (*Guindon et al., 2010*) to reconstruct a phylogeny from the core genome alignment and we will refer to this tree as the *core tree* from here on (*Figure 1*).

We first checked to what extent the alignments of individual genomic loci are statistically consistent with the core tree. For each 3 kb block of the core alignment, we used PhyML to reconstruct a phylogeny and then compared its log-likelihood with the log-likelihood that can be obtained when the phylogeny is constrained to have the topology of the core tree. We find that essentially all 3 kb alignment blocks reject the core tree topology (*Figure 1—figure supplement 2*, left panel). Moreover, each alignment block rejects the topologies that were constructed from all other alignment blocks (*Figure 1—figure supplement 2*, right panel).

Although it thus appears that the phylogeny at each genomic locus is statistically significantly distinct, it is still possible that all these phylogenies are highly similar. In order to quantify the differences between the core tree and the phylogenies of 3 kb blocks we calculated, for each split in the core tree, the fraction of 3 kb blocks for which the same split occurred in the phylogeny reconstructed from that alignment block. As shown in the top row of *Figure 1*, the phylogenies of individual blocks differ substantially from the core tree: roughly two-thirds of the splits in the core tree occur in less than half of 3 kb block phylogenies and half of the core tree splits occur in less than a quarter of all 3 kb block phylogenies. Particularly, the splits higher up in the core tree do not occur in the large majority of block phylogenies.

These observations are not particularly novel. There is by now a vast and sometimes contentious literature on the role of recombination in prokaryotic genome evolution which is beyond the scope of this article to review. We thus focus on a few key points that are central to the questions and methods we study here. First, systematic studies of complete microbial genomes have shown that horizontal gene transfer is relatively common and can significantly affect phylogenies of individual loci, for example (*Guttman and Dykhuizen, 1994*; *Lawrence and Ochman, 1998*), and many studies have observed that different genomic loci support different phylogenies, for example (*Shapiro et al., 2012*). Such observations caused some researchers to question whether trees can be meaningfully used to describe genome evolution (*Doolittle, 1999*). Interestingly, it has recently been shown that recombination can play a major role in genome evolution even within relatively short-term laboratory evolution experiments (*Maddamsetti and Lenski, 2018*).

In spite of this, many researchers in the field feel that a major phylogenetic backbone can still be extracted from genomic data. For example, it has been observed that, whenever a phylogeny is reconstructed from the alignments of a large number of genomic loci, one obtains the same or highly similar phylogenies, for example (*Wolf et al., 2002*; *Lerat et al., 2003*; *Touchon et al., 2009*). We also observe this behavior for our strains. Phylogenies reconstructed from a random sample of 50% of all 3 kb blocks look highly similar to the core tree, that is with two thirds of the core tree's splits occurring in *all* phylogenies (*Figure 1*, bottom row).

How should we interpret this convergence of phylogenies to the core phylogeny as increasing numbers of genomic loci are included? One interpretation that has been proposed is that once a large number of genomic segments is considered, effects of horizontal transfer are effectively averaged out and the phylogeny that emerges corresponds to the clonal ancestry of the strains, for example (*Wolf et al., 2002*; *Bobay et al., 2015*). Indeed, it has become quite common for researchers to detect and quantify recombination using methods that compare local phylogenetic patterns with an overall reference phylogeny constructed from the entire genome, for example (*Bobay et al., 2015*; *Didelot and Wilson, 2015*; *Yahara et al., 2016*; *Mostowy et al., 2017*; *Garud et al., 2019*; *Croucher et al., 2015*). However, the validity of such approaches rests on the assumption that this reference phylogeny really represents the clonal phylogeny, and it is currently unclear whether this is justified.

Indeed, some recent studies have argued that recombination is so common in some bacterial species that it is impossible to meaningfully reconstruct the clonal tree from the genome sequences, and that these species should be considered freely recombining, for example (*Rosen et al., 2015*). However, if members of the species are freely recombining, one would expect the core tree to take on a star-like structure as opposed to the clear and consistent phylogenetic structure that phylogenies converge to as more genomic regions are included in the analysis. Addressing this puzzle is one of the topics of this work.

## Simulations of a simple evolutionary model including drift, mutation, and recombination

To confirm the validity of our methods introduced below, and to provide a reference for comparison of the results we observe on the *E. coli* data with those from a well-understood evolutionary dynamics, we performed simulations of a relatively simple evolutionary model that includes drift, (neutral) mutation, and recombination. As described in the Materials and methods, we simulated populations of constant size $N$ with non-overlapping generations, a constant mutation rate $\mu$ per generation per base, and assuming all mutations are neutral. We also included recombination by inserting genomic fragments of length $L_r$ from randomly chosen other individuals in the population at a rate $\rho$ per position per generation. In order to make the simulation results comparable to our core genome alignment of *E. coli* strains, we focused on genome alignments of a sample of $S = 50$ individuals from the population and set the mutation rate $\mu$ such that the fraction of polymorphic columns in the simulated alignment is similar to that observed in the real data, that is 10%. Apart from simulations without recombination, we performed simulations with a wide range of recombination to mutation rates from $\rho/\mu = 0.001$ to $\rho/\mu = 10$. We used $L_r = 12kb$ for the length of the recombination segments which is on the lower end of the recombinant segments we observe in *E. coli*, as we will see below (*Figure 2J*). In addition, in order to facilitate comparison of statistics across simulations with different recombination rates, we made sure to design these simulations such that a clonal tree of the sample of $S = 50$ genomes is drawn once according to the well-known Kingman coalescent (*Wakeley, 2008*), and then used the same clonal tree in all simulations. In each simulation, we explicitly tracked how many times each position in the genome was overwritten by recombination along each branch of the clonal phylogeny (see Materials and methods).

As the ratio of the recombination and mutation rate $\rho/\mu$ is increased, progressively more of the clonal history is lost. *Figure 1—figure supplement 3* shows, for different values of $\rho/\mu$, the clonal tree with branches colored according to the fraction of positions in the genome that were clonally inherited (that is not overwritten by recombination) along the branch. At a very low rate of recombination of $\rho/\mu = 0.001$, the large majority of all positions in each branch are clonally inherited. When the ratio $\rho/\mu$ is raised to 0.01, evolution is still mostly clonal along most of the branches toward the leaves of the tree, but for the few long internal branches near the root that separate the major 'clades', most positions have already been affected by recombination. Once the ratio $\rho/\mu$ is further raised to 0.1, almost all branches are dominated by recombination except for a few branches near the leaves. For $\rho/\mu = 0.3$ or larger, almost every position in every branch has been overwritten by recombination.

## Quantifying recombination through analysis of pairs of strains

As a first analysis of the impact of recombination, we follow an approach recently proposed by Dixit et al. based on the pairwise comparison of strains (*Dixit et al., 2015*). The simplest measure of the distance between a pair of strains is their nucleotide divergence, that is the fraction of mismatching nucleotides between the two strains in the core genome alignment. For pairs of strains with very low divergence, for example D6 and F2 with divergence $4 \times 10^{-4}$ (*Figure 2A*), the effects of recombination are almost directly visible in the pattern of SNP density along the genome. While the SNP density is very low along most of the genome, that is $0 - 2$ SNPs per kilobase, there are a few segments, typically tens of kilobases long, where the SNP density is much higher and similar to the typical SNP density between random pairs of *E. coli* strains, that is $10 - 30$ SNPs per kilobase. These high SNP density regions almost certainly result from horizontal transfer events in which a segment of DNA from another *E. coli* strain, for example carried by a phage, made it into one of the ancestor cells of this pair, and was incorporated into the genome through homologous recombination. For pairs of

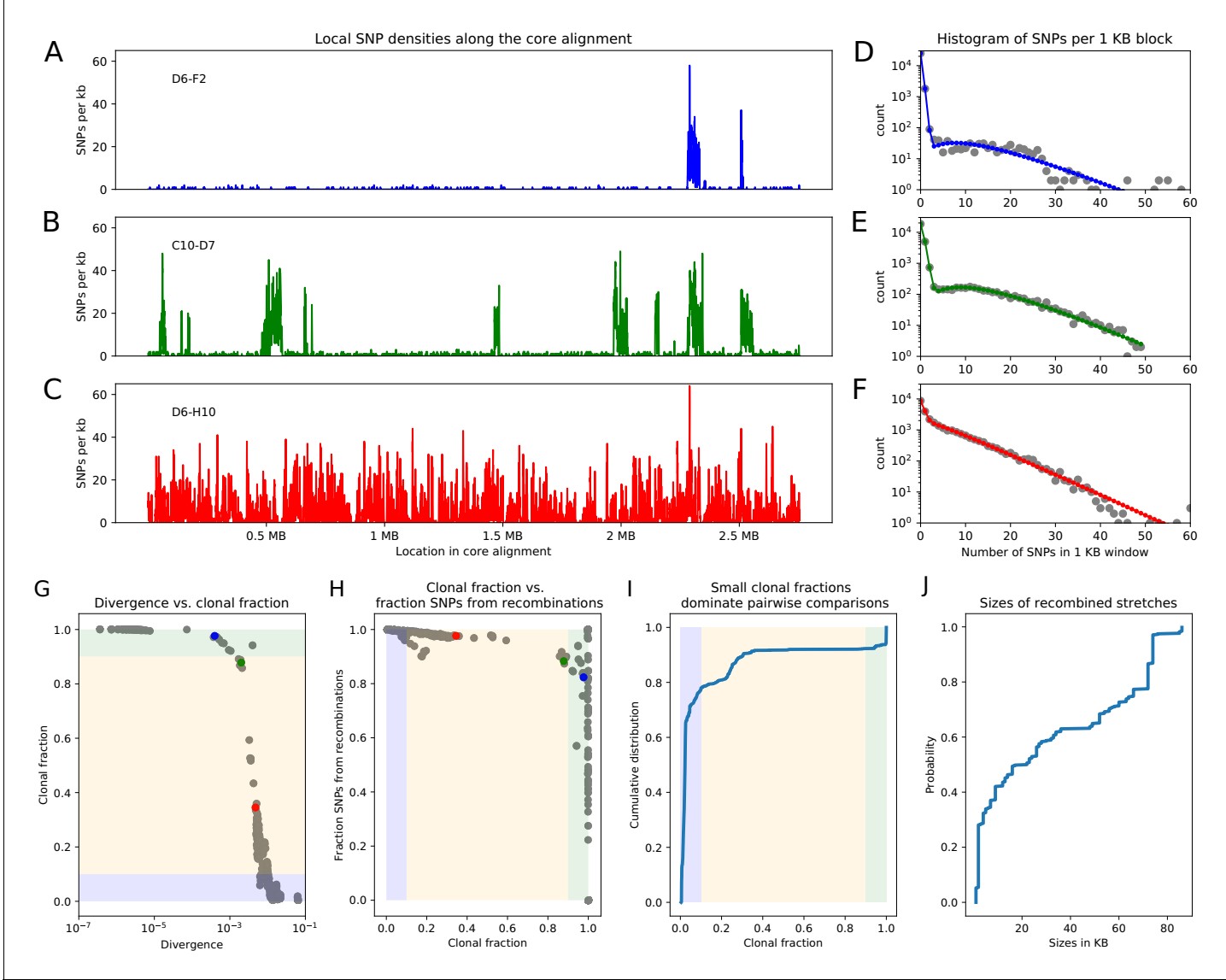

**Figure 2.** Pairwise analysis of recombination in the SC1 strains. (A-C) SNP densities (SNPs per kilobase) along the core genome for three pairs of strains at overall nucleotide divergences of $4 \times 10^{-4}$ (D6–F2), 0.002 (C10–D7), and 0.0048 (D6–H10). (D-F) Corresponding histograms for the number of SNPs per kilobase (dots) together with fits of the mixture model for D6-F2 (blue), C10-D7 (green), and D6-H10 (red). Note the vertical axis is on a logarithmic scale. (G) For each pair of strains (dots), the fraction of the genome that was inherited clonally is shown as a function of the nucleotide divergence of the pair, shown on a logarithmic scale. The three pairs that were shown in panels (A-F) are shown as the blue, green, and red dots. The light green, yellow, and blue segments show strains that are mostly clonal, a mixture of clonal and recombined, and fully recombined, respectively. (H) Fraction of all SNPs that lie in recombined regions as a function of the clonally inherited fraction of the genome. (I) Cumulative distribution of the clonal fractions of the pairs. (J) Cumulative distribution of the lengths of recombined segments for pairs that are in the mostly clonal regime. The mean length of recombined regions is 31′197, with first quartile 2000, median 19′500, and third quartile 66′000.

The online version of this article includes the following figure supplement(s) for figure 2:

**Figure supplement 1.** Statistics of the pairwise analysis on the simulated data.

**Figure supplement 2.** The pairwise analysis accurately estimates both the clonally inherited fractions and the sizes of the recombined segments for the simulation data.

increasing divergence, for example the pair C10-D7 with divergence 0.002 in *Figure 2B*, the frequency of these recombined regions increases, until eventually most of the genome is covered by such regions (pair D6-H10 in *Figure 2C*).

For close pairs, the histograms of SNP densities also clearly separate into two components: a majority of clonally inherited regions with up to at most 3 SNPs per kilobase, and a long tail of recombined regions with up to 50 or 60 SNPs per kilobase (*Figure 2D–E*). As explained in the methods, the distributions of SNP densities are well described by mixtures of a Poisson distribution for the clonally inherited regions plus a negative binomial for the recombined regions (solid line fits in *Figure 2D–F*). In this way we can estimate, for each pair of strains, the fraction $f_c$ of the genome that was clonally inherited, and the fraction $f_r$ of SNPs that fall in clonally inherited versus recombined regions. In addition, to estimate the lengths of the recombined fragments we focused on very close pairs for which recombination events are sparse enough so that overlapping events are very unlikely, and used a Hidden Markov model to estimate the distribution of lengths of recombined regions (see Materials and methods). We find that most recombined segments are in the range of $10 - 70$ kilobases long with a mean of 31 kilobases (*Figure 2*, and see Appendix 2 for additional comments on the accuracy of this estimate).

From this analysis we see that, whenever the pairwise divergence is less than 0.001, the large majority of blocks is clonally inherited, which is indicated as the light-green segment in *Figure 2G*. However, over a narrow range of divergence between 0.001 and 0.01 the fraction of clonally inherited DNA drops dramatically (yellow segment in *Figure 2G*) and at a divergence of about 0.014 essentially the entire alignment has been overwritten by recombination and all clonally inherited DNA is lost (blue segment in *Figure 2G*). Notably, 80% of all strain pairs lie in this fully recombined regime (*Figure 2I*). Thus, for the large majority of pairs of strains, none of the DNA in their alignment derives from their clonal ancestor, making it impossible to estimate the distance to their clonal ancestor from comparing their sequences.

Moreover, as shown in *Figure 2H*, even for pairs that are so close that most of their genomes are clonally inherited, the large majority of the substitutions derives from the recombined regions. That is, for pairs of strains that are so close that the clonally inherited fraction is around $f_c = 0.9$, the fraction of substitutions deriving from recombination is roughly $f_r = 0.9$ as well. We note that in previous studies the quantitative importance of recombination has often been summarized by a ratio $R/M$ of the rates at which substitutions are introduced by recombination and mutation, for example (*Vos and Didelot, 2009*). Note that, if we assume a fixed ratio of recombination-to-mutation $\rho/\mu$, then $R/M$ equals $\rho/\mu$ times the average number of substitutions that are introduced per recombination event. In our pairwise analysis, the ratio $f_r/(1 - f_r)$ corresponds to the ratio of the number of substitutions deriving from recombination and mutation, so that one can obtain an estimate of $R/M$ by calculating $f_r/(1 - f_r)$ for very close pairs with $f_c \approx 1$, that is the dots at the right border in *Figure 2H*. However, we see that in this limit the fraction $f_r$ of SNPs deriving from recombination becomes highly variable, giving a first indication that $R/M$ may vary across different pairs of closely related strains. Indeed, we will see below that recombination rates vary over a wide range across different lineages, so that it is inherently misleading to quantify recombination by a *single* ratio $R/M$.

To test the accuracy of the pair analysis, we applied the same pairwise analysis to alignments of the $S = 50$ sample genomes from each of the simulations. *Figure 2—figure supplement 1* shows the results, analogous to those of *Figure 2G–I*, for the data from simulations with recombination to mutation ratios of $\rho/\mu = 0, 0.001, 0.01, 0.1$, and $0.3$. For the simulations without recombination, the pairwise analysis correctly infers that all of the genome is clonally inherited for all pairs. For the very low recombination rate $\rho/\mu = 0.001$, the model also correctly infers that clonal evolution dominates, that is more than 50% of the genome is clonally inherited for all pairs, and more than 90% of the genome is clonally inherited for about 40% of all pairs. The pairwise analysis of the simulation data with recombination rate $\rho/\mu = 0.01$ are also consistent with the ground truth of *Figure 1—figure supplement 3, i* for half of the pairs there is a substantial fraction of clonally inherited genome, whereas for the other half of more distally related pairs recombination has already affected more than 90% of the genome. Similarly, for $\rho/\mu = 0.1$ the pairwise analysis correctly infers that pairs which are not yet fully recombined have become rare, and for $\rho/\mu = 0.3$ essentially all pairs have become fully recombined. The pairwise analysis of the simulated data thus paints a correct picture of the ground truth shown in *Figure 1—figure supplement 3*.

To get a more quantitative assessment of the accuracy of the pairwise analysis, *Figure 2—figure supplement 2* directly compares the true fraction of clonally inherited genome for each pair with the estimated fraction from the pairwise analysis for the simulations with $\rho/\mu = 0.01$ and $\rho/\mu = 0.1$. We see that for all pairs the estimated fraction of the genome that was clonally inherited is close to the

true fraction. In fact, it appears that the estimation method tends to slightly overestimate the fraction of clonally inherited genome for almost all pairs. Finally, as for the real data, we also used close pairs from the simulations with $\rho/\mu = 0.01$ and $\rho/\mu = 0.1$ to estimate the sizes of the recombined segments. As shown in *Figure 2—figure supplement 2*, the sizes of the recombined regions estimated by the method are very close to the ground truth of 12 kb recombination segments.

In summary, application of the pairwise analysis to the simulation results strongly support that the pairwise analysis accurately quantifies the fraction of the genome that was affected by recombination for each pair, as well as estimate the lengths of the recombined segments.

For later comparison with the data on other species, we summarize our observations from the pairwise analysis by a few key statistics. First, half of the genome is recombined at a *critical divergence* of 0.0032. Second, at this critical divergence, the fraction of all SNPs that is in recombined regions is 0.95. Third, the fraction of mostly clonal pairs is 0.077, and finally, the fraction of fully recombined pairs is 0.78 (see Materials and methods). All these statistics suggest that pairwise divergences between strains are almost entirely driven by recombination and do not reflect distances to their clonal ancestors. To understand how a consistent phylogenetic structure can still emerge when the full core genomes of all strains are compared, we need to go beyond studying pairs.

## SNPs in the core genome alignment correspond to splits in the local phylogeny

Although there may not be a single phylogeny that captures the evolution of our genomes, each *single position* in the core genome alignment, that is each alignment column, will have evolved according to some phylogenetic tree. A key observation is that our set of strains is sufficiently closely related that there are almost no alignment columns for which more than one substitution occurred in its evolutionary history. In particular, of the almost 2.8 million columns in the core genome alignment, almost 90% show an identical nucleotide for all 92 strains, that is only 10.85% are polymorphic. Moreover, almost all of these SNP columns are bi-allelic, that is for 93.6% of the SNPs only two nucleotides appear, 6.3% have three nucleotides, and in 0.2% all four nucleotides occur. These statistics strongly suggest that most positions have not undergone any substitutions, and that columns with multiple substitutions are rare. Notably, these statistics are still inflated due to the occurrence of an outgroup of nine strains that is far removed from the other strains (the clade from B5 to B3 visible on the right in *Figure 1*). We observe that almost 36% of all SNPs correspond to SNPs in which all nine strains of this outgroup have one nucleotide, and all other 83 strains have another nucleotide. If we remove the outgroup from our alignment, the fraction of SNPs in the alignment drops from 10.85% to 6.7%, and the fraction of SNPs that are bi-allelic increases to 95.5%.

Whenever a bi-allelic SNP corresponds to a single substitution in the evolutionary history of the position, the SNP pattern provides an important piece of information about the phylogeny at that position in the alignment: whatever this phylogeny is, it must contain a split, that is a branch bi-partitioning the set of strains, such that all strains with one letter occur on one side of the split, and all strains with the other letter on the other side (*Figure 3*).

As illustrated in *Figure 3*, pairs of SNPs can either be consistent with a common phylogeny, that is columns *X* and *Y* or columns *Y* and *Z*, or they can be inconsistent with a common phylogeny, that is columns *X* and *Z*. The pairwise comparison of SNP columns for consistency with a common phylogeny is known as the four-gamete test and is very commonly used in the literature on sexual species, for example to give a lower bound on the number of recombination events in an alignment (*Hudson and Kaplan, 1985*). However, so far it has rarely been used for quantifying recombination in bacteria (*Lai and Ioerger, 2018* and (*Arnold et al., 2018*) being the only exceptions we are aware of). In the rest of this paper, we show how analysis of bi-allelic SNPs (which from now on we will just call SNPs) can be systematically used to quantify not only the overall amount of recombination in alignments, but also the relative rates with which different lineages have recombined.

Since all these analyses assume that bi-allelic SNPs correspond to single substitutions, it is important to quantify how accurate this assumption is. In particular, some apparent SNPs might correspond to sequencing errors rather than true substitutions and, more importantly, some bi-allelic SNPs may correspond to multiple substitution events (often called homoplasies). In the Materials and methods, we show that sequencing errors must be so rare that they can be safely neglected. In addition, to estimate the fraction of bi-allelic SNPs that correspond to homoplasies we analyzed the frequencies of columns with 1, 2, 3, and 4 different nucleotides using a simple substitution model,

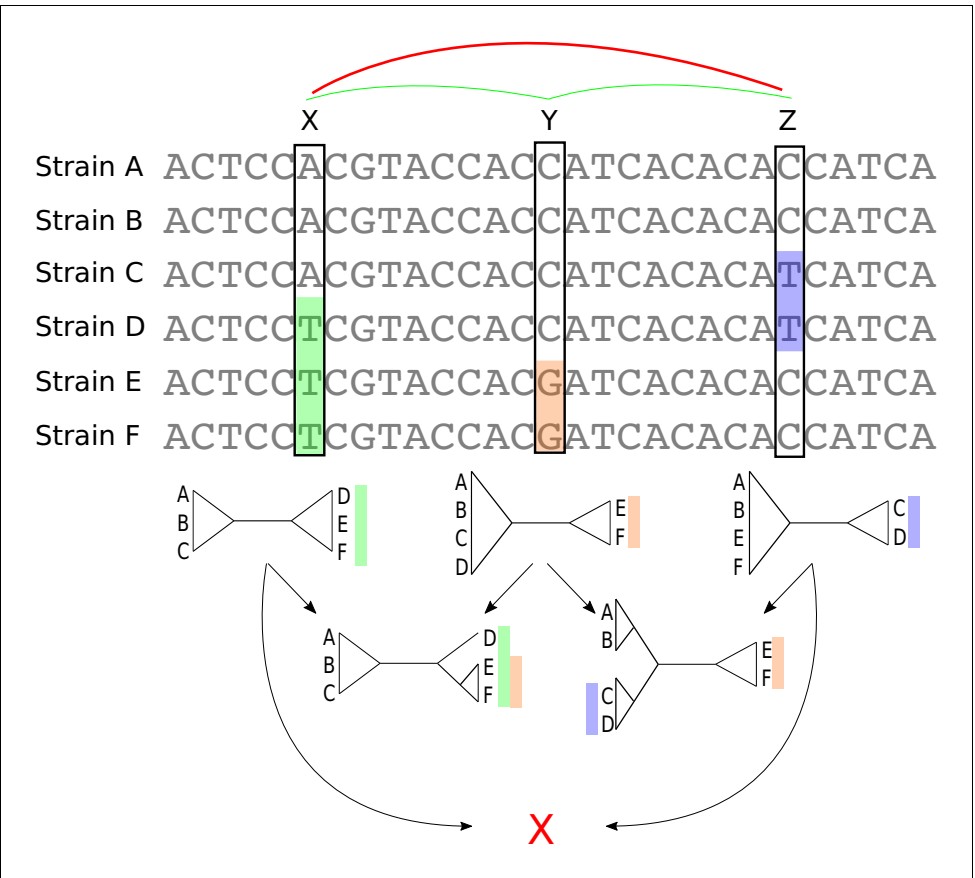

**Figure 3.** Bi-allelic SNPs correspond to phylogeny splits. A segment of a multiple alignment of 6 strains containing three bi-allelic SNPs, *X*, *Y*, and *Z*. Assuming that each SNP corresponds to a single substitution in the evolutionary history of the position, each SNP constrains the local phylogeny to contain a particular split, that is bi-partition of the strains, as illustrated by the three diagrams immediately below each SNP. In this example, the neighboring pairs of SNPs $(X, Y)$ and $(Y, Z)$ are both consistent with a common phylogeny and can be used to further resolve the phylogeny in the local segment of the alignment as shown in the second row with two diagrams. However, SNPs *X* and *Z* are mutually inconsistent with a common phylogeny (red cross at the bottom) indicating that somewhere between *X* and *Z* at least one recombination event must have occurred.

The online version of this article includes the following figure supplement(s) for figure 3:

**Figure supplement 1.** Comparison of the observed frequencies of columns with different numbers of nucleotides with predictions from a simple substitution model.

separately analyzing positions that are under least selection (third positions of fourfold degenerate codons) and positions under most selection (second positions in codons), and either including or excluding the outgroup (see Materials and methods, and *Figure 3—figure supplement 1*). These analyses indicate that only 2–6% of bi-allelic SNPs correspond to homoplasies. We confirmed the accuracy of this estimation procedure using our simulation data, for which the true fraction of homoplasies is 2.56% and our simple method estimates 2.43%.

In addition, to put an upper bound on how much our results could be affected by a small fraction of homoplasies, we developed a method that removes, from the core genome alignment, a given fraction of alignment columns that exhibit most inconsistencies with alignment columns in their neighborhood (see Materials and methods), and checked how much the various statistics that we calculate are altered when we remove either 5% or 10% of such potentially homoplasic positions. Finally, we also apply each of our analyses to the data from the simulations with known ground truth, to further validate the accuracy of our methods.

## SNP statistics are inconsistent with a single phylogeny

One of the key uses of a phylogeny is in describing the differences between the strains by an evolutionary dynamics that takes place along the branches of the phylogeny. However, for such an approach to apply, it is important that the large majority of evolutionary changes were indeed introduced along the branches of the phylogeny.

That there are significant differences between the SNP statistics as predicted by the core tree, and the actual SNP statistics observed in the data is already evident from the fact that the observed pairwise divergences between strains are systematically less than their divergences along the branches of the core phylogeny ( *Figure 4—figure supplement 1*, left panel). In addition, for many branches of the core tree, the observed number of SNPs corresponding to that branch is up to 100-fold lower than number of SNPs that are predicted to occur on that branch (*Figure 4—figure supplement 1*, right panel).

To test to what extent the observed evolutionary changes occurred along the branches of the core tree, we calculated the fraction of observed SNP splits that fall on the branches of the core phylogeny. Overall, 58% of the SNPs that are shared by at least two strains correspond to a branch of the core tree, whereas 42% clash with it (SNPs that occur in only a single strain are consistent with any phylogeny). However, this relatively high fraction results almost entirely from SNPs on the single branch connecting the outgroup to the other strains, which is responsible for almost 36% of all SNPs. When the outgroup is removed, only 27.4% of all SNPs are consistent with the core tree. Since the core tree was constructed using a maximum likelihood approach that assumes the entire alignment follows one common tree, we investigated to what extent the number of tree supporting SNPs can be improved by specifically constructing a tree to maximize the number of supporting SNPs (see Materials and methods). However, this only marginally improves the number of supporting SNPs by 0.1%.

Since the overall fraction of SNPs that correspond to branches of the core phylogeny is dominated by a few very common SNP patterns, it is more informative to assess to what extent each individual branch of the core tree is consistent with the observed SNPs. We thus calculated, for each

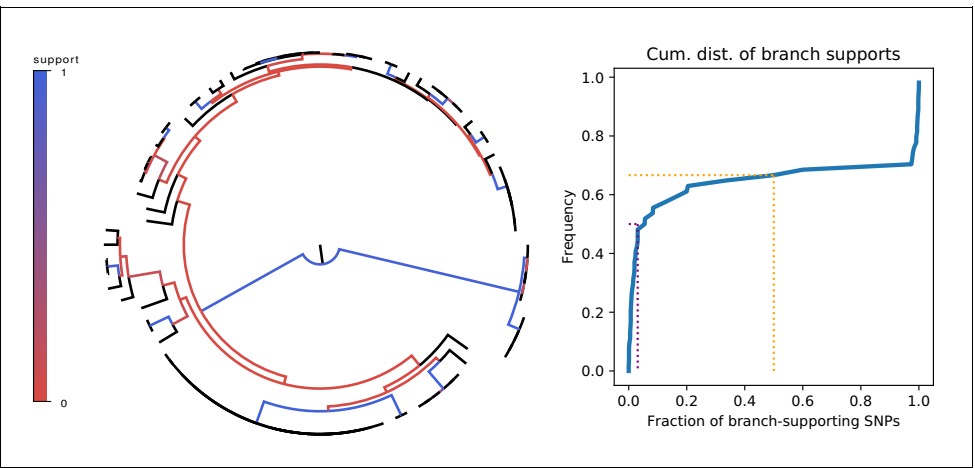

**Figure 4.** Most branches of the core genome tree are rejected by the statistics of individual SNPs. Left panel: Fraction of supporting versus clashing SNPs for each branch of the core tree. Right panel: Cumulative distribution of the fraction of supporting SNPs across all branches. The purple and orange dotted lines show the median and the frequency of branches with 50% or less support, respectively.

The online version of this article includes the following figure supplement(s) for figure 4:

**Figure supplement 1.** The pairwise distances and number of SNPs on each branch as predicted by the core tree do not match the pairwise distances and SNP numbers observed in the data.

**Figure supplement 2.** Homoplasies do not significantly effect the fractions of supporting SNPs for *E.coli*.

**Figure supplement 3.** Distributions of SNP support for the data from the simulations.

**Figure supplement 4.** Supporting versus clashing SNPs for trees that were built bottom-up, while minimizing SNP clashes.

**Figure supplement 5.** Quartets of roughly equidistant strains have no consensus phylogeny.

branch in the core tree, the number of supporting SNPs $S$ that match the split, and the number of clashing SNPs $C$ that are inconsistent with the split, to calculate the fraction $f = S/(S + C)$ of SNPs supporting the branch. *Figure 4* shows that, for two-thirds of the branches, there are more clashing than supporting SNPs. Moreover, for half of the branches in the core tree, the fraction of supporting SNPs is less than 5%, that is there are at least 20-fold more clashing than supporting SNP columns. To check to what extent homoplasies may affect these statistics, we calculated the same statistics on core genome alignments from which either 5% or 10% of potentially homoplasic sites were removed, and observed that the distribution of SNP support is almost unchanged (*Figure 4—figure supplement 2*).

To further elucidate the meaning of these SNP support distributions, we calculated the same statistics for the simulations with different recombination rates, including on alignments where 5% or 10% of potentially homoplasic columns were removed (*Figure 4—figure supplement 3*). In general the removal of 5% or even 10% of potentially homoplasic sites has only a minor effect on the distribution of SNP support except for the simulations without recombination. When there is no recombination all inconsistencies are due to homoplasies, and we indeed see that all branches are fully supported after 5% of potential homoplasic sites have been removed. For a very small recombination rate of $\rho/\mu = 0.001$, most branches in the core tree still have strong SNP support but already at a recombination rate of $\rho/\mu = 0.01$ more than 80% of the branches are supported by less than half of the informative SNPs, and half of the branches have less than 20% support. It is instructive to compare this result with the fractions of the genome that are clonally inherited along each branch, that is the bottom left panel in *Figure 1—figure supplement 3*, and the pair statistics (middle row of *Figure 2—figure supplement 1*) for this simulation. We see that, although there is some recombination in most branches, most of the genome is clonally inherited for all but the long inner branches. However, even though most of the genome is clonally inherited along most branches, for most pairs the large majority of the SNPs derive from recombination (middle panel in *Figure 2—figure supplement 1*). This means that, at $\rho/\mu = 0.01$, we are in a parameter regime where most of the genome is still clonally inherited, and the clonal tree can thus also be reconstructed from the data, but the large majority of the differences between strains already derive from recombination events. This shows that, even if recombination is rare enough that the clonal tree can be successfully reconstructed from the genome sequences, it might already be incorrect to assume most genomic changes were introduced along the branches of this clonal phylogeny.

For recombination rates of $\rho/\mu = 0.1$ or larger, recombination dominates completely in that there are essentially no branches with significant SNP support. The distribution of support for *E. coli* does not look like any of the distributions from the simulations, but could be described as a hybrid of one third of branches with strong clonal support and two thirds of branches dominated by recombination. Besides the branch to the outgroup, all supported branches lead to groups of highly similar strains near the bottom of the tree. We thus wondered if it would be possible to construct well supported subtrees for clades of closely related strains near the bottom of the tree. We devised a method that builds subtrees bottom-up by iteratively fusing clades so as to minimize the number of clashing SNPs at each step (see Materials and methods and *Figure 4—figure supplement 4*, left panel). As shown in *Figure 4—figure supplement 4*, while the fraction of clashing SNPs is initially low, it rises quickly as soon as the average divergence within the reconstructed subtrees exceeds $10^{-4}$, which is more than 100-fold below the typical pairwise distance between *E. coli* strains. Thus, while some groups of very closely related strains that have a recent common ancestor can be unambiguously identified, only a minute fraction of the overall sequence divergence falls within these groups, and the bulk of the sequence variation between the strains is not consistent with a single phylogeny.

To assess whether any of these statistics could be skewed due to a few strains with aberrant behavior, we also investigated whether SNPs are more consistent with a dominant phylogeny when we do not consider all strains, but only subsets of the strains. We focused on the smallest subsets of strains that have meaningfully different phylogenetic tree topologies. For a quartet of strains $(I, J, M, N)$, there are three possibly binary trees, that is with $(I, J)$ and $(M, N)$ nearest neighbors, with $(I, M)$ and $(J, N)$, or with $(I, N)$ and $(J, M)$ (see *Figure 4—figure supplement 5*). We selected quartets of roughly equidistant strains and checked, for each quartet, whether the SNPs clearly supported one of the tree possible topologies. However, we find that alternative topologies are always

supported by a substantial fraction of the SNPs, and that for most quartets the most supported topology is supported by less than half of the SNPs (*Figure 4—figure supplement 5*).

In summary, consistent with the picture that emerged from our analysis of pairs of strains, most of the differences between the *E. coli* strains did not occur along the branches of a single phylogeny. This suggests that, rather than describing the relationships between the strains by a single phylogeny, we should think of multiple different phylogenies occurring along the genome alignment.

## The phylogeny changes every few SNPs along the core alignment

So far, we have analyzed SNP consistency without regard to their relative positions in the alignment. We now analyze to what extent mutually consistent SNPs are clustered along the alignment. In particular, we calculate the lengths of segments along the alignment that are consistent with a single phylogeny.

We first assessed the length-scale over which phylogenies are correlated by calculating a standard linkage disequilibrium (LD) measure as a function of distance along the alignment (*Figure 5A* and Materials and methods). LD drops quickly over the first 100 base pairs and becomes approximately constant at distances beyond $200 - 300$ base pairs, indicating that segments of correlated phylogenies are much shorter than the typical length of a gene. Very short linkage profiles were recently also observed in thermophilic *Cyanobacteria* isolated in Yellowstone National Park (*Rosen et al., 2015*). Instead of using correlation between SNPs at different distances, one can also calculate the probability for a pair of SNPs to be consistent with a common phylogeny, as a function of their genome distance (*Arnold et al., 2018*). As shown in in *Figure 5—figure supplement 1*, like LD, pairwise compatibility of SNPs also drops quickly over the first 100 base pairs. Note, however, that even at large distances the pairwise compatibility of SNPs is close to 90%. The reason for this is that most SNPs are shared by only a small subset of strains, and as long as two SNPs are shared by non-overlapping subsets of strains, they will be compatible with a tree. In order to more efficiently detect recombination using SNP compatibility, we need to check for the mutual consistency of *all* SNPs within a given segment of the alignment.

Starting from each SNP *s*, we determined the number of consecutive SNPs *n* that are all mutually consistent with a common phylogeny. As shown in *Figure 5B*, the distribution of the lengths of tree-compatible stretches has a mode at $n = 4$, and stretches are very rarely longer than $n = 20$

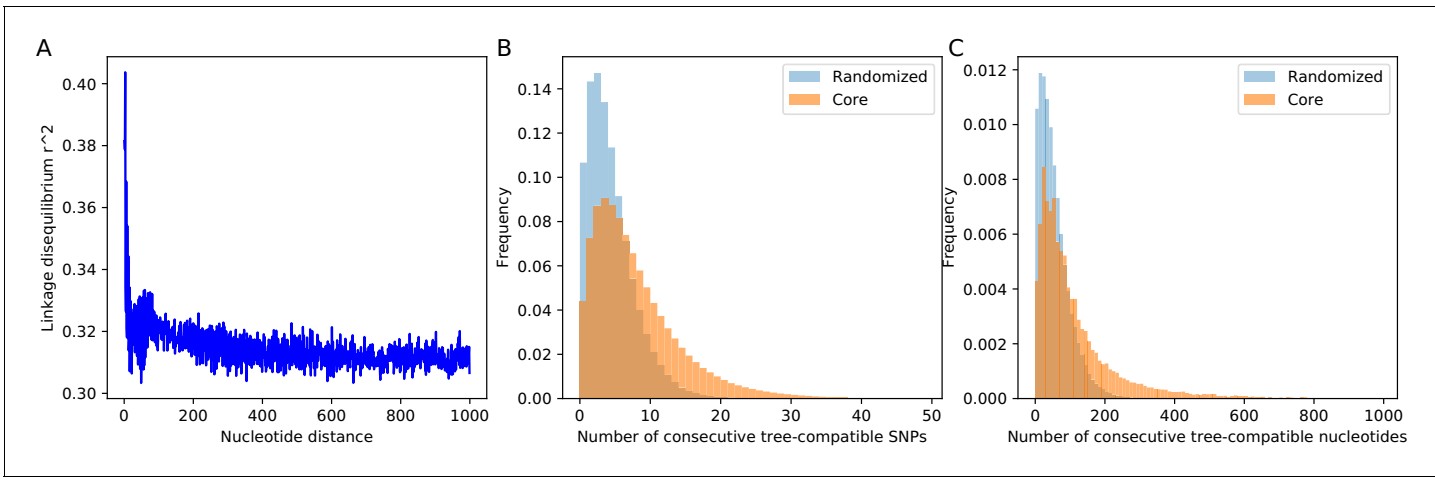

**Figure 5.** SNP compatibility along the core genome alignment shows tree-compatible segments are short. (**A**) Linkage disequilibrium (squared correlation, see Materials and methods) as a function of the separation of a pair of columns in the core genome alignment. (**B**) Probability distribution of the number of consecutive SNP columns that are consistent with a common phylogeny for the core genome alignment (orange) and for an alignment in which the positions of all columns have been randomized (blue). (**C**) Probability distribution of the number of consecutive alignment columns consistent with a common phylogeny for both the real (orange) and randomized alignment (blue).

The online version of this article includes the following figure supplement(s) for figure 5:

**Figure supplement 1.** Pairwise SNP compatibility as a function of genomic distance.
**Figure supplement 2.** Lengths of tree-compatible segments for the data from the simulations.

consecutive SNPs. In terms of number of base pairs along the genome, tree-compatible segments are typically just a few tens of base pairs long, and very rarely more than 300 base pairs (*Figure 5C*). Thus, stretches of tree-compatible segments are very short. For comparison, we also calculated the distribution of tree-compatible segment lengths in an alignment where the positions of all columns have been completely randomized and observe that these are still a bit shorter (blue distributions in *Figure 5B and C*). Thus, while there is some evidence that neighboring SNPs are more likely to be compatible than random pairs of SNPs, this compatibility is lost very quickly, typically within a handful of SNPs.

To elucidate how the recombination-to-mutation ratio determines the lengths of tree-compatible stretches for the simulations, we calculated the same distribution for the data from the simulations with different recombination rates. In addition, to assess the affect of homoplasies on the distribution of the lengths of tree-compatible stretches, we also calculated these distributions for alignments from which 5% or 10% of potentially homoplasic sites were removed (*Figure 5—figure supplement 2*). In addition, *Appendix 1—table 1* shows the average length of tree-compatible segments for each of these datasets. We see that removal of homoplasies only has a significant effect on the distribution of tree-compatible stretches for $\rho/\mu \leq 0.01$. That is, homoplasies only significantly affect the distribution of tree-compatible stretches when $\rho/\mu$ is less than the rate of homoplasies (which is 2.5%). For the *E. coli* data, removal of homoplasies has only a minor effect, that is the average number of consecutive tree-compatible SNPs increases from 7.6 to 8.8 when 5% potentially homoplasic sites are removed. Although there is of course no reason to assume that the simple evolutionary dynamics of any of our simulations realistically describes the genome evolution of the *E. coli* strains, we note that the distribution of tree-compatible stretches for *E. coli* is between what is observed for simulations with $\rho/\mu = 0.3$ and $\rho/\mu = 1$. Notably, as is clear from *Figure 1—figure supplement 3*, *Figure 2—figure supplement 1*, and *Figure 4—figure supplement 3*, when $\rho/\mu \geq 0.3$ the evolution along every branch of the phylogeny is almost completely dominated by recombination.

In summary, our analysis shows that, as one moves along the core genome alignment, the phylogeny changes typically every 5–10 SNP columns, that is every 50–100 nucleotides. We next use this to put a lower bound on the ratio of the number of phylogeny changes to mutations in the alignment.

## A lower bound on the ratio of phylogeny changes to substitution events

Every time inconsistent SNP columns are encountered as one moves along the core genome alignment, the local phylogeny must change. For example, somewhere between columns *X* and *Z* in *Figure 3* the phylogeny must change. This in turn implies that the start (or end) of at least one recombination event must occur between columns *X* and *Z*. By going along the core genome, and determining the minimum number of times the phylogeny must change, one can thus derive a lower bound on the total number of recombination events and, in the study of sexually reproducing species, this has been a standard method to put a lower bound on the number of recombination events within a genome alignment (*Hudson and Kaplan, 1985*) (see Materials and methods). Using this we find that the phylogeny must change at least $C = 43'575$ times along the core phylogeny. However, this neglects that some of the inconsistencies may result from homoplasies. To correct for this, we remove 5% of potential homoplasic positions by removing 5% of the SNP columns that are most inconsistent with neighboring columns (Materials and methods) and find that the phylogeny must still change $C = 34'030$ times along this 5% homoplasy-corrected alignment. Because homoplasies are relatively rare, the number of bi-allelic SNPs in the alignment is a good estimate for the total number of mutations in the alignment and the ratio $C/M$ thus provides a lower bound for the ratio between the total number of phylogeny changes and substitutions that occur along genome alignment. For the 5% homoplasy-corrected alignment we obtain a ratio $C/M = 0.129$.

Apart from the full alignment, we can calculate the lower bound on the ratio of phylogeny changes to substitution events $C/M$ for any subset of strains. *Figure 6* shows the ratio $C/M$ for random subsets of our 92 strains as a function of the number of strains in the subset, using again the 5% homoplasy-corrected alignment. For comparison, *Figure 6—figure supplement 1* also shows the same results for the full alignment, which shows ratios $C/M$ that are about 20% higher.

We see that, for small subsets of strains, the ratio $C/M$ shows substantial fluctuations. For example, for subsets of $n = 10$ strains, the ratio $C/M$ ranges from 0.026 to 0.112, with a median of 0.075. However, as the number of strains in the subset increases, the ratio converges to the value

$C/M = 0.129$ and for large subsets of strains there is little variation in the ratio $C/M$. Thus, for alignments of large sets of strains, the phylogeny must change at least every $7 - 8$ SNPs.

## $C/M$ within phylogroups

Apart from random subsets of strains, we also calculated $C$, $M$ and $C/M$ for subsets of strains from the same phylogroup (*Appendix 1—table 2*, and see *Figure 1—figure supplement 1* for the phylogroup annotation). The ratios $C/M$ that are observed for the phylogroups increase with the overall divergence within the phylogroup. For phylogroups that are more than 1% diverged (B1, B2, and D), the ratio $C/M$ is close to that of the full alignment, and there are at least thousands of phylogeny changes along their sub-alignments. The two phylogroups with lower divergence, that is A with divergence 0.0024 and the outgroup with divergence 0.003 have lower ratios $C/M$. The ratio is particularly low for the outgroup O for which only two phylogeny changes are detected. While this may suggest that recombination rates are particularly low within lineages of the outgroup, it should be noted that such low values of $C/M$, while rare, were also observed for some random subsets of strains of the same size (*Figure 6*).

## $C/M$ for the simulation data

Note that, because each recombination event introduces at most two phylogeny changes (one at the start of the recombined segment and one at its end), $C/2$ is a lower bound on the number of recombination events that occurred in the evolutionary history of an alignment. However, this lower bound may significantly underestimate the true number of recombination events. To obtain more insight into the relationship between this lower bound and recombination rates, we compared the ratios $C/M$ of each simulated dataset (using the 5% homoplasy-corrected alignment) with the ratio $\rho/\mu$ of recombination and mutation rate used in the simulation (*Figure 6—figure supplement 2*). The results show that when recombination rates are very low, that is $\rho/\mu \leq 0.01$, the ratio $C/(2M)$ is almost exactly equal to $\rho/\mu$. In this regime recombination events are so sparse on the alignment that many SNP columns occur between every two consecutive phylogeny changes, and this causes almost every phylogeny change to introduce an inconsistency. Since each recombination event introduces two breaks, $C/M$ equals twice the number of recombinations per mutation $\rho/\mu$. However, as $\rho/\mu$ increases, fewer SNP columns occur between consecutive phylogeny changes, and more and more of the phylogeny changes go undetected because they do not introduce inconsistencies between the SNPs. Consequently, the ratio $C/M$ becomes systematically lower than $\rho/\mu$ and the difference can become very large. For example, at $\rho/\mu = 1$, the observed ratio $C/M$ is almost tenfold lower than $\rho/\mu$. Note also that since the lower bound $C/M$ cannot exceed 1 (i.e. a phylogeny change at every SNP), the ratio $\rho/\mu$ can exceed $C/M$ by arbitrarily large factors at high $\rho/\mu$.

## Each genomic position has likely been overwritten hundreds of times by recombination

We can also provide a rough estimate for the average number of times $T$ that a randomly chosen position in the core genome alignment has been overwritten by recombination in its history, that is since the genetic ancestors of the position in the alignment diverged from a common ancestor.

If $L$ is the total genome length, $L_r$ the average length of recombination segments, and $C/2$ the lower bound on the number of recombination events in the alignment, then a lower bound on the average number of times positions in the genome have been overwritten by recombination is $T = L_r C/(2L)$. For the *E. coli* data with $C = 34'030$ phylogeny changes for the 5% homoplasy-corrected alignment of length $L = 2'756'541$, and the value of $L_r \approx 31'000$ base pairs that we estimated previously (*Figure 2J*), we obtain $T \approx 190$. That is, on average each position in the genome has been overwritten at least 190 times by recombination. We note that, since this lower bound for $T$ is proportional to the estimated average length of recombination segments $L_r$, it of course depends on this estimate as discussed in Appendix 2.

In the simulations, we specifically tracked the number of times each position in the alignment was overwritten by recombination along each branch of the clonal phylogeny, and we thus know precisely how many times each position of the alignment was overwritten by recombination in its evolutionary history. *Figure 6—figure supplement 3* shows the distribution of the number of times each position in the alignment was overwritten for the simulations with $\rho/\mu$ ranging from 0.001 to 10. We

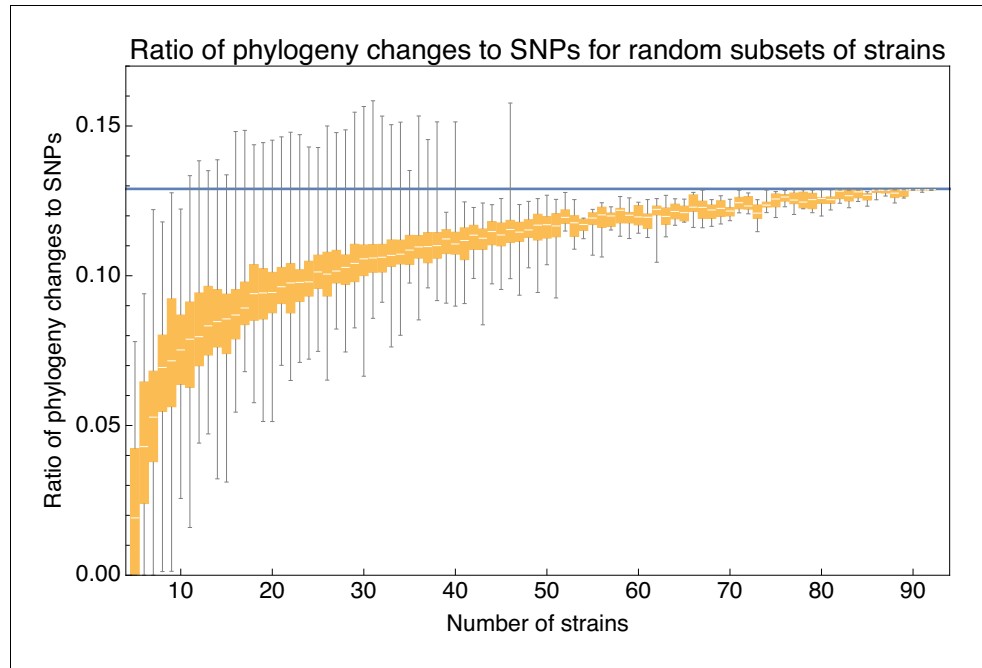

**Figure 6.** Ratio $C/M$ of the minimal number of phylogeny changes $C$ to substitutions $M$ for random subsets of strains using the alignment from which 5% of potentially homoplasic positions have been removed. For strain numbers ranging from $n = 4$ to $n = 92$, we collected random subsets of $n$ strains and calculated the ratios $C/M$ of phylogeny changes to SNPs in the alignment. The figure shows box-whisker plots that indicate, for each strain number $n$, the 5th percentile, first quartile, median, third quartile, and 95th percentile of the distribution of $C/M$ across subsets. The blue line shows $C/M = 0.129$.

The online version of this article includes the following figure supplement(s) for figure 6:

**Figure supplement 1.** Ratios $C/M$ for homoplasy-corrected and full alignments of random subsets of strains.

**Figure supplement 2.** Lower bound $C/M$ on the ratio of phylogeny changes to mutations, versus the ratio $\rho/\mu$ of recombination to mutation rate, for the data of the simulations.

**Figure supplement 3.** Distributions of the number of times each position in the alignment has been overwritten by recombination for the data of the simulations.

**Figure supplement 4.** Comparison of the estimated and true average number of times each position in the alignment has been overwritten by recombination for the data of the simulations.

---

see that, in line with all previous analyses, positions that are clonally inherited along the whole tree only occur for the lowest recombination rate of $\rho/\mu = 0.001$, at $\rho/\mu = 0.01$ each position has been overwritten $5 - 25$ times, and at $\rho/\mu = 0.1$ each position has already been overwritten $120 - 200$ times. For each $\rho/\mu$ we calculated the average number of times $T_{\text{true}}$ that positions were overwritten by recombination and compared this with the estimated number of times using the simple estimate $T_{\text{est}} = L_r C/(2L)$. *Figure 6—figure supplement 4* shows that, while the estimate is in fact quite accurate at very low mutation rate $\rho/\mu = 0.001$, as the recombination rate increases the estimate severely underestimates the true number of times positions in the alignment were overwritten by recombination, for example at $\rho/\mu = 1$ the true number $T_{\text{true}} = 1561$ is more than twenty times as large as the estimated number $T_{\text{est}} = 67$. Thus, for the *E. coli* data it is certainly not implausible that each position in the alignment was in fact overwritten by recombination more than 1000 times. As discussed in Appendix 2, a rough estimate of $T$ can also be derived from an analysis of the pair statistics of *Figure 2*, and this estimate suggests that $T$ lies somewhere between $T = 200$ and $T = 500$, which is consistent with our lower bound based on $C$.

The observed ratio $C/M$ for the *E. coli* alignments is very similar to the value of $C/M$ observed in the simulations with $\rho/\mu = 1$ (*Figure 6—figure supplement 2*). Although it is tempting to conclude from this that the ratio of recombination to mutation rate must be close to one for the *E. coli* strains, such a conclusion would be unwarranted. The evolutionary dynamics of the simulations makes several strong simplifying assumptions, that is that the clonal phylogeny is drawn from the Kingman

coalescent process, and that the population is completely mixed so that all strains are equally likely to recombine. Both these assumptions may not apply to the evolution of *E. coli* in the wild. Indeed, we will see below that there is strong evidence that relative recombination rates vary highly across lineages so that instead of a single recombination rate, there is a wide distribution of recombination rates between different lineages. Therefore, it could be misleading to describe *E. coli*'s evolution by a single recombination rate $\rho$ and we instead focus on providing a lower bound $C/M$ on the number of phylogeny changes per SNP column, which is a meaningful quantity independent of the precise evolutionary dynamics that caused the substitutions and phylogeny changes, and can be calculated independently for any subset of strains.

## Other bacterial species show similar patterns of recombination-dominated genome evolution

To investigate to what extent the observations we made for *E. coli* generalize to other species of bacteria, we selected five additional species from different bacterial groups for which sufficiently many complete genome sequences of strains were available, and used Realphy to obtain a core genome alignment of the strains for each species (see *Appendix 1—table 3* for a list of the species, the number of strains, and other core genome statistics for each species). We then performed most of the analyses that we presented above for *E. coli* on each of these core alignments. *Figure 7* presents a summary of the results that we observe across the species.

*Figure 7A* shows the cumulative distributions of pairwise divergences between strains for all species. We see that, while among our *E. coli* strains that were sampled from a common habitat there is a small percentage of very close pairs with divergence below $10^{-6}$, for the strains of the other species the closest pairs are at divergence $10^{-5}$. With the exception of *M. tuberculosis*, where the median pair divergence is around $10^{-4}$, the median pairwise divergence in all other species is around $10^{-2}$ or larger. The vertical lines in *Figure 7A* indicate the critical divergences, for each species, where half of the alignment is recombined. With the exception of *M. tuberculosis*, where all pairs are mostly clonal, the critical divergences lie in a fairly narrow range of 0.003–0.01. *Figure 7B* shows the cumulative distributions, across pairs of strains, of the fraction of the alignment that is clonally inherited, that is as for *Figure 2I* for *E. coli*. Note that, for all species except *M. tuberculosis*, the large majority of the pairs is fully recombined, ranging from about 15% of pairs with a substantial fraction of clonally inherited DNA for *S. aureus*, to virtually no pairs with clonally inherited DNA for *H. pylori*. Thus, we see that for almost all species the situation is similar to what we observed in *E. coli*: for most pairs the distance to their common ancestor cannot be estimated from their alignment, because the entire alignment has been overwritten by recombination events. Note also that, for all species, there is only a relatively small fraction of pairs that lie in the partially recombined regime (yellow segment in *Figure 7B*).

For *E. coli* we found that, even for close pairs for which a substantial fraction of the genome was clonally inherited, most of the SNPs between them still derive from recombination (*Figure 2H*). We find that, with the exception of *M. tuberculosis*, this applies to the other species as well. *Figure 7C* shows, for each species, the fraction of all SNPs that derive from recombination, for pairs of strains that are at the critical divergence where half of the alignment is recombined. Even though this critical divergence occurs for pairs that are relatively close compared to the typical distance between pairs, for all species more than 90% of the SNPs derive from recombination. That is, we also see that for all five species the divergence between close strains is dominated by SNPs that are introduced through recombination.

Another way to quantify to what extent the observed sequence variation is consistent with evolution along the branches of the core genome each branch of the core tree. *Figure 7D* summarizes the distributions of support of the branches of the core tree as violin plots, that is as shown for *E. coli* as a cumulative distribution in *Figure 4*. In *E. coli* most branches have many more SNPs that reject the split than support it, and even stronger rejection of the branches of the core tree are observed for *B. subtilis* and *H. pylori*. For the other species, an almost uniform distribution of branch support is shown, that is for these species there are roughly as many branches that are strongly supported by the SNPs, strongly rejected by the SNPs, or supported and rejected by roughly equally many SNPs.

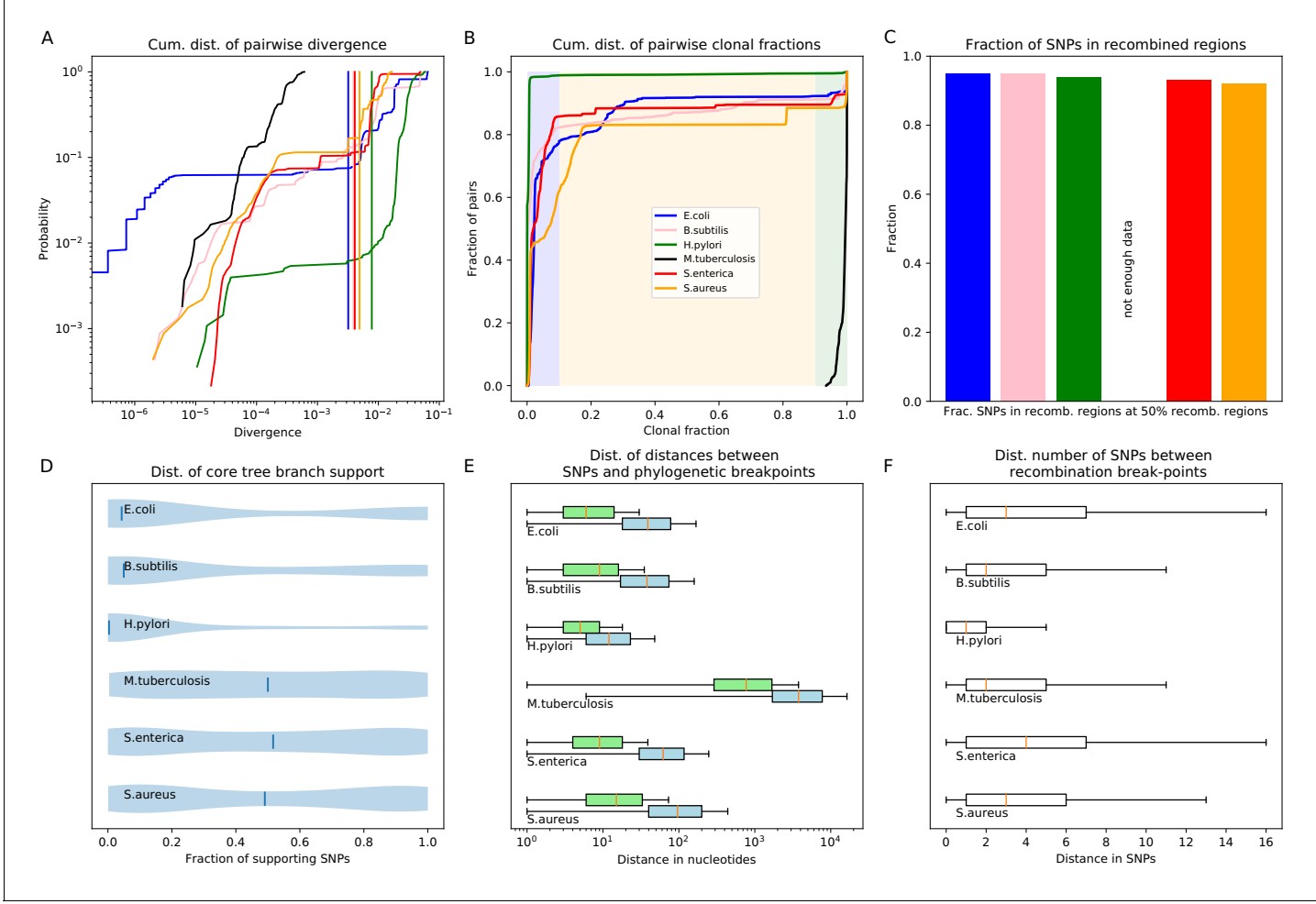

**Figure 7.** Quantification of the importance of recombination across species. (A) The cumulative distribution of pairwise divergences is shown as a different colored line for each species (see legend in panel B). Both axes are shown on logarithmic scales. The vertical lines in corresponding colors show the critical divergences at which half of the genome is recombined for each species. (B) Cumulative distributions of clonal fractions across the pairs of strains for each species, with the blue, yellow, and green shaded regions indicating the fully recombined, partially recombined, and mostly clonal regimes, respectively, that is analogous to *Figure 2I* (C) For each species, the height of the bar shows the fraction of SNPs that fall in recombined regions for pairs of strains for which half of the genome is recombined, that is see *Figure 2H*. (D) The violin plots show, for each species, the distribution of branch support, that is the relative ratio of SNPs supporting versus clashing with each branch split, analogous to the right panel of *Figure 4*. The blue lines correspond to the medians of the distributions. (E) Box-whisker plots showing the 5, 25, 50, 75, and 95 percentiles of the distributions of nucleotide distances between consecutive SNPs (green) and phylogeny breakpoints (blue, that is analogous to *Figure 5C*), for each species. The axis is shown on a logarithmic scale. (F) Box-whisker plots of the distributions of the number of consecutive SNPs in tree-compatible segments, that is analogous to *Figure 5B*.

The online version of this article includes the following source data for figure 7:

**Source data 1.** List of database accessions of the genomes used.

*Figure 7E* summarizes, for each species, the distribution of distances between SNPs along the core alignment as box-whisker plots (green) as well as the distribution of distances between phylogeny breakpoints (blue, that is as shown in *Figure 5C* for *E. coli*. The figure shows that, with the exception of *M. tuberculosis*, the inter-SNP distances range from a few to a few dozen base pairs, with a median inter-SNP distance of 4 (*H. pylori*) to 15 (*S. aureus*) base pairs. For these five species, the median distances between phylogeny breakpoints range from around 10 (*H. pylori*) to about 100 base pairs for *S. aureus*. Note that, for all species, the left tails of the distributions stretch to very short distances between breakpoints, whereas distances between breakpoints of more than 200 bps are very rare for all these five species. Thus, for these species the segments that are consistent with a single phylogeny are always much shorter than the typical length of a gene. In contrast, for *M.*

*tuberculosis* both the distances between SNPs and the distances between breakpoints are almost two orders of magnitude larger, indicating that these strains are much more closely related than the strains of the other species.

*Figure 7F* shows box-whisker plots for the distributions of the number of consecutive SNPs between breakpoints, as was shown for *E. coli* in *Figure 5B*. We see that for all species, including *M. tuberculosis*, there are typically less than a handful of SNPs in a row before a phylogeny breakpoint occurs, and very rarely more than a dozen SNPs. The smallest number of SNPs per breakpoint is observed for *H. pylori*, that is typically less than 2 SNPs per breakpoint, but the range of SNPs per breakpoint is very similar across all species.

Finally, *Appendix 1—table 4* shows the fraction of alignment columns that are SNPs, the lower bound $C$ on the number of phylogeny changes, and the lower bound $C/M$ on the ratio of phylogeny changes to SNP columns, for each of the six species. We see that with the exception of *H. pylori*, which has a high ratio $C/M \approx 0.3$, and *M. tuberculosis* which has significantly lower ratio $C/M \approx 0.08$, the other species have ratios $C/M$ similar to that observed for *E. coli*. Note that the high rate of recombination that we infer for *H. pylori* is consistent with the consensus in the literature that genome evolution is dominated by recombination in this species, for example (*Suerbaum et al., 1998*).

In contrast, the consensus in the field of *M. tuberculosis* evolution is that essentially no recombination occurs in this species. Although there have been reports of evidence for recombination (*Liu et al., 2006*; *Namouchi et al., 2012*; *Phelan et al., 2016*), follow-up analyses suggested that these may well be a result of problems with genome assemblies and genome alignments (*Godfroid et al., 2018*). Although our statistics indicate that evolution in *M. tuberculosis* is predominantly clonal, we do find a significant fraction of SNPs that clash with the whole genome tree. However, because SNPs are relatively rare in *M. tuberculosis*, that is on average one SNP every 1200 base pairs, SNPs that clash with the core phylogeny occur only once every 4700 base pairs, and it is not hard to imagine that problems with the genome assemblies of just a few strains could cause artefactual SNPs to occur at that rate. Indeed, we found that 5 of the *M. tuberculosis* genomes in our dataset have recently been retracted from the database because of evidence of contamination. Notably, we found that these strains were responsible for the large majority of the clashing SNPs. That is, after removal of these five strains, clashing SNPs now only occur once every 12'700 base pairs on average. Since it is not implausible that there may be problems with the genome assemblies of one or more of the remaining strains, this suggests that these remaining clashing SNPs may well be artefacts as well. In conclusion, although we cannot definitively conclude there is no recombination at all in *M. tuberculosis*, our analysis confirms that it must be rare.

In summary, with the exception of *M. tuberculosis*, all other species show the same pattern as *E. coli* with genome evolution being dominated by recombination. For most pairs, no DNA in their alignment was clonally inherited, even for close pairs most of the SNPs derive from recombination events, the phylogeny changes thousands of times along the core genome alignment, typically within a handful of SNPs, and each position in the alignment has been overwritten by recombination many times in its history.

## Phylogenetic structures reflect population structure

All our results so far show that the core tree cannot represent the evolutionary relationships between the strains and that a large number of different phylogenies occurs across the alignment. It may thus seem all the more puzzling that, when trees are constructed from sufficiently many genomic loci, the core tree reliably emerges (*Figure 1*, bottom).

As we mentioned in the introduction, some researchers interpret this convergence to the core tree to mean that the core tree must correspond to the *clonal* phylogeny of the strains. The interpretation is that the SNP patterns in the data are a combination 'clonal SNPs' that fall on the clonal phylogeny, plus a substantial number of 'recombined SNPs' that act so as to introduce noise on the clonal phylogenetic signal. In this interpretation, trees build from individual loci can differ from the core tree because the 'recombination noise' can locally drown out the clonal signal, but once sufficiently many loci are considered, this recombination noise 'averages out' and the true clonal structure emerges. However, this interpretation rests on two assumptions that do not necessarily hold. First, in order for the effects of recombination to 'average out' when many loci are considered, one has to assume that the recombination 'noise' has no systematic structure itself. That is, one has to

assume that there is no population structure, that is that all lineages are equally likely recombine with each other. Second, for the clonal phylogeny to emerge one has to also assume that there are sufficiently many loci that have not been affected by recombination. However, the results above show that recombination is so common that each locus has been overwritten many times by recombination, that is there are essentially no loci that are unaffected by recombination.

Note that if the phylogenies along the alignment were a combination of clonal and randomly recombined SNPs, then one would expect that removal of all the clonal SNPs would destabilize the core tree. However, if we remove all SNP columns from the core genome alignment that correspond to branches of the core tree, and then reconstruct a phylogeny from this edited alignment, the resulting tree is still highly similar to the core tree (*Figure 8—figure supplement 1*). That is, we obtain a very similar tree from this edited alignment, even though virtually *all* SNPs in this alignment are inconsistent with this tree. This confirms that the structure in the core tree does not derive from the subset of SNPs that fall on the core tree, but rather reflects overall statistical properties of all SNPs.

We propose that, rather than thinking of the SNP patterns as deriving from a single 'true' phylogeny plus unstructured recombination noise, the statistics of the SNP patterns reflect structure in the recombination process itself. In particular, we propose that because of population structure, that is the fact that different lineages recombine at different rates, the recombination process is itself structured, and that the distribution of phylogenies that occur along the alignment reflects this population structure. To illustrate this interpretation, we will use data from a species for which there is no question that recombination dominates and no 'clonal' structure can exist.

## Phylogenies reconstructed from human genome sequences also exhibit robust phylogenetic structure

We randomly selected 40 genomes from the 1000 Genomes project, used PhyML to build a phylogenetic tree from chromosomes 1–12 of these genomes, and then investigated to what extent branches in this tree also occur in trees build from random subsets of the genomic loci. As shown in *Figure 8*, a robust phylogeny also emerges for the human data. In particular, individuals with the same geographic ancestry consistently form clades in the trees build from large numbers of loci.

However, it is of course clear that this phylogeny cannot correspond to the 'clonal phylogeny' of the human sequences, because there is no such thing as a 'clonal phylogeny' for the human sequences. Perhaps, the closest analog of a 'clonal tree' for the human data would be the phylogeny of the strictly maternal lineage. However, since at each generation roughly half of the autosomal chromosomes derives from the mother, and half from the father, there are few if any loci in the genome that follow this strictly maternal lineage, that is almost every locus was paternally inherited in at least *some* generations. Instead, it is clear that there are many different phylogenies along the genome, each with different ancestries.

The reason that a robust phylogeny still emerges is that the different phylogenies that occur along the genome are not completely random. That is, human populations are not completely mixed but people are more likely to mate with others from the same geographic area. Because of this, recombination tends to occur among people of the same geographic area, and because of this population structure the phylogenies that occur along the alignment of a set of human genomes are not sampled uniformly from all possible phylogenies, but some topologies are more likely to occur than others. In particular, individuals from the same geographic area will have recent ancestors in a larger fraction of the phylogenies along the genome than individuals from different geographic areas. Indeed, a simple principal component analysis of SNP statistics in genomes of European ancestry recapitulates geographic structure in remarkable detail (*Novembre et al., 2008*).

Of course, there are good reasons that human population geneticists do not typically represent the ancestral population structure of human sequences by a single phylogeny. We know that many different phylogenies occur along the genome and it would be misleading to pretend that this distribution of phylogenies can be summarized by a single tree. However, if one insists on representing the population structure by a single tree and asks PhyML to build a single phylogeny from many loci of the human sequences, then one does get convergence to a common phylogeny for the human data as well. The reason one gets this convergence is because asking for a single tree is analogous to asking for the 'average' of a distribution. The average of almost any distribution becomes very

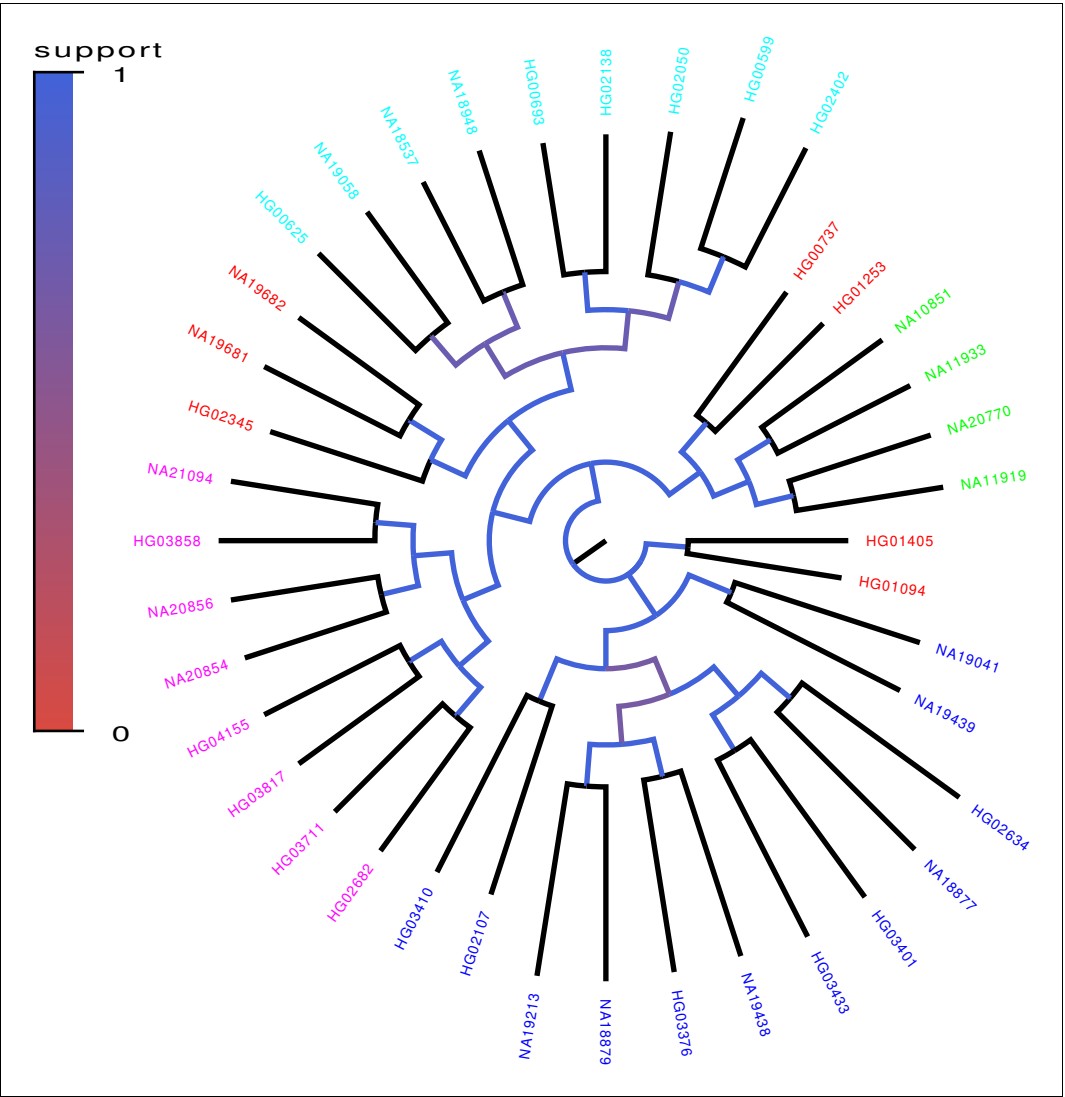

**Figure 8.** Core tree build by PhyML from the sequences of chromosomes 1–12 of 40 randomly chosen human genomes of the 1000 Genome project. The colors indicate what fraction of the time each split in the core tree occurred in trees build from random subsets of half of the genomic loci. The colors on the leaves indicate the annotated ancestry of the individuals, with blue corresponding to African ancestry, green to European ancestry, red to (South) American ancestry, cyan to East Asian ancestry, and magenta to South Asian ancestry. Individuals from the same geographic area reliably form clades in the tree with the exception of South Americans, two of which form an outgroup of the Europeans, three an outgroup of East Asians, and two an outgroup of Africans. The online version of this article includes the following figure supplement(s) for figure 8:

**Figure supplement 1.** The *E.coli* core tree is relatively insensitive to removal of all SNPs that correspond to branches of the core tree.

robust given enough samples, even if this average does not actually represent any typical sample. For example, instead of the total divergence between a pair of individuals representing the distance to their clonal ancestor, this divergence represents the *average* distance to the many different common ancestors of the pair across the many different phylogenies along the genome, and this average is a function of the rates at which their genetic ancestors have recombined. That is, the distances in this tree reflect the relative rates of recombination among the different lineages.

We propose that the situation for the bacterial data is very similar. Our analysis above has shown that many different phylogenies occur along the alignment and that the core tree must reflect the statistics of this distribution of phylogenies. We propose that, in the same way as for the human

data, the core tree that results from applying PhyML to the core genome of a set of bacterial strains reflects population structure, that is the relative rates with which different lineages have recombined. To support this, we return to the *E. coli* data and use the SNP statistics to quantify the relative rates with which different lineages have recombined.

## Recombination rates across lineages follow approximately scale-free distributions

The analyses above have shown that the core alignment of the *E. coli* strains consists of tens of thousands of short segments with different phylogenies. Thus, one approximate way of thinking about the core genome alignment is that the phylogeny at every genomic locus is independently drawn from some distribution over all possible phylogenies. Conceptually, whereas for purely clonal evolution each strain will have a unique clonal ancestor at each time *t* in the past, due to the frequent recombination each strain will have a large number of different genetic ancestors, that are responsible for different parts of the strain's genome. We will refer to the genetic ancestors of each strain as one moves back in time as the *lineages* of each strain. Population structure corresponds to the fact that different lineages, that is the genetic ancestors of different strains, did not all recombine at the same rate, but that some recombined much more frequently than others.

The distribution of observed SNP types in fact contains extensive information about the relative frequencies with which different lineages have recombined at different times in the past. For example, imagine a SNP where two strains share a nucleotide which differs from the nucleotide that all other strains possess. We will denote such SNPs as 2-SNPs or pair-SNPs. If, at some genomic locus *g*, we find a 2-SNP shared by strains $s_1$ and $s_2$, then it follows that, whatever the phylogeny is at locus *g*, the strains $s_1$ and $s_2$ must be nearest neighbors in the tree, and the SNP corresponds to a substitution that occurred on the branch connecting the ancestor of $s_1$ and $s_2$ to all other strains. As an example, *Figure 9A* graphically shows the frequencies of all pair-SNPs $(A1, s)$ in which A1 shares a SNP with one other strain *s*. Note that, if the data consisted of a clonal phylogeny plus random recombinations, we would expect A1 to have a substantial number of 2-SNPs with its nearest neighbor in the clonal phylogeny, plus a small number of 2-SNPs with a random collection of other strains. Also, if all lineages were freely recombining, we would expect roughly equal frequencies of all possible 2-SNPs $(A1, s)$. However, this is not what we see. A1 shares 2-SNPs with 17 of the 92 strains in our collection at a wide range of frequencies. Instead of a clear 'clonal' neighbor with most SNPs, there are three strains that share 2-SNPs almost equally commonly with A1, that is A2, A11, and D8 with about 200 2-SNPs each. There are some strains with intermediate numbers of 2-SNPs, that is D4, and H6 with about 70 occurences each, 10 strains with 10 or less 2-SNPs each, and four strains with only a single 2-SNP.

We propose that these relative numbers of SNP-sharing directly reflect relative rates of recombination between the genetic ancestors of these strains. Note that since all loci are overwritten many times by recombination, essentially every locus *g* along the alignment has its own unique lineage of genetic ancestors, as determined by the recombination events overlapping this locus, resulting in its own unique phylogeny $\phi_g$. Tracing the genetic ancestors of a pair of strains $s_1$ and $s_2$ into the past, we will say that their lineages 'have recombined' at time *t* in the past, when their most recent common genetic ancestor occurs at time *t* in the past. In order for strains $s_1$ and $s_2$ to share a 2-SNP at locus *g*, they must be nearest neighbors in phylogeny $\phi_g$, that is their lineages must thus have recombined with each other before recombining with the lineages of any of the other strains. In addition, a substitution must have occurred on the lineage from their common ancestor to the next common ancestor in phylogeny $\phi_g$.

Thus, the number of times $n_{(s_1, s_2)}$ we expect to see a 2-SNP of type $(s_1, s_2)$ is proportional to the fraction $f_{(s_1, s_2)}$ of genomic loci *g* for which $s_1$ and $s_2$ have recombined before recombining with any of the other lineages, times the average length $t_{(s_1, s_2)}$ of the branches from the genetic ancestors of $s_1$ and $s_2$ to the next common ancestor in these phylogenies $\phi_g$, that is

$$n_{(s_1, s_2)} \propto t_{(s_1, s_2)} f_{(s_1, s_2)}. \tag{1}$$

The fraction $f_{(s_1, s_2)}$ reflects the rate with which the lineages of $s_1$ and $s_2$ have recombined, relative to the rates with which the lineages of $s_1$ and $s_2$ recombine with the lineages of all other strains. In

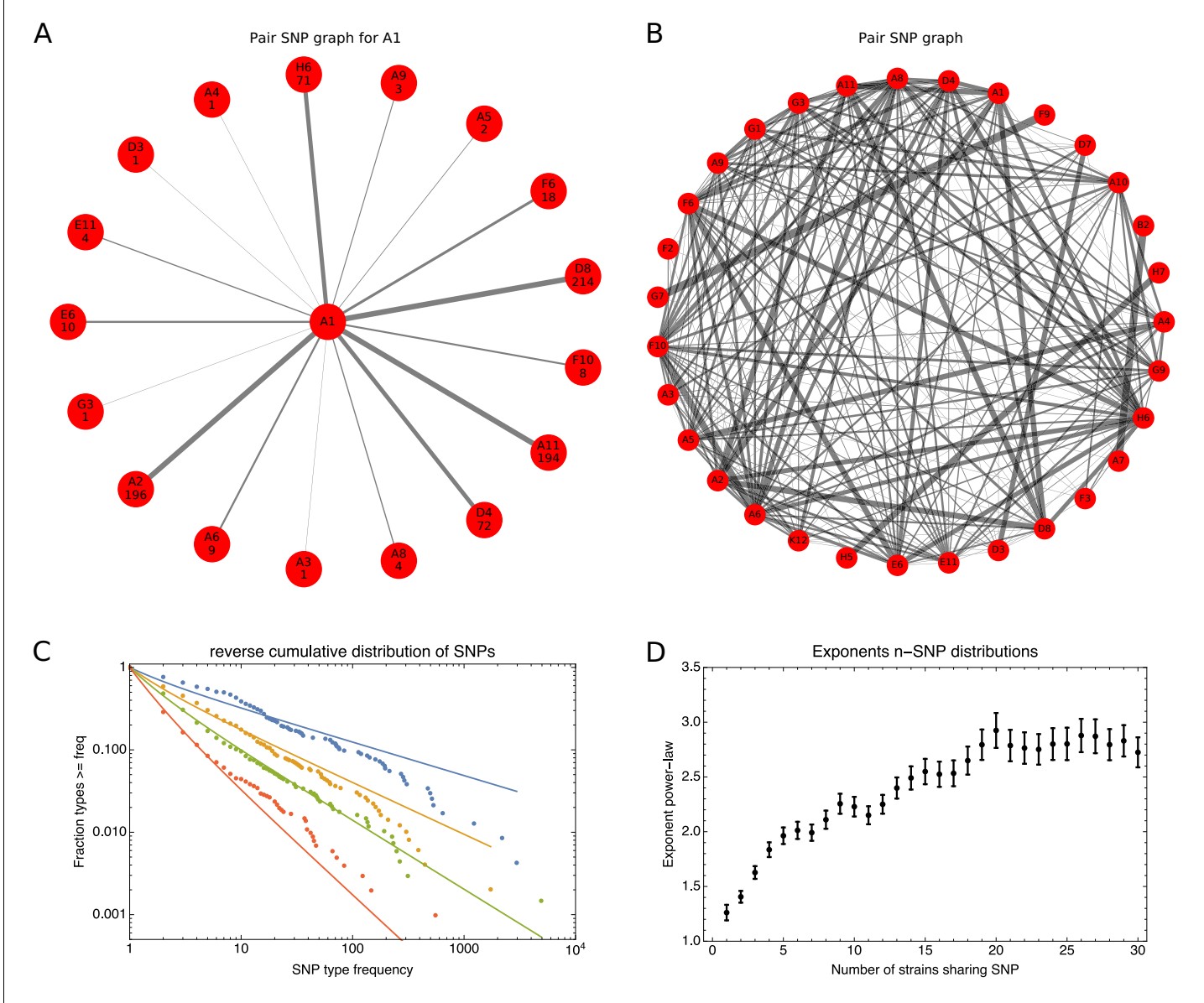

**Figure 9.** SNP-type frequencies follow approximately power-law distributions. (**A**) Frequencies of 2-SNPs of the type $(A1, s)$ in which a SNP is shared between strain A1 and one other strain $s$. Each edge corresponds to a 2-SNP $(A1, s)$ and the thickness of the edge is proportional to the logarithm of the number of occurrences of this 2-SNP. The frequency of each edge is also indicated at the corresponding outer node. (**B**) A graph showing all 2-SNPs $(s, s')$ that were observed in the core genome alignment. Each node corresponds to a strain and each edge to a 2-SNP, with the thickness of the edge proportional to the logarithm of the number of occurrences of the SNP. (**C**) Reverse cumulative distributions of the frequencies of all observed 2-SNPs (blue dots), 3-SNPs (orange dots), 4-SNPs (green dots), and 12-SNPs (red dots). The solid lines in corresponding colors show power-law fits. Both axes are shown on a logarithmic scale. (**D**) Exponents of the power-law fits to the $n$-SNP frequency distributions, as a function of the number of strains sharing a SNP $n$. Error bars correspond to 95% posterior probability intervals.

The online version of this article includes the following figure supplement(s) for figure 9:

**Figure supplement 1.** The $n$-SNP distributions are insensitive to removal of potential homoplasies and removal of $n$-SNPs corresponding to branches of the core tree.

contrast, the average branch length $t_{(s_1, s_2)}$ reflects the *total* rate at which the lineages of the ancestors of $s_1$ and $s_2$ recombine with any of the other lineages. Since the total rates of recombination are determined by sums of many rates, we believe that it is likely that variation in the average branch lengths $t_{(s_1, s_2)}$ across pairs $(s_1, s_2)$ is significantly less than the variation in $f_{(s_1, s_2)}$. That is, we believe the

variation in the number of occurrences $n_{(s_1,s_2)}$ across pairs $(s_1,s_2)$ mostly reflects variation in the fractions $f_{(s_1,s_2)}$, which directly reflect variation in the rates of recombination.

To investigate how the relative rates of recombination vary across all pairs, we calculated the frequencies $n_{(s_1,s_2)}$ of 2-SNPs across all pairs of strains. *Figure 9B* shows a graph representation of all observed pair-SNPs, with the thickness of the edges proportional to the logarithm of the frequency of occurrence of each 2-SNP type. We see that each strain is connected through 2-SNPs to a substantial number of other strains, indicating a high diversity of recent recombination events across the strains. At the same time, the large variability in the thickness of the edges indicates that some pairs occur much more frequently than others. *Figure 9C* shows the reverse cumulative distribution of the frequencies of all observed 2-SNPs (blue dots), that is the distribution of the thickness of the edges in *Figure 9B*. Note that, if the strains were to recombine freely, each 2-SNP would be equally likely to occur, and 2-SNP frequencies would show little variation. Instead, we see that 2-SNP frequencies $f$ vary over more than 3 orders of magnitude, that is from an occurrence of just $f = 1$ for many 2-SNPs to $f = 2965$ occurrences for the most common 2-SNP. Since the reverse cumulative distribution of 2-SNP frequencies follows an approximate straight-line in a log-log plot, the frequency distribution roughly follows a power-law distribution $P(f) \propto f^{-\alpha}$. Fitting the 2-SNP data to a power-law (see Materials and methods) we find that the exponent equals approximately $\alpha \approx 1.41$ (blue line in *Figure 9C*). While clearly not a perfect power-law, the distribution is long tailed and much better fit by a straight line in a log-log plot than by a straight line in either a linear or semi-log plot.

Beyond SNPs shared by pairs of strains, we can of course also look at SNPs shared by triplets, quartets, and so on. Besides the distribution of 2-SNP frequencies, *Figure 9C* also shows the reverse cumulative distributions of 3-SNPs (orange dots), 4-SNPs (green dots), and 12-SNPs (red dots). We see that all these distributions can be approximated with power-law fits (solid lines). We find that essentially all *n*-SNP distributions are approximately scale-free, that is can be fitted with power-laws. Thus, while some sets of *n* strains share SNPs much more often than others, their frequencies fall along a scale-free continuum, so that there is no natural way of dividing the clades of *n* strains into 'common' and 'rare' clades. Note also that each *n*-SNP corresponds to a mutation that occurred on the branch leading to the ancestor of a group of *n* strains. Therefore, *n*-SNPs for larger *n* typically correspond to mutational events that occurred further back in time. As shown in *Figure 9D*, the exponent $\alpha$ of the *n*-SNP distribution increases with *n*, ranging from $\alpha \approx 1.25$ for singlets, that is $n = 1$, to $\alpha \approx 2.8$ for $n \geq 20$. The fact that *n*-SNP distributions become more steep as *n* increases means that the average number of occurrences per *n*-SNP decreases as *n* increases. Thus, the diversity of *n*-SNPs tends to be larger further back in time (see also Appendix 4).

One might wonder to what extent 'clonal' SNPs also contribute to the observed distributions of *n*-SNPs, for example whether the most frequent *n*-SNPs in the tails of the distributions might correspond to clonal SNPs. However, removal of all *n*-SNPs that correspond to branches of the core tree has little effect on the *n*-SNP distributions (*Figure 9—figure supplement 1*). In addition, the *n*-SNP distributions for the 5% homoplasy-corrected alignments also look virtually identical (*Figure 9—figure supplement 1*), showing that homoplasies do not significantly affect the *n*-SNP distributions either.

If, as we have argued, these long-tailed distributions of *n*-SNP frequencies indicate complex population structure, that is that recombination rates of different lineages vary along a wide continuum, then one would expect that the *n*-SNP statistics that are observed for clonally evolving populations, or populations that are freely recombining, look fundamentally different. This is indeed what we observe. Appendix 4 presents an in-depth comparison of *n*-SNP statistics observed for the *E. coli* data, with the *n*-SNPs observed for simulations with different amounts of recombination. To summarize, when recombination rates are so low that populations evolve mostly clonally, that is $\rho/\mu \leq 0.01$, the diversity of *n*-SNP types is low. Although the distribution of *n*-SNP frequencies exhibits long tails, these frequencies mostly reflect the lengths of the branches in the clonal tree, and the exponents are much smaller and approximately independent of *n*. When recombination rates are higher, and more similar to what we inferred for the *E. coli* data, there is a very large diversity of *n*-SNPs, but the *n*-SNP distributions do not exhibit long tails. Instead, all *n*-SNP types have similar frequencies. Thus, the combination of the high diversity of *n*-SNP types with the long-tailed distributions that get more steep as *n* increases, is unique for the *E. coli* data, supporting that these distributions reflect population structure.

Finally, although it is tempting to interpret the $n$-SNP distributions as reflecting an almost 'scale free' population structure, they depend in a complex manner on both the distributions of topologies and branch lengths of the phylogenies across the alignment. As we do not yet have a general theory for the evolutionary process that generates the distribution of phylogenies along the alignment, it is at this point difficult to disentangle the contributions from variations in topologies and branch lengths to the $n$-SNP distributions. Consequently, it is difficult to give a precise interpretation of either the approximate power-law form or the meaning of the exponents.

## Phylogenetic entropy profiles of individual strains

Another way to characterize the structure of the $n$-SNP distributions is to quantify, for each strain $s$, how diverse the clades are that $s$ occurs in at different $n$. To illustrate the idea, we refer back to **Figure 9A**, which shows all the 2-SNPs in which strain A1 occurs. Note that these 2-SNPs are all mutually inconsistent, that is in any one phylogenetic tree the strain A1 can only occur in a pair with *one* other strain. Therefore, the relative frequencies of the different 2-SNPs in which A1 occurs reflect the relative frequencies of phylogenies in which A1 is paired with different strains, and the diversity of these pairs can be quantified by the *entropy* of this distribution. That is, if $n_{(s,A1)}$ is the number of 2-SNPs of type $(s,A1)$, then the fraction of pairs for which A1 occurs with $s$ is $f_s = n_{(A1,s)}/[\sum_{s'} n_{(A1,s')}]$, and the entropy of the distribution of 2-SNPs for A1 is $H_{A1}(2) = -\sum_s f_s \log(f_s)$.

Note that the same calculation can be done for any strain $s$ and any $n$. For example, if the strain $s$ occurs in 10 different quartets of strains $q$, then all quartets $q$ are mutually inconsistent, and the diversity of quartets in which $s$ occurs can be calculated as the entropy $H_4(s) = -\sum_q f_q \log(f_q)$ of the relative frequencies with which the different quartets $q$ occur. In this way, for each strain $s$ we can calculate an entropy profile $H_s(n)$ that contains the entropies of the $n$-SNP distributions in which strain $s$ occurs, as a function of $n$ (see Materials and methods). **Figure 10** shows the entropy profiles of all strains (right panel), as well for a selection of six example strains (left panel), as calculated on the 5% homoplasy-corrected alignments.

Note that, if evolution were strictly clonal, then all entropies $H_n(s)$ would be zero, but instead we see entropies going up to 6–8 bits for all strains. However, probably the most striking feature of the entropy profiles is their great variability across the strains. If all lineages were recombining with each other at equal rates, we would expect the diversity of $n$-SNPs to increase in the same way for each strain, but the data show almost as many distinct entropy profiles as there are strains. This shows directly that the recombination rates across lineages must be highly heterogeneous.

For most strains the entropy increases quickly with $n$, for example for strain A10 the entropy rises to $H_4(A10) \approx 6$ bits, which is equivalent to strain A10 occurring in approximately 64 different quartets with equal frequency. In contrast, for strain E7 the entropy stays zero until $n>6$, and for strain G8 even until $n>20$. As can be seen in the core tree (**Figure 1**), E7 and G8 are part of groups of $n=6$ and $n=20$ very closely related strains, respectively. Both these groups had a clonal ancestor so recently that essentially no recombination events occurred since. Consequently, the strains in these groups occur in at most 1 type of $n$-SNP for $n \leq 6$ and $n \leq 20$, respectively. Thus, for groups of $m$ strains with a very recent common ancestor, the entropies $H_n(s)$ are essentially zero when $n \leq m$. In this way, the entropy profiles directly show when strains are parts of a clonal clade.

Although the entropies generally go up with $n$, they do not increase monotonically. For example, for strain B11, the entropy increases to around $H_n(B11) \approx 4$ for $n=5$ to $n=8$, but then drops back to almost zero at $n=9$. Notably, this strain B11 is a member of the outgroup of nine strains that are highly diverged from all the other strains (**Figure 1**) and the fact that $H_9(B11) = 0$ reflects the fact that 9-SNPs shared by all strains of the outgroup are vastly more numerous than any other 9-SNP in which B11 occurs. That is, whereas B11 occurs in different clades of sizes $n=5$ through 8, the phylogenies of almost all its genomic loci contain the same clade of $n=9$ strains corresponding to the outgroup.

Another feature that is evident is that the entropy profiles of some strains appear to nearly merge at large $n$. For example, the entropy profiles of A10 and E7 become very similar from $n=34$ onward, and from about $n=40$ on-wards the profile of K12 becomes very similar to these two as well. However, not all entropy profiles merge, and different groups of profiles remain up until $n=46$. As can be seen in **Figure 10—figure supplement 1**, we generally observe that strains from the same phylogroup tend to merge their entropy profiles at large $n$. Importantly, however, although the entropy

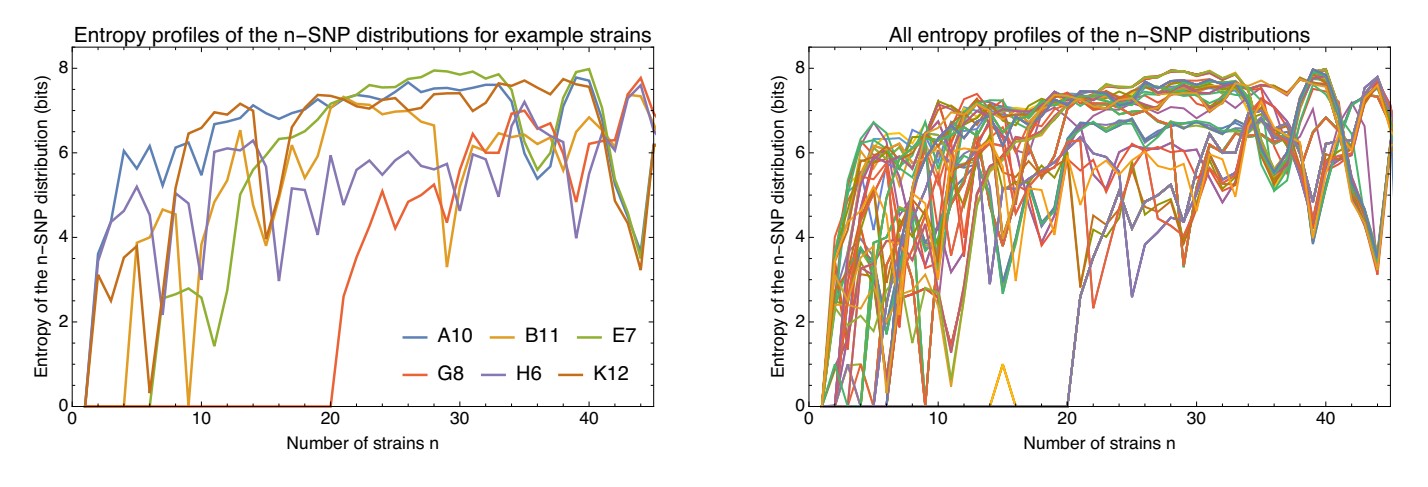

**Figure 10.** Phylogenetic entropy profiles of the *E.coli* strains. Left panel: Entropy profiles $H_s(n)$ (in bits) for six example strains, indicated in the legend. Right panel: Entropy profiles $H_s(n)$ for all *E. coli* strains.
The online version of this article includes the following figure supplement(s) for figure 10:

**Figure supplement 1.** Entropy profiles of the *n*-SNP distributions for each of the *E.coli* phylogroups.

**Figure supplement 2.** Statistical significance of the difference in *n*-SNP statistics for all pairs of strains.

**Figure supplement 3.** Comparison of the entropy profiles of the *n*-SNP distributions of *E.coli* with those of the data of the simulations.

profiles become highly similar at large *n*, they do not become identical. In contrast, for strains with a recent clonal ancestor, the entropy profiles are perfectly identical. For example, the entropy profiles of the close pairs (B6, C1) from phylogroup F, and (A7, B2) from phylogroup B2, completely overlap so that only one of the two colors of each pair is visible in the plot (***Figure 10—figure supplement 1***). We developed a simple statistical test, based on the Fisher-exact test (Materials and methods) to quantify the extent to which the *n*-SNP statistics of a pair of strains are significantly distinct. As shown in ***Figure 10—figure supplement 2***, we find that only very closely related pairs of strains that had a recent common ancestor, corresponding to about 5% of all pairs, have statistically indistinguishable *n*-SNP statistics. Thus, while strains with a recent clonal ancestor have identical entropy profiles, the entropy profiles of strains from a phylogroup have similar but not identical *n*-SNP distributions at large *n*. Moreover, strains from different phylogroups can have very different entropy profiles even at large *n*. To provide additional insights into what the *n*-SNP statistics show about recombination patterns, Appendix 5 provides an in-depth analysis of the *n*-SNP statistics for the relatively small phylogroup B2.

Thus, the general picture that emerges from the entropy profiles is that, while some of the structure at small *n* is due to clonal relationships, at large *n* the entropy profiles reflect the statistics of recombination of different lineages. While strains from the same phylogroup tend to show highly similar *n*-SNP statistics at high *n*, suggesting that their lineages have similar recombination statistics sufficiently far into the past, strains from different lineages have clearly different recombination statistics even far into the past.

Finally, the entropy profiles observed for the *E. coli* strains differ fundamentally from what is observed for the data from simulations (***Figure 10—figure supplement 3***). For the simulations without recombination, the entropies are almost all zero and at the very low recombination rate of $\rho/\mu = 0.001$, where clonal SNPs dominate along all branches of the clonal tree, all entropies are small. In contrast, in the regime where recombination dominates along all branches of the clonal tree, that is for $\rho/\mu \geq 0.3$, the entropy profiles of all strains are highly similar. This confirms that when lineages recombine at equal rates, highly similar entropy profiles result. Therefore, the fact that the entropy profiles are highly heterogeneous for *E. coli*, and that virtually every strain has a distinct entropy profile, directly shows that the lineages of almost every strain have a distinct pattern of recombination rates with the lineages of the other strains.

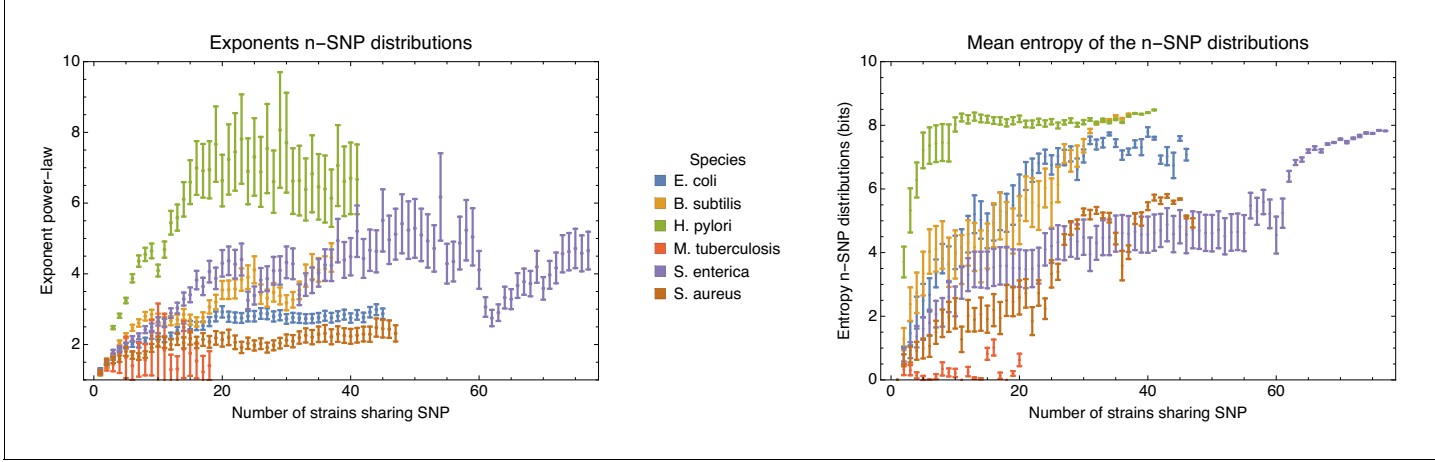

**Figure 11.** Left panel: Exponents of the power-law fits to the *n*-SNP frequency distributions, as a function of the number of strains sharing a SNP *n* for each of the species (different colors). Error bars correspond to 95% posterior probability intervals. Right panel: Mean entropy of the entropy profiles $H_n(s)$, averaged over all strains *s*, as a function of the number *n* of strains sharing the SNP, for each of the species (different colors). The error bars correspond to two standard-errors of the mean.

The online version of this article includes the following figure supplement(s) for figure 11:

**Figure supplement 1.** Power-law fits of the *n*-SNP distributions for all six species.

**Figure supplement 2.** Entropy profiles of all the strains for each of the six species.

**Figure supplement 3.** *n*-SNPs distributions and entropy profiles for the human data.

## *n*-SNP statistics and entropy profiles for the other bacterial species

We next investigated whether the *n*-SNPs of the other species also exhibit approximately power-law distributions, as observed in *E. coli*. *Figure 11—figure supplement 1* shows the reverse cumulative distributions of 2-SNPs, 3-SNPs, 4-SNPs, and 12-SNPs across all six species together with power-law fits. Although the curves often deviate substantially from simple straight lines, they all exhibit long tails and range over several orders of magnitudes, that is up to five orders of magnitude for 2-SNPs in *S. enterica*. Note that *M. tuberculosis* is again an exception. The total number of different *n*-SNP types is so small for this species that the *n*-SNP distributions are not well defined, suggesting again that this species is mostly if not completely clonal, and we will not further comment on its *n*-SNP statistics.

*Figure 11* (left panel) shows the fitted exponents of the power-law distributions of *n*-SNPs as a function of *n* for all species. We see that the exponents generally increase with *n* indicating that the phylogenetic diversity generally increases as one moves further back in time, that is to larger *n*. Consistent with other observations, *H. pylori* shows the highest exponents, that is the highest diversity. It is also interesting that, while the exponents become roughly constant when $n>20$ for *E. coli*, *H. pylori* and *S. aureus*, *B. subtilis* and *S. enterica*, exhibit more complex patterns with sudden drops in exponent at particular values of *n*, suggesting more complex population structures at large *n* for these species.

*Figure 11—figure supplement 2* shows the entropy profiles $H_n(s)$ for all strains *s* in each of the species. Consistent with all other statistics, all entropies are low for the *M. tuberculosis* strains, further supporting that this species is predominantly clonal. With the exception of *H. pylori*, where the entropy profiles are quite similar for most strains, consistent with an almost 'freely recombining' population (*Suerbaum et al., 1998*), all other species exhibit highly diverse entropy profiles across strains, showing that also in these other species almost every strain has a unique 'fingerprint' of frequencies with which its lineages recombine with the lineages of other strains. That is, the entropy profile analysis suggests that, in most species, population structure is complex, with highly heterogeneous recombination rates across the lineages of different strains.

Although the entropy rises quickly to values in the range $4-8$ for most strains, we also see strains for which the entropy only rises after *n* exceeds some fairly large value of *n*, for example at $n=10$ for some strains in *H. pylori*, and at $n=24$ and $n=62$ for some *S. enterica* strains. This suggests that

the corresponding strains are part of clonal groups of very closely related strains. To summarize the entropy profiles of each species, the right panel of *Figure 11* shows the mean and standard-error of the entropy profiles, averaged over all strains, as a function of *n*. As for most other statistics, *M. tuberculosis* is an outlier whose strains generally only show low phylogenetic entropy. For all other species, the average entropy clearly increases as *n* increases, indicating again that the phylogenetic diversity increases further back in the past. For four of the six species, the mean entropy at large *n* falls in a narrow range between 7 and 8 bits, suggesting that the effective number of different ancestries far back in the past is relatively similar for these species.

Finally, for comparison we decided to also calculate these patterns on the human data. In particular, we extracted SNP data for chromosomes $1-12$ for 2504 humans from the 1000 Genomes project (*1000 Genomes Project Consortium et al., 2015*). *Figure 11—figure supplement 3* shows examples of the *n*-SNP distributions for human together with the fitted exponents for *n* ranging from 1 to 30, as well as the entropy profiles for the 40 randomly chosen individuals used previously in *Figure 8*. Apart from the 2-SNPs, which appear to show a biphasic pattern, all other *n*-SNP distributions in human are all well fit by power-law distributions. Moreover, for $n \geq 5$ the exponents are almost constant around a value of 2.8. Interestingly, the entropy profiles of all 40 individuals quickly rise to around 10 bits. However, from that point onward the entropy profiles of individuals with African ancestry deviate from those of the other individuals, showing consistently higher entropy, confirming the well-known fact that African populations have higher genetic diversity. However, in contrast to what we observe for most bacterial species, the entropy profiles for individuals with the same geographic ancestry are all very similar. Among the bacterial species we studied, only *H. pylori* shows similar entropy profiles. In this respect it is noteworthy that previous studies have argued that *H. pylori* is almost freely recombining (*Suerbaum et al., 1998*), and that the population structure that it exhibits in fact reflects the population structure of its human host (*Falush et al., 2003*).

## Discussion

Despite the fact that complete bacterial genomes have been available for more than 20 years, we still lack a clear picture of the relative contributions of recombination and clonal evolution in shaping the observed sequence variation among strains of a bacterial species. In our view, a key problem is that previous methods for quantifying the role of recombination virtually all make specific simplifying assumptions about the underlying evolutionary dynamics. However, we do not know to what extent these assumptions hold and affect the conclusions. To address this problem, we have here developed new methods that avoid making specific assumptions about the evolutionary dynamics as much as possible. In particular, showing that almost all bi-allelic SNPs in the core genome alignment correspond to single mutations in the history of the coresponding positions, we showed several new ways in which these SNPs can be used to quantify phylogenetic structures and the role of recombination in genome evolution.

Our analysis shows that, for all but one of the species studied here, evolution of the core genome is almost entirely driven by recombination. Recombination is so common that for most pairs of strains none of the DNA in their pairwise alignment derives from their clonal ancestor. Although pairs of strains exist that are so closely related that most of their DNA was clonally inherited, even for such pairs the large majority of substitutions that separate them derive from recombination events. When considering the core genome alignment of a collection of roughly 100 strains, we found that each genomic locus has been overwritten many times by recombination in the history of the strains, that is hundreds if not more than a thousand times for *E. coli*. Consequently, few if any of the loci follow the clonal phylogeny, and we found that the core genome alignment fragments into many thousands of short segments with different phylogenies. Thus, approximating sequence evolution in the core genome as occurring along the branches of a fixed phylogenetic tree is clearly inappropriate.

One might infer from these results that the dynamics of genome evolution within a bacterial species could be better approximated as quasi-sexual with free recombination between lineages. However, as has been noted previously (*Yang et al., 2019*), such an approximation is also inconsistent with the data. Strains do not appear roughly equidistant and phylogenies build from large numbers of genomic loci clearly converge to a well-defined phylogeny. We here argued that this clear phylogenetic structure reflects population structure, that is variation in the relative rates with which

different lineages recombine. As a consequence of this population structure, the many different phylogenies that occur along the genome alignment are drawn from a highly non-uniform distribution, and the core genome phylogeny represents an effective average of this distribution of phylogenies.

These conclusions are clearly at odds with the prevailing view that, although *E. coli* is subject to substantial amounts of recombination, an overall phylogenetic backbone representing clonal relationships can still be identified (*Touchon et al., 2009*; *Didelot et al., 2012*; *Bobay et al., 2015*; *Denamur et al., 2021*), either by constructing a maximum-likelihood phylogeny from whole genome alignments (*Touchon et al., 2009*; *Didelot and Wilson, 2015*), or using more sophisticated approaches based on detecting loci that have not been affected by recombination (*Didelot et al., 2012*; *Croucher et al., 2015*), and inferring the clonal tree from those. If, as we are arguing, this prevailing view is mistaken, this raises the question why these previous approaches reached misleading conclusions. The answer is that these approaches rely on assumptions that do not apply. First, methods that aim to reconstruct the clonal phylogeny from a subset of positions that have not been affected by recombination are inherently problematic because our analysis shows that, for most species, such positions simply do not exist. An arguably more fundamental problem with previous analyses is that they assume recombination is completely random, that is without population structure. For example, the sophisticated ClonalFrame method (*Didelot and Falush, 2007*), which has been used to reconstruct a clonal phylogeny of *E. coli* strains (*Didelot et al., 2012*), uses a model in which along each branch of the clonal phylogeny a fraction of the loci is overwritten by recombination, while the rest is inherited clonally. However, instead of explicitly modeling recombination between lineages, it models the effects of recombination as simply introducing random substitutions at a fixed rate into the recombined segments. Consequently, recombination cannot introduce any phylogenetic structure by construction, and any phylogenetic structure evident in the data will thus be interpreted as reflecting clonal relationships. In short, a key problem with previous methods is that, by either explicitly or implicitly relying on models that assume that there is no population structure, the phylogenetic patterns resulting from population structure have been mistaken for clonal relationships. Note that this also implies that methods that aim to detect or quantify recombination by deviation from clonal phylogenies inferred by such methods are also fundamentally flawed.

Many studies have aimed to gain insight into bacterial genome evolution by simulating simple models of evolutionary dynamics, and comparing statistics in the resulting data with analogous statistics in real data. However, when these simple models make assumptions that do not hold in the real world, one may easily reach misleading conclusions. For example, a recent study used simulations of a simple genome evolution model with completely random recombination to conclude that, even which recombination is very frequent, phylogenies reconstructed from whole genome alignments can still correctly identify clonal relationships (*Stott and Bobay, 2020*). This is not surprising because, as we just noted, when recombination is random any phylogenetic structure in the data can only reflect clonal relationships by construction. However, in the real world, recombination is not completely random, that is there is population structure.

To further support that phylogenetic structures in whole genome alignments reflect population structure, we also developed methods for quantifying this population structure in more detail. In particular, we showed that bi-allelic SNPs contain much information about the statistics of relative recombination rates across lineages. Whenever a particular $n$-SNP occurs in the alignment, the $n$ strains sharing the SNP must occur as a clade in the phylogeny at that locus. Consequently, the relative frequencies of different $n$-SNP types reflect the relative frequencies with which different subsets of strains occur as clades across the phylogenies. We find that the frequencies of $n$-SNP types vary over 3–5 orders of magnitude and follow roughly power-law distributions. Notably, since the $n$-SNP distributions follow smooth long-tailed distributions that do not appear to have a characteristic scale, it is not possible to naturally subdivide subsets into highly and rarely recombining types. Rather, there seems to be a large continuum of relative rates, with population structure on all scales. Given that recombination rates vary over orders of magnitude across different lineages, the idea of an effective single recombination rate for a bacterial species might be misleading, and it thus seems problematic to fit the data to models that assume a constant rate of recombination within a species, for example (*Vos and Didelot, 2009*; *Lin and Kussell, 2019*).

Essentially, all population genetics and coalescent models start from assuming one or more populations of individuals that, for the purpose of the model, are exchangeable. However, our analysis of the entropy profiles and $n$-SNP statistics showed that, apart from some groups of extremely

closely related strains that share a common ancestor before any recombination events occurred, almost every strain has its own distinct $n$-SNP statistics. In particular, virtually every strain $s$ has a unique entropy profile $H_n(s)$ of the distributions of frequencies with which it occurs in different clades of size $n$. This suggests that almost every strain has a unique pattern of relative recombination rates between its lineages and the lineages of the other strains. Consequently, models that start from populations of exchangeable individuals may be inappropriate by definition.

Given that models that assume either a single consensus tree, a fixed rate of recombination across strains, or exchangeable individuals, are all clearly at odds with the data on prokaryotic genome evolution, this raises the question of what would be an appropriate mathematical 'null model' that can capture the statistics that we observed here. In such a model, almost every lineage must have distinct rates of recombination with other lineages, these rates must vary over multiple orders of magnitude, and the model should reproduce the roughly power-law distributions of $n$-SNP frequencies, ideally with exponents that can be tuned by parameters in the model. It is currently unclear how to construct such a model.

Apart from the question of how to mathematically model the observed patterns, a second key question is what determines the highly variable relative rates of recombination across lineages. Just natural selection acts on mutations and thereby shapes the observed patterns of substitutions, selection also undoubtedly strongly acts on recombination events and shapes the recombination patterns that we detect in the core genome alignments. Indeed, strong selection on horizontal transfer events between two *B. subtilis* strains was recently observed in a laboratory evolution experiment (*Power et al., 2020*). However, it is currently not clear to what extent the differences in observed recombination rates are shaped mainly by natural selection, for example that due to epistatic interactions only recombinant segments from other strains with similar 'ecotypes' are not removed by purifying selection, as suggested by some previous works (*Shapiro et al., 2012*; *Cadillo-Quiroz et al., 2012*), or that recombination rates may be set mostly by which lineages co-occur at the same geographical location. It is also conceivable that phages are a major source of transfer of DNA between strains, so that recombination rates may reflect the rates at which different lineages are infected by the same types of phages.

It is well-known that homologous recombination requires sufficient homology between the endpoints of the DNA fragment and the homologous segment in the host genome. Thus, recombination rates will intrinsically decrease with the nucleotide divergence between strains. An intriguing and theoretically attractive proposal is that relative recombination rates are simply set by sequence divergence and that bacterial species may essentially be defined by the collection of strains that are sufficiently close to allow efficient recombination (*Dixit et al., 2017*). Previous studies have estimated that the rate of successful recombination decreases exponentially with nucleotide divergence (*Vulić et al., 1997*; *Oliveira et al., 2016*), and in *Vulić et al., 1997* it was estimated that every 1% of sequence divergence leads to a roughly two-fold reduction in recombination efficiency. However, since most *E. coli* strains are within 1–3% sequence divergence, this would suggest less than 10-fold variability in the relative recombination rates across our strains, whereas the $n$-SNP statistics suggest a much larger range of relative recombination rates.

Finally, while we here studied the frequency distribution of $n$-SNP types as well as the entropies $H_n(s)$ of the $n$-SNP distributions for each strain, it appears to us that this is just the tip of the iceberg of possible ways in which $n$-SNPs can be used to study the evolution of a set of strains from their core genome alignment. Our analyses indicate that prokaryotic genome evolution is driven by recombination that occurs at a very wide distribution of different rates between different lineages, and there is now a strong need for identifying what sets these rates, and the development of new mathematical tools and models that can accurately describe this kind of genome evolution.

## Materials and methods

### Data

The *E. coli* sequences analyzed here can be accessed on NCBI Bioproject via the accession number PRJNA432505 (*Ishii et al., 2007*; Field, 2020, in preparation). These genomes were sequenced with Illumina HiSeq 2000 technology. Samples were multiplexed, 24 per lane, producing 100 bp paired

end reads which resulted in an average coverage depth of between 125x and 487x (with four strains over-represented at more than 1000x).

Genome sequences for all other species were downloaded from http://ftp.ncbi.nlm.nih.gov/genomes/refseq/bacteria. The following two source data files contain accessing numbers for all the genomes used in this study.

*Figure 1—source data 1*. Table with source data for *Figure 1—figure supplement 1*. *Figure 7—source data 1*. Table with source data for *Figure 7*.

In order to facilitate reproduction of the results presented here, we also provide a comprehensive collection of data files containing not only all raw genome sequences and the core genome alignments for each of the species, but also processed data files containing the results of the pairwise analysis and SNP statistics. These data files are available as object 10.5281/zenodo.4420880 from https://zenodo.org/record/4420880.

## Core genome alignment and core tree

To build a core genome alignment for the SC1 strains, we used the Realphy tool (*Bertels et al., 2014*) with default parameters and Bowtie 2 (*Langmead and Salzberg, 2012*) for the alignments. The Illumina sequencing used to sequence the *E. coli* strains can still make sequencing errors at a certain rate and obtaining accurate base calls genome-wide relies on having sufficient read coverage at each position to confidently call the consensus nucleotide. We note that, in order to avoid sequencing errors, Realphy is quite conservative in its construction of the core genome alignment. In particular, in order for a position to be included each strain has to be represented, the coverage has to be at least 10 in each strain, and at least 95% of the reads from each strain have to agree on the nucleotide. A rough bound on the rate of sequencing errors can be obtained by noting that some of our strains are extremely closely related, that is while all 92 strains are unique in their full genome, there are only 82 unique core genomes. This means that there are 10 genomes that are identical in their core to another genome. Given that the length of the core alignment is $L = 2'756'541$, this means there were $10 \times L$ nucleotides sequenced without a sequencing error and a simple Bayesian calculation shows that this implies that, with 99% posterior probability, the rate of sequencing errors is less than $\mu_s = 1.1 \times 10^{-7}$.

Realphy used PhyML (*Guindon et al., 2010*) with parameters -m GTR -b 0 to infer trees from the whole and parts of the core alignment. The tree visualizations were made using the Figtree software (*Rambaut, 2018*).

## Analysis of core alignment blocks

For each 3 kb block of the core alignment, we used PhyML using the option -c one to infer a phylogeny while restricting the number of relative substitution rate categories to one. Furthermore, to calculate the log-likelihood of a given 3 kb block under the tree topologies of other blocks, we reran PhyML using the -o 'lr' option, which only optimizes the branch lengths as well as the substitution rate parameters but does not alter the topology of the phylogeny.

## Pairwise analysis and mixture modeling

For each pair of strains we slide a 1 kb window over the core genome alignment of the pair, shifting by 100 bp at a time, and build a histogram of the number of SNPs per kilobase by counting the number of SNPs in each window. That is, we obtain the distribution $P_n$ of the fraction of 1 kb windows that have $n$ SNPs. We then assumed that the one kilobase blocks can be separated into a fraction $f_a$ of 'ancestral blocks', that is regions that were inherited from the clonal ancestor of the pair, and a fraction $(1 - f_a)$ that have been recombined since the pair diverged from a common ancestor. Although in previous work a simple ad hoc scheme was used in which it was assumed that blocks with less than a particular number of SNPs are ancestral and blocks with more SNPs are recombined (*Dixit et al., 2015*), we found that this approach is not satisfactory and results significantly depend on the cut-off chosen.

We thus decided to employ a more principled mixture model approach. For the ancestral regions, the number of SNPs per kilobase should follow a simple Poisson distribution $P_n = \mu^n e^{-\mu}/n!$, with $\mu$ the expected number of mutations per block. For the recombined regions, we note that these regions themselves will consist of mosaics of sub-regions that have been recombined previously.

Consequently, the recombined regions will consist of a mixture of Poisson distributions with different rates. It is well-known that mixtures of Poisson distributions with rates that are (close to) Gamma-distributed follow a negative binomial distribution and we found empirically that negative binomial distributions give excellent fits to the observed SNP distributions in our data. For the recombined regions, we thus assume a negative binomial distribution of the form

$$P_n = \frac{\Gamma(n+\alpha)}{\Gamma(\alpha)n!}\lambda^n(1-\lambda)^\alpha,$$ (2)

where $0<\lambda<1$ and $\alpha \geq 1$ are parameters of the distribution. We thus fit the observed distribution of SNPs per block $P_n$ using the following mixture:

$$P_n = f_a\frac{\mu^n}{n!}e^{-\mu} + (1-f_a)\frac{\Gamma(n+\alpha)}{\Gamma(\alpha)n!}\lambda^n(1-\lambda)^\alpha,$$ (3)

where $f_a$ is the fraction of the genome that is inherited from the clonal ancestor. Fits were obtained using maximum likelihood. While expectation maximization was used to optimize the parameters $f_a$, $\mu$, and $\lambda$, a grid search was employed to find the optimal dispersion parameter $\alpha$.

Note that, in terms of the fitted parameters, the total number of mutations in ancestral blocks is $\mu f_a$, and the number of mutations in recombined blocks is $(1-f_a)\alpha\lambda/(1-\lambda)$.

To estimate the lengths of recombination events, we first extracted pairs that are sufficiently close (divergence less than 0.002) such that multiple overlapping recombination events are unlikely. We then used a two-state HMM with the same two components, that is a Poisson and a negative binomial component corresponding to ancestral and recombined segments, and having fixed transition rates from the ancestral to the recombined state and vice versa, to parse the pairwise alignment into ancestral and recombined segments. Note that the parameters of the HMM are fitted separately for each pair of strains. For each pair, we took as recombined segments those contiguous stretches that were assigned to the recombined state by the HMM.

We define mostly clonal pairs as pairs with more than 90% of the alignment classified as ancestral, fully recombined pairs as pairs with less than 10% of the alignment classified as ancestral, and all other pairs as transition pairs. In order to estimate the critical divergence at which half of the genome is recombined we fit a linear model to the observed relationship between divergence and clonal fraction in all transition pairs, and define the critical divergence as the divergence at which the linear fit has a clonal fraction of 50%. To calculate the fraction of mutations that derive from recombined segments at the critical divergence, we compute the fraction of mutations in recombined segments for all transition pairs (using the results from the mixture model) and fit a linear model to the observed dependence between the ancestral fraction an the fraction of mutations in recombined segments. We then define the fraction of mutations in recombined regions at the critical divergence as the fraction of mutations in the linear fit when the ancestral fraction is 50%.

## Simulated data sets

In order to simulate genome evolution under a simple evolutionary model that includes drift, mutation, and recombination, we developed new software (manuscript in preparation) based on general purpose GPU programming (GPGPU), which is available from https://github.com/thomsak/GPUprokEvolSim (*Sakoparnig, 2021*; copy archived at swh:1:rev:ed15c4b013068d684874c3d7c9-d710e289e3f31c). Our software explicitly simulates the evolution of a population of $N$ DNA sequences of length $L_g$ using a Wright-Fisher model with non-overlapping generations. In order to be able to not only evolve the sequences, but also track recombination events occurring along the clonal history of a sample of $S$ genomes from the population of $N$, we proceed as follows.

The evolution of the $N$ genomes is simulated for $8N$ generations. First, a 'guide' clonal history for the sample of $S$ sequences is determined using the Kingman coalescent (*Wakeley, 2008*) for $S$ individuals within the population of $N$. In particular, for the subset of $S$ genomes, their ancestry is determined for $8N$ generations into the past. Notably, because the $S$ sequences will have a common ancestor much more recently than $8N$ generations in the past, for many of these generations there will only be a single ancestor.

We then simulate $8N$ generations of the evolution of $N$ genomes of length $L_g$ forward in time by iterating:

1.  For each individual corresponding to an ancestor from the clonal phylogeny of the sample of *S* genomes, its parent in the previous generation is chosen according to the guide clonal phylogeny. For each other individual, a random parent is chosen from the previous population of *N* individuals.

2.  For each individual of the new generation, the genome of the ancestor is copied, scanned from left to right, and at each position a recombination event is initiated with probability ρ. If a recombination is chosen to occur at position *i*, one of the $N - 1$ other members of the population is chosen at random, and the section of its genome from position $i + 1$ to $i + L_r$ is copied into the current genome. After this, each position in the genome is mutated with probability μ. The target nucleotide is chosen randomly using a transition-transversion ratio of 3-to-1.

Apart from tracking the *N* genome sequences, we also track, for each branch of the clonal phylogeny of the *S* individuals, how many times each position was overwritten by recombination during the evolution along the branch. This allows us to calculate what fraction of positions are inherited clonally along each branch, how many times each position in the final alignment of *S* genomes was overwritten by recombination during its clonal history, and what fraction of positions was clonally inherited for each pair of strains.

## Parameter settings

To keep the simulations computationally feasible, we simulated a population of $N = 3600$ individuals. To make the simulations directly comparable with the data from *E. coli*, we focused on samples of $S = 50$ individuals, used a genome size of $L_g = 2'560'000$ bp, and used a size of $L_r = 12'000$ bp for the recombined fragments. Note that this length of recombined fragments is toward the lower range of the sizes of recombined fragments estimated from close pairs of *E. coli* strains. For a Kingman coalescent, the expected fraction of columns that is not polymorphic in an alignment of *S* individuals is given by

$$\prod_{k=1}^{S} \frac{k}{k + 2N\mu},\tag{4}$$

and we set $\mu$ such that this fraction is 0.9, that is similar to what is observed for the alignment of *E. coli* strains. In particular we set $\mu = 3.28 \times 10^{-6}$ so that $N\mu = 0.012$.

Finally, apart from simulations without recombination, that is $\rho = 0$, we performed simulations with six different recombination rates, corresponding to ratios of recombination to mutation of $\rho/\mu = 0.001$, $\rho/\mu = 0.01$, $\rho/\mu = 0.1$, $\rho/\mu = 0.3$, $\rho/\mu = 1$, and $\rho/\mu = 10$, that is covering four orders of magnitude. Note that the clonal phylogeny for the subset of *S* strains was created once, and then used in all simulations with different recombination rates to facilitate direct comparisons.

## Estimating the fraction of SNPs that correspond to single substitution events

Our analyses assume that bi-allelic SNPs correspond to single substitution events in the evolutionary history of the genomic position. This assumption may break down due to apparent SNPs caused by sequencing errors, as well as due to homoplasies, that is bi-allelic columns for which more than one substitution event occurred. We here estimate the frequency of both types of events.

### SNPs due to sequencing errors are negligible

Above we estimated, from the fact that we observe 10 strains that are identical in their core genome to another strain, that the rate of sequencing errors is at most $\mu_s = 1.1 \times 10^{-7}$. Using this, we expect a sequencing error to occur in at most $L\left(1 - (1 - \mu_s)^{92}\right) = 28$ of the columns of our length $L = 2'756'541$ core genome alignment. Note that, because the fraction of columns with more than one sequencing error is negligible, all these sequencing errors will produce a mutant nucleotide in only one strain, because the expected number of columns with two sequencing errors is negligible. Thus, if a sequencing error occurs in a column that is otherwise non-polymorphic, it will create a bi-allelic SNP in which only one strain carries a differing base and such SNPs do not affect any of the phylogenetic analyses. Thus, sequencing errors affecting the phylogenetic analysis only occur when

the sequencing error occurs in a SNP column, and creates one of the two nucleotides that already existed. Since only about 10% of columns are polymorphic, the expected number of SNP columns that are informative for phylogeny and affected by sequencing errors is at most $28/10 = 2.8$. Given that there are $247'822$ phylogeny informative SNPs in total, the expected fraction of these that are affected by sequencing errors is roughly $10^{-5}$.

## Estimating the rate of homoplasy

The relatively low frequency of SNPs and the fact that almost all SNPs are bi-allelic strongly suggests that almost all bi-allelic SNPs correspond to single substitution events in the evolutionary history of the position. Here, we use a simple substitution model to estimate the fraction of bi-allelic SNPs that correspond to multiple substitution events, that is homoplasies. To do this, we will analyze the observed frequencies of columns with 1, 2, 3, and 4 different nucleotides under a simple substitution model. Note that, in this simple model we assume that all substitutions are neutral, so that there is essentially no difference between mutations and substitutions. In the real data some mutations are deleterious and some of these are removed by purifying selection, leading to lower rates of substitutions at some positions than at others. Indeed, for our core genome alignment of *E. coli* strains, we observed that SNPs occur almost 10 times more frequently at synonymous sites than at second positions in codons. To assess the effects of selection, we will fit our model to the frequencies of columns with 1, 2, 3, and 4 different nucleotides both for the subset of positions that should be under relatively little selection, that is third positions in fourfold degenerate codons, and positions that should be under relatively strong selection, that is second positions in codons. Since a large fraction of all SNPs corresponds to SNPs in which the outgroup of nine strains has a nucleotide that differs from the nucleotide of all other strains, we will also fit the model separately for all strains, and all strains minus the nine strains of the outgroup.

We will consider the following simple model for the substitutions in a single alignment column. Note that, because the phylogeny may change across positions in the alignment, the sum of the length of the branches of the phylogeny at a given position will vary from position to position. Let $\mu$ denote the product of the substitution rate times the total length of the branches in the phylogeny at the position of interest. The variable $\mu$ thus corresponds to the expected number of substitutions in the evolutionary history of this position. The probability that $n$ substitutions took place at this position is then given by a Poisson distribution:

$$P_n = \frac{\mu^n}{n!} e^{-\mu}, \tag{5}$$

and each of these substitutions is equally likely to occur anywhere on the branches of the phylogeny.

We want to calculate the probability $P(d|n)$, that if $n$ substitutions occurred along the phylogeny, that $d$ different nucleotides will occur at the leaves. Clearly, if no substitutions occurred, there will be only one nucleotide, so that $P(d|0) = \delta_{d1}$, with $\delta_{ij}$ the Kronecker delta function. Also, if one substitution occurred, then we know there must be two nucleotides occuring at the leaves, that is one nucleotide in all leaves downstream of the branch in which the substitution occurred, and one other nucleotide at all other leaves, that is $P(d|1) = \delta_{d2}$. However, when two or more substitutions occur, the situation is more complicated. Let $\alpha$ denote the ancestral nucleotide and assume that the first substitution mutated $\alpha$ to $\beta$. The second substitution either occurs in one of the branches carrying the ancestral nucleotide $\alpha$, or in one of the branches carrying $\beta$. To remain at $d = 2$ nucleotides, one has to either have another substitution $\alpha \to \beta$, or a substitution $\beta \to \alpha$. In all other cases the number of nucleotides will increase to 3.

Under this simple model, we can calculate the general probabilities $T(d'|d)$ that, if there are currently $d$ different nucleotides and another substitution is added, that there will be $d'$ different nucleotides after the substitution. These probabilities depend on the relative probability for transitions and transversions to occur and we will assume that transitions are $r$ times as likely to occur as transversions. Besides having $T(2|1) = 1$, we find that $T(2|2) = (2 + r^2)/(2 + r)^2$, $T(3|2) = (2 + 4r)/(2 + r)^2$, $T(3|3) = 2/3$, $T(4|3) = 1/3$, and $T(4|4) = 1$. All other transition probabilities are zero.

Starting from a single nucleotide in the column, the probability $P(d|n)$ to end up with $d$ different nucleotides after $n$ mutations is given by the $n$-th power of the transition matrix $T$, that is

$P(d|n) = T^n(d|1)$. From this, we can work out the probability $P(d|\mu)$ to end up with $d$ different nucleotides as a function of the expected number of mutations μ as

$$P(d|\mu) = \sum_{n=0}^{\infty} T^n(d|1) \frac{\mu^n}{n!} e^{-\mu}, \tag{6}$$

which can be easily numerically evaluated to sufficient precision.

Assume we observe $c_d$ columns with $d$ different nucleotides, with $d$ running from 1 to 4. The log-likelihood of this count data given $\mu$ is

$$L(\mu) = \sum_{d=1}^{4} c_d \log[P(d|\mu)], \tag{7}$$

and by maximizing this log-likelihood with respect to $\mu$ (numerically), we obtain an estimate $\mu_*$ given the counts $c_d$. Finally, given $\mu_*$, the fraction $f_h$ of homoplasies, that is bi-allelic SNPs that correspond to multiple substitutions, is given by the fraction of all columns for which $d = 2$ but $n > 1$. This fraction is given by

$$f_h = 1 - \frac{\mu_* e^{-\mu_*}}{P(2|\mu_*)}. \tag{8}$$

*Table 1* shows the estimated expected number of mutations per column $\mu_*$ and the estimated fraction of homoplasies $f_h$ for the five different subsets of columns, using a transition-to-transversion ratio of $r = 3$. We see that the fraction $f_h$ is at most 6.3% and less than 1% for second positions in codons.

In addition, *Figure 3—figure supplement 1* shows a comparison of the observed and predicted frequencies of columns with 1, 2, 3, and 4 letters. Since effects of selection are likely least for the synonymous positions, we expect the simple model to fit the data best and we indeed observe that, for the synonymous positions, the simple model can reasonably accurately fit the observed frequencies, and even for the set of all alignment columns the fits are quite accurate (*Figure 3—figure supplement 1*). In contrast, for the second positions in codons, we can see the effects of selection in that, from the larger fractions of columns without SNPs, the model infers a lower $\mu_*$, and this leads to an underestimation of columns with four nucleotides. Thus, the true fraction $f_h$ is more likely close to the values inferred from the synonymous positions. Note that $f_h = 0.063$ when including the outgroup and $f_h = 0.033$ when the outgroup is excluded. The difference between these two estimates derives from the very high fraction of SNP columns in which the 9 strains of the outgroup have another nucleotide than all other strains. For this subset of SNPs, the fraction of columns that have more than one mutation is much higher than for any other SNP column. Thus, for all other SNP columns, the estimate that a fraction of 3.3% correspond to homoplasies is likely the most accurate. For completeness, we provide the full statistics of the observed polymorphisms in *Table 2*.

As an additional test, we also applied this simple method to estimate the rate of homoplasy for the simulation data. For the simulation data without recombination we explicitly kept track of homoplasies and determined that a fraction $f_h = 0.0256$ of bi-allelic positions correspond to homoplasies. Applying our estimation method to the alignment of the $S = 50$ sample genomes from this

**Table 1.** Estimated expected number of mutations per position $\mu_*$ and estimated fraction of homoplasies for five different subsets of core alignment columns: all columns, all synonymous positions (third positions in fourfold degenerate codons), second positions in codons, synonymous positions excluding the outgroup, and second positions in codons excluding the outgroup.

| Column set | $\mu_*$ | $f_h$ |
|---|---|---|
| All columns | 0.118 | 0.026 |
| Synonymous positions | 0.287 | 0.063 |
| Second positions in codons | 0.0258 | 0.006 |
| Synom. pos. without outgroup | 0.149 | 0.033 |
| Sec. pos. without outgroup | 0.0172 | 0.004 |

**Table 2.** Detailed statistics on the polymorphisms in the whole core genome alignment, at synonymous positions (third positions in fourfold degenerate codons) and at second codon positions (where all substitutions are non-synonymous).
First, for each set of positions the table lists the total number of positions, and the number of positions at which 1, 2, 3 or 4 different nucleotides appear. Second, for the subset of positions where two nucleotides appear, the table lists the total number of transitions and transversions, and the number of positions with each of the six possible two-nucleotide subsets.

| Statistic | All columns | Synom. codon pos. | Sec. codon pos. |
|---|---|---|---|
| Total columns | 2,880,516 | 349,311 | 960,172 |
| 1-letter columns | 2,484,831 | 299,536 | 936,588 |
| 2-letter columns | 363,164 | 46,807 | 22,505 |
| 3-letter columns | 30,611 | 2838 | 1029 |
| 4-letter columns | 1910 | 130 | 50 |
| Transitions | 275,134 | 36,866 | 13,420 |
| Transversions | 88,030 | 9941 | 9085 |
| A ↔ G | 138,728 | 21,767 | 6676 |
| C ↔ T | 136,406 | 15,099 | 6744 |
| G ↔ T | 23,679 | 2198 | 963 |
| A ↔ C | 23,636 | 3294 | 3510 |
| A ↔ T | 21,036 | 2416 | 2549 |
| C ↔ G | 19,679 | 2033 | 2063 |

simulation, we estimated $f_h = 0.0243$, which is within 2% of the true value, further supporting our method for estimating the homoplasy rate.

In summary, our estimates strongly suggest that the rate of homoplasies among bi-allelic SNPs in our core genome alignment of the *E. coli* strains lies somewhere in the range of 2-6%.

## Removing potentially homoplasic positions from the core genome alignment

Even though the analysis of the previous section has shown that homoplasies are only a very small fraction of all bi-allelic SNPs, we decided to investigate to what extent this small fraction of homoplasies may affect the various statistics that we calculate. Ideally, if we knew which alignment columns correspond to homoplasies, we could simply remove all homoplasic columns and recalculate all statistics of interest on the reduced alignment from which these homoplasic sites were removed. However, since we only know the approximate fraction $f_h$ of homoplasies and do not know which columns are homoplasies, a conservative approach is to remove those columns that are most phylogenetically inconsistent with other columns in their neighborhood.

In particular, for each bi-allelic SNP *s* in the alignment, we check its phylogenetic consistency with the nearest 200 SNP columns to the left and nearest 200 SNP columns to the right, that is its consistency with others SNPs within a roughly 4 kb region. We then assign each SNP column an inconsistency score $I_s$ corresponding to the fraction of the 400 neighboring columns that are phylogenetically inconsistent with it. We then sort all SNP columns by their inconsistency $I_s$ and remove a fraction *f* of SNP columns with the highest inconsistency. After this, we can recalculate all statistics of interest on the alignment from which these potentially homoplasic columns have been removed. In particular, we generated reduced alignments from which a fraction $f = 0.05$ and a fraction of $f = 0.1$ of most inconsistent columns were removed.

## Constructing a tree that maximizes the number of compatible SNPs

We classify all SNPs in the core genome alignment into *SNP types* as follows. For each bi-allelic SNP, we map all letters with the majority nucleotide to a 0 and the minority nucleotide to a one and sort the bits according to the alphabetic order of the strain names. For SNPs where one allele occurs in

exactly half of the strains the minority allele is not defined and the ambiguity is resolved by setting the first bit of the string to 0. In this way, each SNP is mapped to a binary sequence of length 92. This binary sequence defines the SNP type. Note that a SNP type corresponds to a particular bi-partition of the strains.

We next counted the number of occurrences $n_t$ of each SNP type $t$ and sorted the SNP types from most to least common. We then used the following greedy algorithm to a collect a subset $T$ of mutually compatible SNP types that accounts for as many SNPs as possible. We seed $T$ with the most common SNP type, that is the SNP type occurring at the top of the list. We then go down the list of SNP types, iteratively adding SNP types $t$ to the set $T$ that are compatible with all previous types in the set $T$.

## Bottom up tree building

In this procedure, we build phylogenies of subclades in a bottom-up manner, starting from the full set of 92 strains and iteratively fusing pairs, minimizing the number of incompatible SNPs at each step.

For any subset of strains $S$, we define the number of supporting SNPs $n_S$ as the number of SNPs that fall on the branch between the subset $S$ and the other strains, that is the number of SNPs in which all strains in $S$ have one letter, and all other strains another letter. Similarly, we define the number of clashing SNPs $c_S$ as the number of SNPs that are incompatible with the strains in $S$ forming a subclade in the tree.

The iterative merging procedure is initiated with each of the 92 strains forming a subclade $S$. At each step of the iteration we calculate, for each pair of existing subclades $S_1$, $S_2$, the number of clashing SNPs $c_S$ and supporting SNPs $n_S$ for the set of strains $S = S_1 \cup S_2$ consisting of the union of the strains in $S_1$ and $S_2$. We then merge the pair $(S_1, S_2)$ that minimizes the clashes $c_S$ and, when their are ties, maximizes the number of supporting SNPs $n_S$. At each step of the calculation, we keep track of the total number of SNPs on the branches of the subtrees build so far, as well as the total number of SNPs that are inconsistent with the subtrees build so far. In addition, we calculate the average pairwise divergences of the strains within the subclades. *Figure 4—figure supplement 4* shows the ratio of clashing to supporting SNPs as a function of the divergence within the subclades.

## Quartet analysis

Quartets were assembled in the following way. We construct a grid of target distances $d$ starting at 0.00001 and having 50 points with 0.0005 sized distance. For every target distance $d$, we scan the alignment for four strains which have all pairwise distances within 1.25-fold of distance $d$. Every target distance $d$ for which no quartet can be found fulfilling these criteria is ignored.

For each quartet, we extract all SNP columns where two strains have a specific nucleotide and the other two strains have another nucleotide. Every such SNP column unambiguously supports one out of three possible tree topologies for this quartet. For each quartet, we determine which topology has the largest number of supporting SNPs, and what the fraction of SNPs is that support this topology.

## Linkage disequilibrium measure

A standard measure of linkage disequilibrium of SNPs at a given distance is given by the average squared-correlation of the genotypes at these positions (*Lewontin, 1988*). For a pair of loci with bi-allelic SNPs there are four possible genotypes which we indicate as binary patterns 00, 01, 10, and 11. If the frequencies of these genotypes are $f_{00}$, $f_{01}$, $f_{10}$, and $f_{11}$, then the squared correlation is calculated as

$$r^2 = \frac{(f_{00}f_{11} - f_{01}f_{10})^2}{f_{1.}f_{0.}f_{.0}f_{.1}}, \tag{9}$$

where the variables with dots correspond to marginal probabilities, for example $f_{1.} = f_{10} + f_{11}$, $f_{.1} = f_{01} + f_{11}$, and so on.

## Distribution of tree-compatible stretches

To calculate the distribution of the number of consecutive tree-compatible SNPs, we start from each SNP $s$ in the core genome alignment and count the number $n_s$ of SNP columns immediately following $s$, until a SNP column occurs that is incompatible with at least one of the $n_s$ SNP columns. Similarly, to obtain the distribution of the number of consecutive tree-compatible nucleotides we start from each position $p$ in the core genome alignment and count the number $n_p$ of consecutive nucleotides until a SNP column occurs that is incompatible with at least one of the SNP columns among the $n_p$ nucleotides.

## Minimum number of phylogeny changes *C*

We iterate over all SNP columns along the core genome alignment and add the current SNP to a list if it is pairwise compatible with all SNPs currently in the list. If it is incompatible with at least one SNP in this list, we empty the list, re-initialize the list with the current SNP, and increase the phylogeny counter $C$ by one.

## Phylogenies of human genome sequences

We selected 40 individuals at random from the 1000 Genome project (*1000 Genomes Project Consortium et al., 2015*) and build a 'core tree' by applying PhyML to the sequences of chromosomes 1 through 12. To investigate the robustness of this core tree, we cut the alignment in 1000 bp blocks and did 100 random resamplings of half of the blocks, building a phylogeny for each resampling using PhyML. We then determined the fraction of times each split in the core tree occurred in the trees of the 100 resamplings.

## Power-law fits of *n*-SNP distributions

We extract each $n$-SNP from the core genome alignment and count the frequency, that is the number of occurrences, $f_t$ of each $n$-SNP type $t$ as well as the total number $T$ of $n$-SNP types that occur at least once. We assume the $n$-SNP type occurrences are drawn from a power-law of the form

$$P(f) = \frac{1}{\zeta(\alpha)} f^{-\alpha}, \tag{10}$$

where $\zeta(\alpha)$ is the Riemann zeta function defined by

$$\sum_{f=1}^{\infty} f^{-\alpha} = \zeta(\alpha). \tag{11}$$

The log-likelihood of the frequencies $f_t$ as a function of $\alpha$ is given by

$$L(\alpha) = -T \log[\zeta(\alpha)] - \sum_t \alpha \log[f_t] = -T(\log[\zeta(\%\alpha)] + \alpha \langle \log[f] \rangle), \tag{12}$$

where $\langle \log[f] \rangle$ is the average of the logarithm of the SNP-type frequencies. Using a uniform prior on $\alpha$, the posterior distribution of $\alpha$ is simply proportional to the likelihood function. The optimal exponent $\alpha_*$ is the solution of

$$\frac{\zeta'(\alpha_*)}{\zeta(\alpha_*)} = -\langle \log[f] \rangle. \tag{13}$$

To calculate error-bars on the fitted exponentials we approximate the posterior by a Gaussian by expanding the log-likelihood to second order around the optimal exponent $\alpha_*$. We then find for the standard-deviation of the posterior distribution:

$$\sigma(\alpha) = \frac{1}{\sqrt{T\left(\frac{\zeta''(\alpha_*)}{\zeta(\alpha_*)} - \frac{\zeta'(\alpha_*)^2}{\zeta(\alpha_*)^2}\right)}}. \tag{14}$$

## Entropy profiles of *n*-SNP distributions

For a given strain *X*, we first extract all SNP types *t* for which *X* is one of the strains that shares the minority nucleotide. We then further stratify these SNP types by the number of strains *n* sharing the minority nucleotide. For each *n*, we thus obtain a set $S(X, n)$ of *n*-SNPs in which strain *X* is one of the strains sharing the SNP. We denote the number of occurrences of an SNP of type *t* by $f_t$ and the total number of *n*-SNPs within set $S(X, n)$ as $|S(X, n)|$, that is

$$|S(X,n)| = \sum_{t \in S(X,n)} f_t \tag{15}$$

The entropy $H(X, n)$ of the *n*-SNP distribution of strain *X* is then defined as

$$H(X,n) = - \sum_{t \in S(X,n)} \frac{f_t}{|S(X,n)|} \log_2 \left[ \frac{f_t}{|S(X,n)|} \right]. \tag{16}$$

## Comparing *n*-SNP statistics of pairs of strains

The considerable variability of the entropy profiles of *E. coli*'s strains suggests that the lineages of different strains must have recombined at different rates with other lineages. Indeed, we observe much less variation in the entropy profiles of the data from simulations in which each strain recombines at an equal rate with each other strain, than we observe for the *E. coli* data (*Figure 10—figure supplement 3*). However, since we currently lack a concrete evolutionary model that can reproduce all the statistics that we observe in the data, it is difficult to quantify how different the recombination rates of different lineages have to be in order to reproduce the observed variation in entropy profiles.

Nonetheless, it is straight-forward to define a simple statistical measure of the difference in the *n*-SNP statistics of a given pair of strains $(x, y)$. Each SNP in the core genome alignment can be categorized by the subset of strains *S* that share the minority allele. Given a pair of strains $(x, y)$, this subset *S* can take on four possible types: $S_0 = (Z)$, $S_2 = (xyZ)$, $S_x = (xZ)$, and $S_y = (yZ)$, where *Z* is a subset of strains that does not include *x* or *y*. That is, either neither *x* or *y* carry the minority allele, they both do, or only one of them does. We are interested in comparing the relative frequencies with which *x* and *y* co-occur with different subsets *Z* in SNPs across the alignment. Note that to this end, we can ignore SNPs of the type $S_0$ and $S_2$ because *x* and *y* occur together in these SNPs, so that the frequency with which different subsets *Z* occur in types $S_0$ and $S_2$ is per definition the same for both *x* and *y*. Thus, the relevant SNPs are of the type $S_x$ and $S_y$.

Let $n_{xZ}$ be the number of SNPs of the type $S_x = (xZ)$, $n_{yZ}$ the number of SNPs of type $S_y = (yZ)$, and the totals $N_x = \sum_Z n_{xZ}$, and $N_y = \sum_Z n_{yZ}$. We can then define two probability distributions over all possible subsets *Z*, that is $p_Z^x = n_{xZ}/N_x$ and $p_Z^y = n_{yZ}/N_y$. These two probability distributions give the relative frequencies with which *x* and *y* are observed to occur in SNPs with all other sets of strains *Z* and there are standard methods to quantify to what extent these two probability distributions are statistically significantly different. A standard orthodox statistical test is the Fisher exact test, which uses the test statistic

$$P(x,y) = \frac{N_x! N_y!}{(N_x + N_y)!} \prod_Z \frac{(n_{xZ} + n_{yZ})!}{n_{xZ}! n_{yZ}!}. \tag{17}$$

Note that, if the counts are large enough so that we can use the Stirling approximation $\log(n!) \approx n \log(n) - n$, this can be rewritten as

$$P(x,y) = \exp\left[ -N_x D(p^x || p) - N_y D(p^y || p) \right], \tag{18}$$

where the distribution *p* is the marginal distribution over the sets *Z*

$$p_Z = \frac{n_{xZ} + n_{yZ}}{N_x + N_y}, \tag{19}$$

and $D(p||q)$ is the Kullback-Leibler divergence (or relative entropy) of the distribution *p* with respect to distribution *q*. Note that the probability $P(x, y)$ corresponds directly to the *p*-value of the Fisher exact test. The cumulative distribution of p-values across all pairs of strains is shown in *Figure 10—*

*figure supplement 2*, left panel, showing that the *n*-SNP profiles are different for about 95% of all pairs. In the right panel we show a scatter of the distance $-\log[P(x,y)]$ of the n-SNP frequency profiles of each pair of strains $(x,y)$ as a function of their nucleotide divergence $d(x,y)$. This shows that only very close pairs that differ by less than 10 SNPs in their core genomes have statistically identical *n*-SNP frequency profiles. In addition, there is a very high correlation between the nucleotide divergence $d(x,y)$ and the distance $-\log[P(x,y)]$ of the *n*-SNP frequency profiles.

Finally, one caveat of the Fisher exact test is that it presumes that all the observed *n*-SNPs of the types $(Zx)$ and $(Zy)$ are statistically independent, and in the absence of a concrete stochastic model for the evolutionary dynamics, we do not know whether this assumption holds. However, the p-values for most pairs are so low that, even if we assume the true numbers of independent events are 500-fold less, the fraction of significantly different pairs would still be larger than 90%.

## Acknowledgements

The authors thank Olin Silander and Diana Blank for their laboratory work on the sequencing of the SC strains. During the development of this work, the authors have benefited from discussions with many researchers including Olin Silander, Frederic Bertels, Sergei Maslov, Purushottam Dixit, Edo Kussell, Boris Shraiman, Eugene Koonin, Daniel Fisher, Paul Rainey, Oskar Hallatschek, Mikhail Tikhonov, Richard Neher, Bruce Levin, Otto Cordero, and Daniel Weissman. This work was supported by the Swiss National Science Foundation grant No. 31003A_135397. In addition, this work was done in part while the authors were visiting the Simons Institute for the Theory of Computing and was thus supported in part by NSF Grant No. PHY17-48958, NIH Grant No. R25GM067110, and the Gordon and Betty Moore Foundation Grant No. 2919.01. Calculations were performed at sciCORE (http://scicore.unibas.ch/), the scientific computing core facility of the University of Basel.

## Additional information

### Funding

| Funder | Grant reference number | Author |
| --- | --- | --- |
| Swiss National Science Foundation | 31003A_135397 | Erik van Nimwegen |

The funders had no role in study design, data collection and interpretation, or the decision to submit the work for publication.

### Author contributions

Thomas Sakoparnig, Conceptualization, Data curation, Software, Formal analysis, Validation, Investigation, Visualization, Methodology, Writing - original draft, Writing - review and editing; Chris Field, Conceptualization, Resources, Data curation, Investigation, Visualization, Methodology, Writing - review and editing; Erik van Nimwegen, Conceptualization, Formal analysis, Supervision, Funding acquisition, Validation, Investigation, Visualization, Methodology, Writing - original draft, Writing - review and editing

### Author ORCIDs

Chris Field https://orcid.org/0000-0002-6434-3745
Erik van Nimwegen https://orcid.org/0000-0001-6338-1312

### Decision letter and Author response

Decision letter https://doi.org/10.7554/eLife.65366.sa1
Author response https://doi.org/10.7554/eLife.65366.sa2

## Additional files

### Supplementary files

• Transparent reporting form

## Data availability

All data generated or analyzed during this study are available from public databases and links to all the source data are provided in the article.

The following datasets were generated:

| Author(s) | Year | Dataset title | Dataset URL | Database and Identifier |
|---|---|---|---|---|
| Field C, Sakoparnig T, van Nimwegen E | 2018 | Sequencing, assembly and annotation of naturalized *E. coli* isolates from Lake Superior Watersheds | https://www.ncbi.nlm.nih.gov/bioproject/?term=PRJNA432505 | NCBI BioProject, PRJNA432505 |
| Sakoparnig T | 2021 | Microbial recombination with population structure | https://doi.org/10.5281/zenodo.4420880 | Zenodo, 10.5281/zenodo.4420880 |

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

# Appendix 1

**Appendix 1—table 1.** Average length of tree compatible segments, both in terms of number of consecutive SNPs and number of consecutive nucleotides, for the full *E. coli* and simulation data, as well as for alignments from which 5% or 10% of potentially homoplasic sites were removed.

| Statistic | Average nb. of SNPs | | | Average nb. of nucleotides | | |
|---|---|---|---|---|---|---|
| Perc. homoplasies removed | 0% | 5% | 10% | 0% | 5% | 10% |
| *E. coli* | 7.59 | 8.79 | 10.75 | 123 | 145 | 179 |
| $\rho/\mu = 0$ | 32 | 2757 | 11606 | 387 | 30435 | 108413 |
| $\rho/\mu = 0.001$ | 32 | 430 | 702 | 371 | 5020 | 8396 |
| $\rho/\mu = 0.01$ | 28 | 57 | 85 | 365 | 733 | 1158 |
| $\rho/\mu = 0.1$ | 14 | 16 | 18 | 211 | 256 | 312 |
| $\rho/\mu = 0.3$ | 8 | 9 | 10 | 135 | 157 | 183 |
| $\rho/\mu = 1$ | 5 | 5 | 5 | 88 | 100 | 114 |
| $\rho/\mu = 10$ | 3 | 3 | 3 | 57 | 62 | 68 |

**Appendix 1—table 2.** Number of substitutions and phylogeny changes within sub-alignments corresponding to known phylogroups.

Starting from the full 5% homoplasy-corrected core genome alignment we extracted, for each phylogroup, the sub-alignment of all strains belonging to that phylogroup and determined the number of bi-allelic SNPs $M$ and number of phylogeny changes $C$. Each row of the table corresponds to one of the known phylogroups and shows the number of our strains in that phylogroup, the SNP rate within the clade $M/L$, the lower bound $C$ on the number of phylogeny changes, the ratio $C/M$, and the estimated average number of times each position in the genome has been overwritten by recombination $T_{\text{est}}$. Note that the phylogroup 'O' stands for the outgroup. See *Figure 1—figure supplement 1* for the phylogroup annotation of our strains.

| Phylogroup | No. of strains | SNP rate $M/L$ | Phyl. changes $C$ | $C/M$ | $T_{\text{est}}$ |
|---|---|---|---|---|---|
| A | 6 | 0.0024 | 178 | 0.027 | 0.65 |
| B1 | 35 | 0.013 | 4540 | 0.130 | 16.5 |
| B2 | 6 | 0.011 | 2664 | 0.088 | 9.7 |
| D | 29 | 0.017 | 5426 | 0.114 | 19.7 |
| E | 1 | - | - | - | - |
| F | 3 | 0.007 | - | - | - |
| O | 9 | 0.003 | 2 | 0.0002 | 0.007 |

**Appendix 1—table 3.** Summary statistics of the core genome alignments of the different bacterial species.

For each species, the number of strains, the median genome size, the size of the core genome alignment, and the total number of informative SNPs (that is SNPs that occur in at least two strains) are listed.

| Species | Strains | Genome size | Core size | Inf. SNPs |
|---|---|---|---|---|
| *Escherichia coli* | 92 | 4,929,299 | 2,756,541 (56%) | 247,822 |
| *Bacillus subtilis* | 75 | 4,155,843 | 2,341,553 (56%) | 182,535 |
| *Helicobacter pylori* | 83 | 1,655,288 | 850,827 (51%) | 114,993 |

*Continued on next page*

*Appendix 1—table 3 continued*

| Species | Strains | Genome size | Core size | Inf. SNPs |
|---------|---------|-------------|-----------|-----------|
| *Mycobacterium tuberculosis* | 40 | 4,465,985 | 4,150,139 (93%) | 3502 |
| *Salmonella enterica* | 155 | 4,810,980 | 2,846,634 (59%) | 192,117 |
| *Staphylococcus aureus* | 95 | 2,881,899 | 2,002,833 (69%) | 73,756 |

**Appendix 1—table 4.** Summary statistics on mutation and recombination for each of the six species. For each species, the table shows the SNP rate (SNPs per alignment column) $M/L$, the lower bound on the number of phylogeny changes $C$, and the lower bound on the ratio of phylogeny changes to mutations $C/M$.

| Species | SNP rate $M/L$ | Phyl. changes $C$ | $C/M$ |
|---------|----------------|-------------------|-------|
| *Escherichia coli* | 0.101 | 43,575 | 0.156 |
| *Bacillus subtilis* | 0.113 | 40,811 | 0.155 |
| *Helicobacter pylori* | 0.202 | 50,743 | 0.295 |
| *Mycobacterium tuberculosis* | 0.002 | 755 | 0.078 |
| *Salmonella enterica* | 0.085 | 31,598 | 0.131 |
| *Staphylococcus aureus* | 0.053 | 13,215 | 0.124 |

## Appendix 2

### Comment on the estimation of recombination segment lengths

The recombination segment lengths that we inferred from comparing closely related pairs of strains (*Figure 2J*), and the average segment length of $L_r = 31kb$, are significantly longer than estimates that have been reported previously, for example (*Vos and Didelot, 2009*; *Touchon et al., 2009*). In contrast to our method, which directly identifies recombined segments as regions of high SNP density in pairwise genome alignments, which can be easily detected by eye (i.e. see *Figure 2A and B*), these previous estimates were based on fitting of more complex population genetics models to the entire multiple alignment. Since these models make a number of assumptions that our results suggest do not hold, this may explain why these models inferred significantly lower fragment lengths. However, one may still ask whether our estimate of the average recombination segment length could be significantly inflated, for example due to our method failing to detect short recombination events.

We do not believe this is plausible for a number of reasons. First, the simple HMM that we used to detect recombined segments uses 1 kb as a minimal segment length, and as much as 30% of the recombination segments that we infer are only $1 - 2$ kb long (*Figure 2J*). This shows that our method has no difficulty in detecting relative short recombination segments.

Second, a simple back-of-the-envelope calculation suggests that segments have to be very short in order for our method to fail to detect them. For the close pairs that we use to estimate the lengths of recombination segments, SNPs in clonally inherited regions are so rare that essentially only 1 kb blocks with 0, 1, or at most 2 SNPs will be assumed to be clonally inherited. Since the recombined segments derive from strains that are typically 1–2% diverged, even a recombination segment of length 200 bp would cause three or more SNPs, and would be detected by our method. Thus, very roughly speaking, segments of length 200 bp or more are expected to be reliably detected and only segments of 100 bp or less could easily be missed.

Third, to substantially affect the average length of the recombination segments, undetectable short segments (i.e. of length less than 100 bp) would have to be extremely frequent. That is, if we denote by $L_r = 31000$ the average length of the detected segments, by $L_u$ the average length of undetected segments, by $L_t$ the true average segment length, and by $f_d$ the fraction of all segments that were detected, we have

$$f_d \left(1 - \frac{L_u}{L_r}\right) = \frac{L_t - L_u}{L_r}. \tag{20}$$

In the limit that $L_u$ is much smaller than both $L_r$ and $L_t$, we have approximately $f_d \approx L_t/L_r$. Thus, in order for the average segment length to be overestimated 10-fold, we would have to assume that for every detected recombination segment of length 200 bp or more, there are 10 undetected recombination segments of length 100 or less. This seems rather implausible to us.

Finally, the average length of the recombination segments $L_r$ is immaterial for almost all of our results, with the exception of our estimated lower bound on the number of times $T$ each locus has been overwritten by recombination. In particular, this lower bound $T$ is directly proportional to the average length of the recombination segments $L_r$. However, the estimate $T$ is also directly proportional to the lower bound $C$ on the number of phylogeny changes along the genome, and we saw in the main text that in our simulations the true number of phylogeny changes can easily be 10-fold larger than this lower bound $C$. Therefore, our lower bound for $T$ can only be larger than the true value if the average segment length $L_r$ is overestimated by a large factor.

## Appendix 3

### Estimating *T* using statistics from the pair analysis

For a pair of strains $(s, s')$ our pairwise analysis estimates the fraction $f_c$ of the genome that was clonally inherited and the divergence $d_c$ (i.e. fraction of positions for which $s$ and $s'$ have different nucleotides) in the clonally inherited regions, from the density of SNPs along their core genome alignment. If we make the assumptions that, since $s$ and $s'$ diverged from a common clonal ancestor, substitutions have occurred at a constant rate $\mu$ per unit time, recombination events have occurred at a constant rate $\rho$ per unit time, and the average length of the recombination segments is $L_r$, then $f_c$ is approximately given by

$$f_c(t) = e^{-2L_r \rho t / L_g},\tag{21}$$

with $L_g$ the length of the core genome alignment, and $t$ the time since the common ancestor of $s$ and $s'$. Similarly, the divergence $d_c$ is given by

$$d_c(t) = 1 - e^{-2\mu t / L_g}.\tag{22}$$

Using these, we can solve for the clonal fraction $f_c$ as a function of the divergence $d_c$ and find

$$f_c = (1 - d_c)^{L_r \rho / \mu}\tag{23}$$

Thus, given an estimate of the fraction of clonally inherited genome $f_c$ and divergence $d_c$ in these clonally inherited regions, we can infer an effective value of $L_r \rho / \mu$ for this pair, that is

$$\frac{L_r \rho}{\mu} = \frac{\log[f_c]}{\log[1 - d_c]}.\tag{24}$$

In order for these simple equations to be applicable, we have to be in a regime where there are enough recombined regions so that their number and average size is not too noisy, but also not so many that the recombined regions start to significantly overlap. We thus extracted all pairs of strains for which $1/4 \leq f_c \leq 3/4$ and plotted the estimate of $L\rho/\mu$ against $f_c$ for each pair (*Figure 1*).

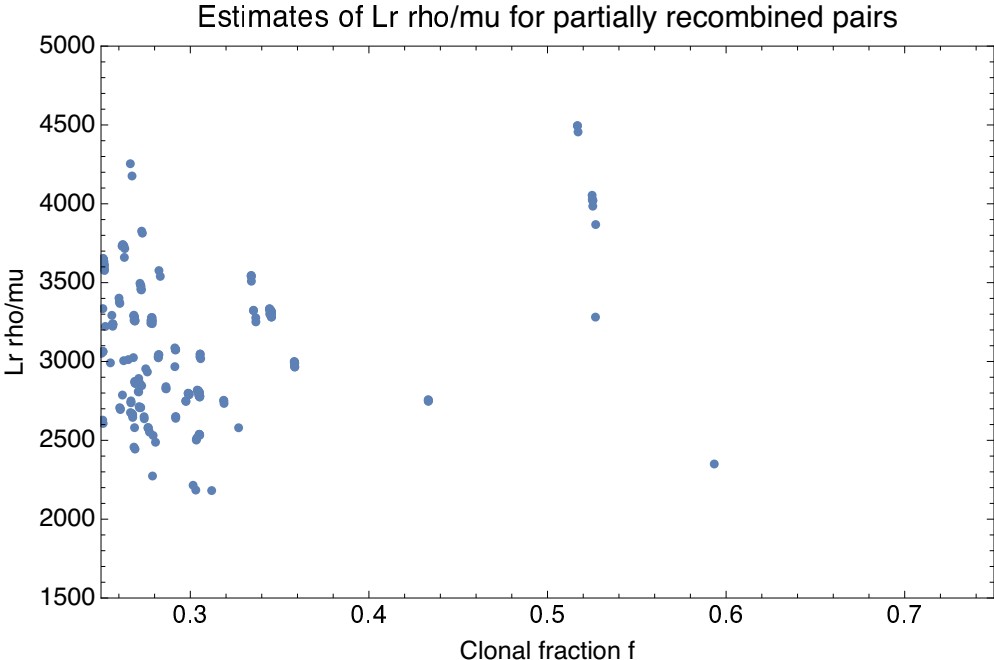

**Appendix 3—figure 1.** Estimate of the effective recombination strength $L_r \rho / \mu$ (vertical axis) as a function of the estimated fraction of clonally inherited genome $f_c$ (horizontal axis) for each pair of strains with $f_c$ between $1/4$ and $3/4$. Each point corresponds to a pair of strains.

We see that based on this simple model, the estimates of $L_r\rho/\mu$ range from about 2100–4600.

Finally, we can use these estimates of $L_r\rho/\mu$ to derive an estimate for the average number of times $T$ that a given locus in the core genome alignment has been overwritten by recombination. Note that, because recombination is very frequent, many thousands of different phylogenies occur along the core genome, but each single position $i$ has some definite phylogeny $\phi_i$. If we denote by $t(\phi_i)$ the total length of the branches of phylogeny $\phi_i$, then the expected number of times position $i$ has been substituted is roughly $\mu t(\phi_i)$ and the expected number of times position $i$ has been recombined is $L_r\rho t(\phi_i)$. From the observation that 10% of the positions in the core genome alignment are polymorphic, it follows that $\mu\langle t\rangle \approx 0.1$, with $\langle t\rangle$ the average of $t(\phi_i)$ across all positions. Consequently, the average number of times $T$ that a position in the genome has been overwritten by recombination is given by

$$T = L_r\rho\langle t\rangle \approx 0.1\frac{L_r\rho}{\mu}. \tag{25}$$

Thus, this simple model predicts, from the statistics of the partially recombined pairs, that $T$ lies somewhere between 210 and 460, which is consistent with our lower bound of $T = 190$ that we estimated based on $C/M$ and the estimated average length of recombination segments $L_r$.

In spite of this consistency, we want to stress that this simple model assumes that there is a fixed rate of recombination, whereas our data indicates that recombination rates vary strongly across lineages. As such, it is inherently misleading to try to summarize the strength of recombination by a single recombination rate $\rho$. Moreover, in going from the estimate of $L_r\rho/\mu$ of each pair, to the estimate of $T$, we implicitly assume that the rates inferred for these partially recombined pairs can be extended to the full core genome alignment. This too is potentially problematic. We therefore consider this method for estimating $T$ to be only a rough order-of-magnitude check on the lower bound on $T$ that we estimated from the lower bound on the number of phylogeny changes $C$.

# Appendix 4

## *E. coli*'s *n*-SNP statistics differ from *n*-SNP statistics of simulations with free recombination

In this section, we systematically compare the *n*-SNP statistics observed for the *E. coli* data with *n*-SNP statistics observed for the simulations with different rates of recombination. We show that the *n*-SNP statistics for *E. coli* differ in several fundamental ways from those observed for simulations of populations of freely recombining individuals.

An often considered statistic in the population genetic analysis of sexually reproducing species is the so-called site frequency spectrum, which in our terminology corresponds to the total number of *n*-SNPs $T(n)$, that is the total number of SNPs shared by *n* strains, as a function of *n*. *Figure 1* shows the site frequency spectrum $T(n)$ for the *E. coli* data, the simulations without recombination, and the simulations with recombination to mutation rates ranging from $\rho/\mu = 0.001$ to 10, using the 5% homoplasy-corrected alignments. To distinguish the contribution of potentially clonal SNPs falling on the core tree, the solid lines show $T(n)$ for all SNPs, and the dashed lines show $T(n)$ for all but the *n*-SNPs falling on the core tree.

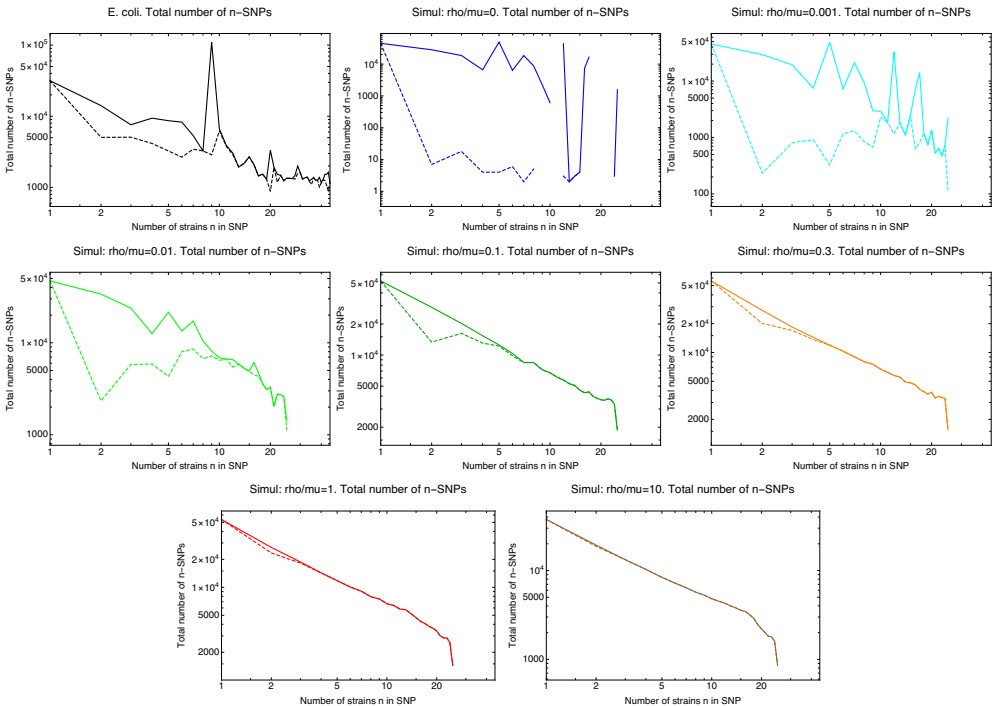

**Appendix 4—figure 1.** SNP frequency spectra. Total number of occurrences of n-SNPs, that is SNPs shared by *n* strains (vertical axes) as a function of *n* (horizontal axes) for the *E. coli* data (top left panel) and all simulated data with different recombination rates, where the ratio of recombination to mutation rate $\rho/\mu$ is indicated above each panel. The solid lines correspond to all *n*-SNPs in the 5% homoplasy-corrected alignment, whereas for the dashed lines all *n*-SNPs corresponding to branches in the core tree have been removed. Note that all axes are shown on logarithmic scales and for high recombination rates $\rho/\mu \geq 0.1$, the *n*-SNP frequencies are approximately proportional to $1/n$.

We first focus on the simulation data. It is well-known that, for populations evolving under a Kingman coalescent, $T(n) \propto 1/n$, that is the total frequency of SNPs shared by *n* strains falls as $1/n$ when averaged over many instantiations of the coalescent process. Indeed, for recombination rates sufficiently high that essentially all branches in the clonal history are dominated by recombination, that is $\rho/\mu \geq 0.3$, the observed site frequency spectrum falls exactly as $1/n$, and removing the SNPs that fall on the core tree has virtually no effect on the *n*-SNP frequencies $T(n)$. In contrast, for the simulations without recombination (top center panel in *Figure 1*) virtually all SNPs fall on the clonal tree, that is after removal of the core tree SNPs only a tiny number of SNPs remain that derive from homoplasies

that escaped the homoplasy correction. Moreover, because the counts $T(n)$ for the simulation without recombination derive from a single tree, we observe significant deviations from the average trend $T(n) \propto 1/n$, for example the clonal tree happens to have particularly long branches toward clades with *n*=5, *n*=7, and *n*=12 .

For $\rho/\mu = 0.001$ all loci are still predominantly clonally inherited across all branches of the tree (see *Figure 1—figure supplement 3*), which is reflected in the fact that the number of SNPs drops dramatically when the clonal SNPs are removed, and that we clearly see the peaks in $T(n)$ at $n = 5$, $n = 7$, and $n = 12$ that derive from the clonal tree. The recombination rate $\rho/\mu = 0.01$ is an interesting intermediate case. While most branches of the tree are predominantly clonally inherited, a few longer branches are already mostly recombined (*Figure 1—figure supplement 3*). Reflecting this, we see that for $\rho/\mu = 0.01$ clonal SNPs dominate the *n*-SNP counts for small *n*, whereas from about $n = 10$ onward, the *n*-SNP counts derive mostly from recombination. At recombination rate, $\rho/\mu = 0.1$ clonal SNPs only affect *n*-SNP counts for *n*<5, and for higher recombination rates the site frequencies $T(n)$ perfectly follow the function $1/n$.

For the *E. coli* data, we see that core tree SNPs make a significant contribution to the *n*-SNP counts for $n \leq 9$. In particular, the peak at $n = 9$, corresponding to the extremely common SNPs toward the outgroup, as well as a smaller peak at $n = 20$, corresponding to a group of 20 extremely close strains, both disappear after removal of the core tree SNPs. From $n = 10$ onward the core tree SNPs do not contribute significantly to the *n*-SNP counts $T(n)$. Interestingly, while the counts $T(n)$ roughly follow the $1/n$ trend until $n = 20$, for *n*>20 the counts $T(n)$ do not further decrease but appear virtually constant. This behavior is not observed in any of the simulated datasets.

Next, we investigated how many different *types* of *n*-SNPs occur as a function of *n* for the different datasets, as well as the average number of occurrence of the each of the *n*-SNP types. Since the exponents of the power-law fits are determined by the geometric average of the *n*-SNP type occurrences (see Materials and methods), we calculated the geometric average of the counts per *n*-SNP type as a function of *n* for each dataset. To focus on the *n*-SNPs associated with recombination, we used the SNPs from the 5% homoplasy-corrected alignment and removed all SNPs corresponding to branches of the core tree. The results are shown in *Figure 2*.

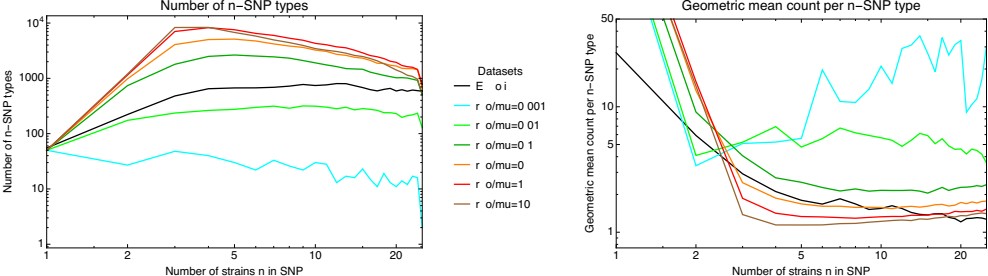

**Appendix 4—figure 2.** Diversity of *n*-SNPs. Left panel: Number of *n*-SNP types (vertical axis) as a function of *n* (horizontal axis) for the *E. coli* data (black line) and for simulations with different recombination to mutation rates $\rho/\mu$ (colored lines, see legend). Both axes are shown on a logarithmic scale. Right panel: Geometric average of the number of occurrences per *n*-SNP type (vertical axis) as a function of *n* for the *E. coli* data (black line) and for simulations with different recombination to mutation rates $\rho/\mu$ (colored lines, see legend). Both axes are shown on a logarithmic scale. Note that, in order to better show the heights of the curves for the different datasets the vertical axis is clipped at 50. For the simulated datasets, the geometric mean of the 1-SNP counts runs from 400 to 1000.

We first note that the number of 1-SNP types is 50 for all the simulated data (which consisted of a sample of $N = 50$) genomes. That is 1-SNPs are observed that are exclusive to each of the 50 genomes. However, the number of 1-SNP types observed for the *E. coli* data is 55, that is much less than the 92 possible types given that there are 92 strains. This reflects the fact that, in our collection of wild *E. coli* isolates, there are some groups of extremely closely related strains with identical core genomes. Note that such closely related strains are unlikely to occur for random samples from a Kingman coalescent.

Second, we see that, for the *E. coli* data, the number of *n*-SNP types increases with *n*, saturating at around $500 - 800$ unique *n*-SNP types for each *n* at $n \geq 5$. In the right panel of **Figure 2** we see that, as the number of *n*-SNP types increases, the geometric mean frequency of the *n*-SNP types decreases smoothly. This smooth decrease of the geometric mean mirrors the increasing exponents of the *n*-SNP power-laws. None of the simulation data resemble the pattern that is observed for the *E. coli* data. First, at the lowest recombination rate $\rho/\mu = 0.001$ the number of *n*-SNP types is low and decreasing with *n*. Second, for recombination rate $\rho/\mu = 0.01$, the number of SNP types increases more gradually with *n*, saturating at about 200 *n*-SNP types for large *n*. However, in contrast to what we observe for the *E. coli* data, the geometric mean number of occurrences for the simulations with $\rho/\mu = 0.01$ is much larger and virtually constant for $n \geq 2$. At higher recombination rates $\rho/\mu \geq 0.1$, the number of *n*-SNP types rises quickly with *n*, reaching a maximum of thousands of unique *n*-SNP types between $n = 3$ and $n = 5$, and then decreases with larger *n*. At the same time, the geometric mean of the number of occurrences per type decreases quickly with *n* and stays at low values of about three for $\rho/\mu = 0.1$ and close to one for the highest recombination rates. That is, when recombination completely dominates, and there is no population structure by construction, almost all SNP types are distinct, occurring only once or a few times.

Note that our previous analysis has shown that the number of phylogeny changes per SNP column C/M for *E. coli* is similar to that observed for $\rho/\mu = 1$. However, as **Figure 2** shows, at such high recombination rates one would expect much higher *n*-SNP diversity still than is observed for the *E. coli* data. The fact that the *E. coli* data shows evidence of high recombination on the one hand, in combination with lower SNP diversity, is consistent with SNP diversity being constrained by population structure.

We next compared the approximately power-law *n*-SNP frequency distributions that we observed for *E. coli*, with the *n*-SNP frequency distributions observed for the simulation data. **Figure 3** shows the *n*-SNP distributions for six different values of *n* ranging from $n = 2$ to $n = 12$.

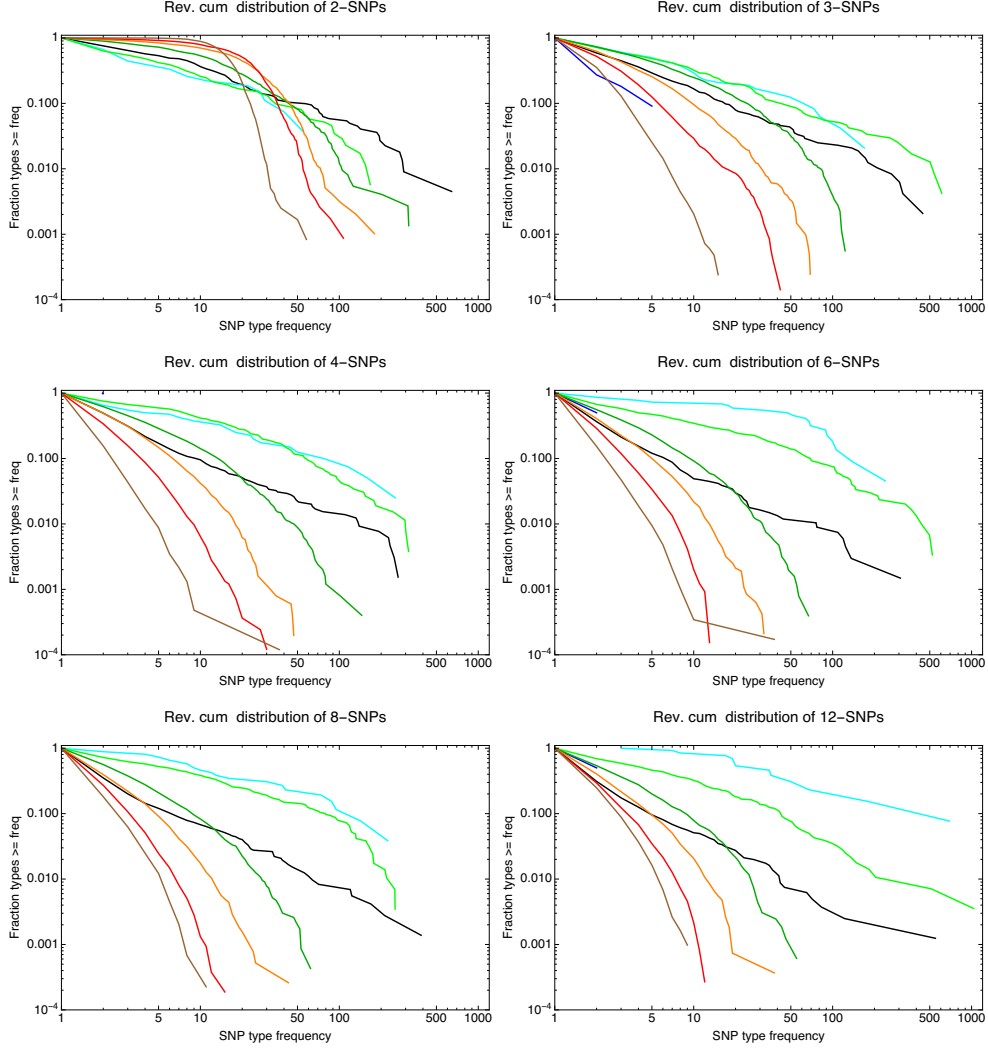

**Appendix 4—figure 3.** Example *n*-SNP distributions for *E. coli* (black lines) as well as for the simulations with different rates of recombination with $\rho/\mu = 0$ shown in blue, $\rho/\mu = 0.001$ in cyan, $\rho/\mu = 0.01$ in light green, $\rho/\mu = 0.1$ in dark green, $\rho/\mu = 0.3$ in orange, $\rho/\mu = 1$ in red, and $\rho/\mu = 10$ in brown. Each panel corresponds to the observed *n*-SNP distributions with the value of *n* indicated at the top of each panel. All axes are shown on logarithmic scales.

Whereas, for the *E. coli* data, all distributions are approximately power-law, almost all of the distributions for the simulated data are clearly bending downwards in the log-log plot, often severely so. Long tailed distributions are only observed for the very low recombination rates $\rho/\mu = 0.001$ and $\rho/\mu = 0.01$, for which we have seen that *n*-SNPs are still largely dominated by clonal SNPs at small values of *n*. However, as we have seen above, such low recombination rates are not consistent with the *E. coli* data. In addition, as we will see in more detail below, the exponents of the *n*-SNP distributions at recombination rates $\rho/\mu = 0.001$ and $\rho/\mu = 0.01$ are much lower than is observed for *E. coli*, and the average number of occurrence per *n*-SNP type are significantly higher than observed for *E. coli* (*Figure 2*, right panel).

All *n*-SNP distributions at recombination rates $\rho/\mu \geq 0.1$ differ clearly from power-laws. That is, instead of long-tailed distributions most of these *n*-SNP frequencies fall within a fairly narrow range. For example, at $\rho/\mu = 1$ most 2-SNPs have frequencies between $10 - 50$, most 3-SNPs have frequencies between $2 - 20$, and 4-SNPs that occur more than 10 times are very rare.

Although the *n*-SNP distributions at mutation rates $\rho/\mu \geq 0.1$ cannot reasonably be fitted to power-laws, we can still determine the exponents of the maximum likelihood fits since these only depend on the geometric average number of occurrences of the *n*-SNP types (see Materials and

methods). **Figure 4** shows the resulting exponents for the simulated data, as well as for the *E. coli* data for reference.

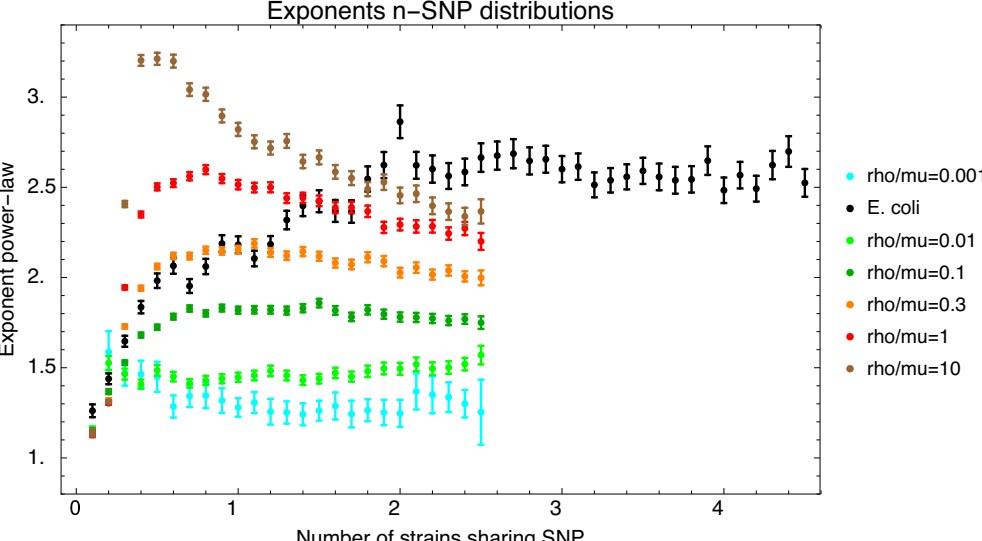

**Appendix 4—figure 4.** Fitted exponents of the *n*-SNP distributions for the *E. coli* data (black) and the data from the simulations with different recombination rates (colors, see legend). The bars show the fitted exponent plus and minus one standard-deviation of the posterior distribution.

We see that, for the simulation data, exponents increase systematically with recombination rate. For low recombination rates $\rho/\mu = 0.001$ and $\rho/\mu = 0.01$ all exponents are below 1.5, whereas for recombination rate $\rho/\mu = 1$ the exponents lie in the range $2.3 - 2.6$. In contrast to the *E. coli* data, for which exponents increase smoothly from about 1.25–2.7 as *n* increases, for the simulations the exponents tend to be more constant as a function of *n*.

In summary, none of the *n*-SNP statistics of the data from simulations resemble the *n*-SNP statistics observed for the *E. coli* data. For example, while fairly long-tailed distributions of *n*-SNPs are observed at very low recombination rates, the exponents of these distributions are much lower than found for the *E. coli* data, and the *n*-SNP diversity is lower than observed for the *E. coli* data. In addition, such low recombination rates are inconsistent with all of our other analyses of the *E. coli* data. In contrast, while high *n*-SNP diversity is observed at higher recombination rates, there the *n*-SNP distributions are not long tailed, but instead *n*-SNP frequencies vary only over a relatively narrow range.

# Appendix 5

## Detailed SNP statistics for Phylogroup B2

In the main text, we illustrated 2-SNPs using strain A1 (*Figure 9*), which is part of the phylogroup B2. The phylogroup B2, which is represented by the six strains A1, A2, A7, A11, B2, and D8 in our dataset, is small enough to allow for an illustrative discussion of what our various analyses show about the role of recombination in the evolutionary history of these strains.

## Pairwise analysis for the sextet of strains (A1,A2,A7,A11,B2,D8)

*Appendix 5—table 1* shows the pairwise divergences between strain A1 and the other strains of this phylogroup.

**Appendix 5—table 1.** Pairwise divergence of strain *A1* with each of the other strains of phylogroup B2.

| Strain | Divergence |
|---|---|
| B2 | 0.00626 |
| A7 | 0.00627 |
| A11 | 0.00702 |
| D8 | 0.00729 |
| A2 | 0.00778 |

Note that these divergences are in the regime where, in general, most of the pairwise alignment has already been overwritten by recombination (see *Figure 2G* and note that the critical divergence at which 50% of the genome has recombined is 0.0032). Indeed, if one looks at the pairwise alignments of A1 with these strains (*Figure 1*), one finds that for each of these five strains there is little if any ancestrally inherited DNA left.

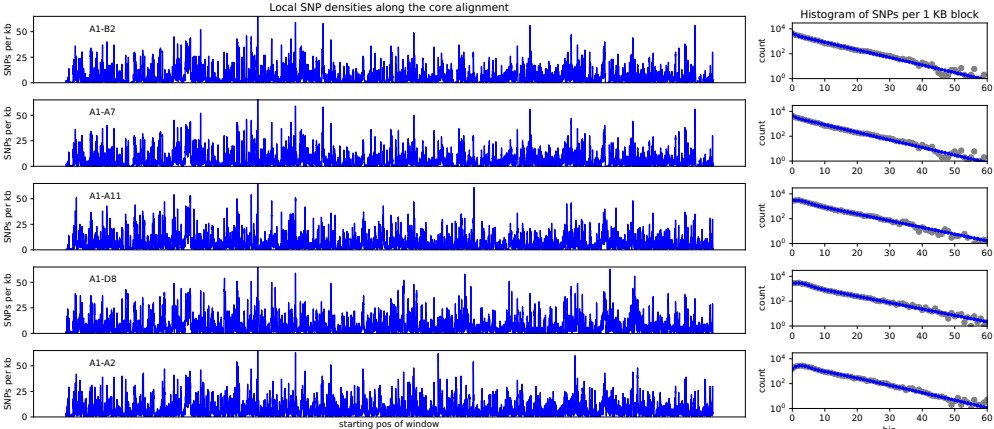

**Appendix 5—figure 1.** Left panels: SNP densities (SNPs per kilobase) along the core genome for the five pairs of strains (A1-B2), (A1-A7), (A1-A11), (A1-D8), and (A1-A2). Right panels: Corresponding histograms for the number of SNPs per kilobase (dots) together with fits of the mixture model. Note the vertical axis is on a logarithmic scale.

This is in fact true for *all* pairs of the phylogroup B2, with the exception of the pair (A7,B2). The divergence of A7 and B2 is less than $10^{-4}$ and their alignment is fully clonal, that is A7 and B2 share a recent common ancestor. However, according to the analysis of SNP densities along the pairwise alignments, all other pairs in this clade are close to fully recombined. Note that this also means that the pairwise divergences are dominated by mutations in recombined regions, not by ancestrally inherited mutations (see *Figure 2H*). Thus, the pairwise analysis rejects that the core genome phylogeny for this phylogroup corresponds to the clonal phylogeny.

## Pair-SNPs of A1 along the core genome alignment

In the main text, we introduced the 2-SNP concept by showing the distribution of 2-SNPs for the strain A1, which belongs to phylogroup B2. This analysis showed that strain A1 occurs in 2-SNPs with 17 different other strains from our collection, and most commonly with strains D8, A2, and A11, which each have around 200 2-SNPs with strain A1. Besides these three strains from phylogroup B2, strains D4 (from phylogroup F) and H6 (which does not have a clear phylogroup) each also have over 70 2-SNPs with A1. *Figure 2* shows how these five most common 2-SNP types involving strain A1 are distributed along the core genome alignment. Note that, wherever a 2-SNP occurs of type $(A1, s)$, it means that the strains A1 and $s$ are nearest neighbors in the phylogeny at that locus. Consequently, the pattern of 2-SNP types along the core genome alignment gives an indication of how different phylogenies are distributed along the genome.

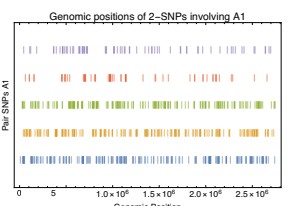 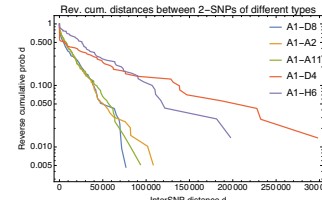 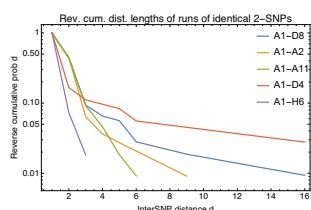

**Appendix 5—figure 2.** Distribution of the five most common 2-SNP patterns involving strain A1 along the core genome alignment. Left: Direct visualization of the positions of each of the 2-SNP patterns along the core genome alignment. Each dashed line corresponds to an SNP and SNPs are colored according to the 2-SNP type (see legend in middle and right panels) and ordered with the most common type at the bottom. Middle: Reverse cumulative distributions of the distances between consecutive SNPs of the same type. Each colored line corresponds to the distribution for one 2-SNP type (see legend). The vertical axis is shown on a logarithmic scale. Right: Reverse cumulative distribution of the length of runs of consecutive 2-SNPs of the same type. Each colored line corresponds to the distribution for one 2-SNP type (see legend). The vertical axis is shown on a logarithmic scale.

The results in *Figure 2* are consistent with the pairwise analysis, showing that the different 2-SNP types are fairly uniformly spread along the core genome alignment, and that the phylogeny changes many times. As shown in the middle panel of *Figure 2*, the tails of the distributions of distances between 2-SNPs of the same type are approximately exponential, indicating that 2-SNPs of the same type occur approximately at a constant rate on a large length-scale. However, on short length-scales there is some evidence for clustering of 2-SNPs of the same type. To further elucidate this, the right panel of *Figure 2* shows the reverse cumulative distribution of consecutive runs of the same 2-SNP type. This indicates that these runs are mostly very short, that is 3 or less 2-SNPs of the same type in a row for 90% of the runs. The longest runs are one run of 16 SNPs in a row of the most common type (A1,D8), and one run of 16 SNPs in a row where strain A1 shares a SNP with the strain D4 from phylogroup F. The fact that 2-SNPs of A1 with strains D4 and H6, both of which do not belong to phylogroup B2, are dispersed all across the core genome alignment underscores again that phylogroups should not be thought of as reflecting *clonal* ancestry, that is there is no single common ancestor cell of the strains in phylogroup B2. Instead, different positions in the core genome alignment have different ancestries and the strains of phylogroup B2 form a 'group' in the sense that, for many positions in the alignment, these strains share ancestors with each other more recently than with other strains.

## Phylogeny changes along the core alignment of phylogroup B2

That recombination is pervasive within this set of strains with relatively low divergence is confirmed by the phylogenetic inconsistencies along the alignment made from the core genomes of just this sextet of strains (A1,A2,A7,A11,B2,D8). In particular, in the 5% homoplasy-corrected alignment there are $M = 30'196$ SNPs and at least $C = 2664$ changes in phylogeny along this core genome alignment. This means that the lower bound on the ratio of the number of phylogeny changes to mutation

events is $C/M = 0.088$, that is a break on average every 11 SNPs. *Figure 3* shows the distribution of segment lengths between phylogeny breaks (either counting the number of consecutive SNPs or total length of the segments) which shows that breaks typically occur within 10 SNPs and within a few hundred base pairs.

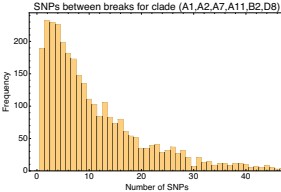 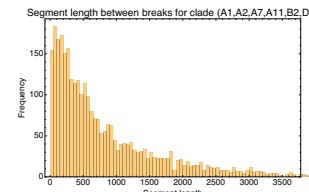 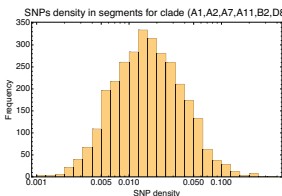

**Appendix 5—figure 3.** Left: Histogram of the number of consecutive SNPs before an inconsistency in the core genome alignment of the sextet of strains (A1,A2,A7,A11,B2,D8) of phylogroup B2. Note that the vertical axis corresponds to the number of segments with the corresponding number of consecutive SNPs. Middle: Histogram of the length of segments without phylogeny breaks. Right: Histogram of the SNP density across the segments that are consistent with a single phylogeny (shown on a logarithmic axis). Note that the SNP densities vary by two orders of magnitude.

*Figure 3* also shows (right panel) that the SNP density varies by as much as hundred-fold across the 2664 segments, consistent with the fact that the transferred fragments are themselves mosaics of previous recombination events.

## SNP types for the strains in phylogroup B2

Another indication that these single-phylogeny segments are the result of recombination comes from looking at what the actual SNP types are that occur in the segments. *Appendix 5—table 2* shows, for the 10 longest segments, the lengths, the total number of SNPs, and SNP-types that occur in each segment.

**Appendix 5—table 2.** Lengths, total SNP count, and SNP types (that is the strains that carry the minority allele) for the ten longest segments (in terms of number of consecutive SNPs) along the core genome of the sextet (A1,A2,A7,A11,B2,D8) of phylogroup B2.

| Length segment | Number of SNPs | SNP types |
|---|---|---|
| 9698 | 109 | (A7, B2) (A1, A7, B2) (A2, D8) |
| 5672 | 100 | (A7,B2) (A1,A2) (A11,D8) |
| 2068 | 100 | (A2,D8) (A11,A7,B2) |
| 2255 | 95 | (A7,B2) (A11,A2) (A1,A11,A2) |
| 11726 | 95 | (A7,B2) (A2,A7,B2) (A1,D8) |
| 11790 | 93 | (A7,B2) (A2,D8) |
| 3614 | 86 | (A11,A2) |
| 4564 | 86 | (A11,A7,B2) (A1,A2) |
| 2890 | 75 | (A7,B2) (A1,A7,B2) |
| 2390 | 71 | (A1,A11) |

Note that the SNP-types that occur in these longest unbroken segments are not only almost all inconsistent with the core genome phylogeny, they are also almost all inconsistent with each other.

That is, each of these segments suggests a different phylogeny. Note also that, in each segment, there are typically multiple SNPs of the same type.

**Appendix 5—table 3.** All 2-SNPs, 3-SNPs, and 4-SNPs involving strains from the sextet (A1 A2 A7 A11 B2 D8), that occur at least 100 times, sorted by their frequency of occurrence.

| Strains | Number of occ. |
| --- | --- |
| A7 B2 | 1227 |
| A11 D8 | 306 |
| A2 D8 | 291 |
| A11 A2 | 284 |
| A1 D8 | 214 |
| A1 A2 | 196 |
| A1 A11 | 194 |
| A1 A7 B2 | 389 |
| A7 B2 D8 | 303 |
| A11 A7 B2 | 265 |
| A2 A7 B2 | 208 |
| A11 A2 D8 | 179 |
| A1 A11 D8 | 172 |
| A1 A11 A2 | 161 |
| A1 A2 D8 | 153 |
| A1 A11 A7 B2 | 265 |
| A1 A11 A2 D8 | 248 |
| A11 A7 B2 D8 | 232 |
| A1 A7 B2 D8 | 226 |
| A1 A2 A7 B2 | 139 |
| A2 A7 B2 D8 | 136 |
| A11 A2 A7 B2 | 110 |

Finally, the fact that the alignment of this sextet is a mixture of segments that follow different phylogenies is also confirmed by the overall relative frequencies of the SNP-types that are observed for these strains. *Appendix 5—table 3* shows the most common 2-SNPs, 3-SNPs, and 4-SNPs together with their number of occurrences, for this sextet of strains. Note that these are SNP-types obtained from the entire alignment. That is, a 4-SNP (A1 A11 A7 B2) means that these strains share a nucleotide that differs from the nucleotide that all other 88 strains have.

*Appendix 5—table 3* shows that, apart from the clonal pair (A7 B2), which is consistently supported by the observed SNPs, the others strains occur at similar frequencies in different mutually inconsistent combinations. For example, the four most common 3-SNPs all consist of the pair (A7 B2) with a different third strain. This diversity of phylogenetic partners for each strain is well summarized by the entropy profiles of these six strains, as shown in *Figure 4*.

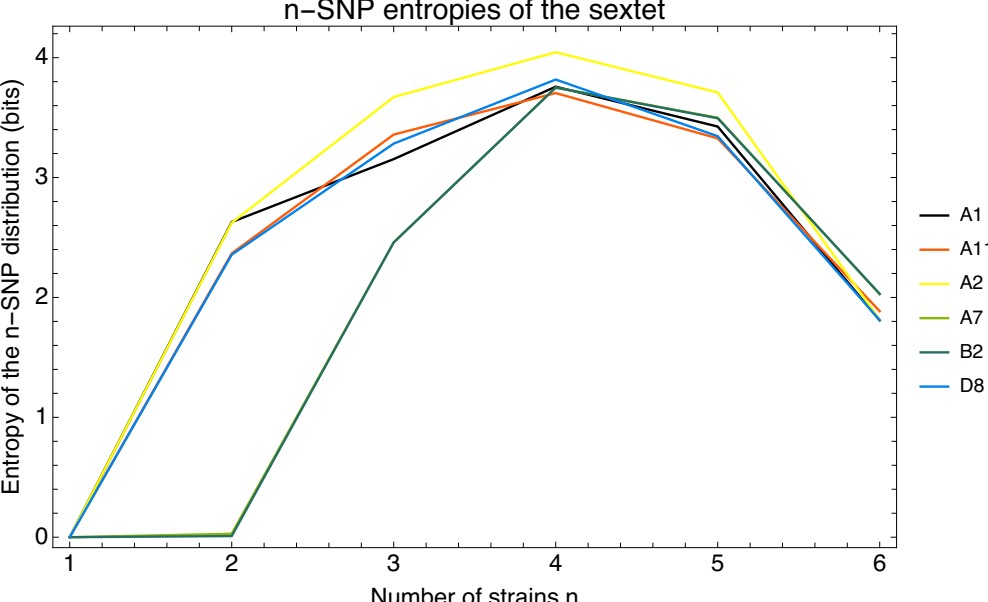

**Appendix 5—figure 4.** Entropy profiles for the six strains (A1, A2, A7, A11, B2, D8).

First, note that the strains of the pair (A7 B2) have virtually identical entropy profiles, indicating that they had a clonal ancestor before much recombination took place, that is their entropy profiles correspond to the entropy profile of their common ancestor. They also have zero pair entropy because they only occur in pairs with each other. However, all other strains have pair entropies around 2.5, which is equivalent to occurring with equal frequency in $5 - 6$ different pairs. For quartets, the entropies range from 3.6 to 4, equivalent to $12 - 16$ equally frequent quartets. That is, there is a diverse collection of phylogenetic relationships in which these six strains occur.

## Summary: the phylogeny of strains (A1,A2,A7,A11,B2,D8)

In summary, all statistics show that even for the relative close sextet of strains (A1, A2, A7, A11, B2, D8), the only clear clonal signal left is the close pair (A7 B2). The alignments between all other pairs have been mostly overwritten by recombination, the phylogeny changes thousands of times along the core genome alignment, and each strain occurs in a diverse collection of multiply inconsistent *n*-SNPs with the other strains. However, as shown in the bottom panel of *Figure 1* of the main text, the branches in the core tree of this sextet are well supported when a tree is constructed from sufficiently many loci. That is, the core genome phylogeny is just the best compromise capturing the statistics with which different phylogenetic patterns appear, and one reliably converges to this best compromise when sufficiently many loci are taken into account.

