## [Decision Letter]

**Acceptance summary:**

This work offers a careful, quantitative analysis of a population of *E. coli* genomes sampled at the same time from one location. Combining genomic analysis, with population genetics models and very extensive simulations to test different evolutionary scenarios, the authors have shown that the observed genetic diversity is inconsistent with both pure clonal evolution and a freely recombining population. Instead, population structure which leads to a broad range of recombination rates within a species must be considered. This careful study, while it does not offer a unique solution, is an important starting point to a broader discussion, both at the level of population genetics modelling and the level of the biological and ecological sources of the population structure.

**Decision letter after peer review:**

Thank you for submitting your work entitled "Whole genome phylogenies reflect the distributions of recombination rates for many bacterial species" for consideration by *eLife*. Your article has been evaluated by Aleksandra Walczak (Senior Editor) and a Reviewing Editor.

This work offers a careful, quantitative analysis of a population of *E. coli* genomes sampled at the same time from one location. Combining genomic analysis, with population genetics models and very extensive simulations to test different evolutionary scenarios, the authors have shown that the observed genetic diversity is inconsistent with both pure clonal evolution and a freely recombining population. Instead, population structure which leads to a broad range of recombination rates within a species must be considered. This careful study, while it does not offer a unique solution, is an important starting point to a broader discussion, both at the level of population genetics modelling and the level of the biological and ecological sources of the population structure.

Both reviewers had access to the multiple rounds of reviews from a previous journal that the authors provided with their submission, and their responses to the concerns raised in these previous reviews. These concerns especially addressed some of the technical aspects of the work, and the reviewers at *eLife* felt that your responses were not only adequate, but also that the additional analyses further elevated the work.

Taking this into account, the reviewers agreed that the work should be published largely as is, with just a few clarifications, noted below:

1) A recent study by J. Power et al. (https://doi.org/10.1101/2020.04.23.057174) has shown prevalence of strong selection on horizontal gene transfer in *B. subtilis* in an experimental evolution setup in the lab. I understand that population diversity in the work of Power et al. arises from a clone in the lab and the time-scale of evolution is substantially different from what you are considering in this manuscript. Nonetheless, I was wondering if you could comment how (if at all) selection on horizontal gene transfer (or recombination) could impact some of the results and the interpretations presented in the manuscript; a few sentences in the Discussion section would be sufficient.

2) Figure 11B very interesting. Is there a biological explanation (from the literature) as why H. Pylori has such high entropy level and a rapid saturation at n~5?

3) Please thoroughly address reviewer #1 comments on language edits.

Reviewer #1:

I suggest that you carefully go through the text and make language edits throughout. For example, there are issues with the usage of past and present tense and sometimes the narrative switched between the two even within a single paragraph. Also, there are typos and issues with the usage of articles and commas, which at times cause confusion.

---

## [Author Response]

Both reviewers had access to the multiple rounds of reviews from a previous journal that the authors provided with their submission, and their responses to the concerns raised in these previous reviews. These concerns especially addressed some of the technical aspects of the work, and the reviewers at eLife felt that your responses were not only adequate, but also that the additional analyses further elevated the work.Taking this into account, the reviewers agreed that the work should be published largely as is, with just a few clarifications, noted below:1) A recent study by J. Power et al. (https://doi.org/10.1101/2020.04.23.057174) has shown prevalence of strong selection on horizontal gene transfer in *B. subtilis* in an experimental evolution setup in the lab. I understand that population diversity in the work of Power et al. arises from a clone in the lab and the time-scale of evolution is substantially different from what you are considering in this manuscript. Nonetheless, I was wondering if you could comment how (if at all) selection on horizontal gene transfer (or recombination) could impact some of the results and the interpretations presented in the manuscript; a few sentences in the Discussion section would be sufficient.

Just as there is no doubt that selection acts on mutations, there is of course also no doubt that selection acts on recombination events. As we already noted in the Discussion, it is likely that natural selection plays an important role in determining the relative rates of recombination that we observe in the data. To what extent natural selection versus other mechanisms such as geographic population structure determine the relative rates of observed recombinations is extremely interesting, but beyond the scope of this work. Here we are just trying to provide quantitative estimates for how much recombination occurred in the history of a set of strains and the statistics of which strains have recombined more or less frequently. These results are not affected by the extent to which selection has shaped the observed patterns. We’ve extended our remarks in the Discussion and added a reference to the preprint that the reviewer mentioned.

2) Figure 11B very interesting. Is there a biological explanation (from the literature) as why H. Pylori has such high entropy level and a rapid saturation at n~5?

Previous studies have noted that recombination rates are very high in *H. pylori* and suggested that this species is’ freely recombining’ (Suerbaum et al., PNAS 1998). That is, *H. pylori* is the one species for which the consensus in the literature already is that its evolution is pretty much dominated by recombination. In addition, it has also been argued that population structure evident in *H. pylori* sequences reflects the population structure of its human host (Falush et al., 2003). We have added some remarks in the text, including citations to these papers, to make clear that the high amount of recombination that we infer for *H. pylori* is consistent with the literature.

3) Please thoroughly address reviewer #1 comments on language edits.

We have followed the reviewer’s advice and gone through the text in search for typos, superfluous commas, and unwarranted switches between tenses. However, we feel that in many places a mixture of tenses is appropriate, e.g. we say that we *found* that *n*-SNP distribution *have* long tails, because while this finding occurred in the past, the distributions did not cease to have long tails.

[Editors' note: we include below the reviews that the authors received from another journal, along with the authors’ responses.]

Reviewer 11.1) This manuscript tackles an interesting question and tries some new and worthwhile approaches. It brings a commendable amount of fresh energy to the problem. However makes many provocative claims in the Abstract that are not justified by the analysis performed. Furthermore, several of the sections have conceptual problems and are unclear. In general the authors rely far too much on verbal argument and intuition about what the quantities they calculate mean. There is a total absence of testing to show that the methods are actually measuring what is claimed. If published in anything like its current form, it would be an extremely confusing addition to the literature.

After damning us with faint praise, the reviewer starts the review by making a number of strong and very critical claims, including:

1) That claims in the Abstract are not justified by our analysis.

2) That we ‘rely on verbal arguments’, provide no quantitative support, nor any testing, to support that the quantities we calculate mean what we claim.

Since the reviewer makes these claims without identifying any concrete issue, method, or statement here, let alone providing an argument for these criticisms, we can at this point only note that we strongly disagree with these claims.

As detailed below, we have performed many additional analyses and validations to comprehensively address every concrete issue raised by the reviewer. These analyses include extensive comparisons with data from simulations of populations evolving under drift, mutation, and recombination (for which we specifically developed new simulation code). All these analyses confirm the accuracy of our methods. Moreover, although some valid technical issues were raised, we also show that quite a number of the critical claims made by the reviewer do not hold up to careful scrutiny, e.g. see responses to points 1.3, 1.4, 1.8, 1.9, 1.10, and 1.12.

In contrast, we do accept the reviewer’s criticism that several conceptual issues were not sufficiently clearly explained and we have now significantly rewritten and extended discussions in the manuscript to make these points clear. Here too we have taken advantage of the simulation results, where the ground truth is known, to better explain the meaning of some of the quantities we calculate. In addition, we introduced a new section in which we illustrate conceptual issues regarding the interpretation of the core genome phylogeny by using data from human genomes, and we also expanded the section on the entropy profiles to better explain what the *n*-SNP statistics show about the relative recombination rates among different strains. Finally, to further clarify the meaning of the results we added analyses on the phylogroups(Appendix 1—table 2, and Figure 10—figure supplement 1), including an in-depth analysis of the SNP patterns for phylogroup B2.

1.2) As the authors remark, the phylogeny section (Figure 1) is not hugely new and the authors provide no way to quantify recombination patterns from degree of incongruence.

The main aim of the analysis of Figure 1 is to introduce the general problem that the paper addresses using well-known previously used methods, i.e. that phylogenies reconstructed from individual 3Kb alignment blocks are incongruent, but that phylogenies reconstructed from a large number of such blocks converge to a robust phylogenetic tree.

We agree that these incongruence measures do not directly quantify recombination patterns. Indeed, this is precisely the problem that our manuscript addresses and much of the rest of the manuscript is exactly about quantifying recombination patterns.

However, it should be noted that the results of Figure 1 do already rule out several suppositions that are sometimes made in the literature. First, if there were a significant fraction of blocks that were unaffected by recombination, than all these blocks would have to be consistent with a common phylogeny. The fact that every 3Kb block significantly rejects the phylogeny of every other block shows that this cannot be the case, i.e. almost every block has to be affected by recombination to some extent. Second, it is also often presumed that, while phylogenies may vary along the genome, that deviations from the core tree reconstructed from all blocks are only minor. Figure 1 also shows that this is not the case but that the incongruences are very substantial, i.e. half of the branches of the core tree occur in less than a quarter of all blocks.

In the revision we have rephrased so as to stress these points.

1.3) The pair of strains analysis (Figure 2) assumes that the simple statistical model that the authors use is fully accurate in identifying recombined regions. The method that the authors used is reasonable (others have used similar models) but is sure to make errors. Moreover, these errors are unlikely to be random within the dataset, since distinguishing recombined regions from non-recombined ones becomes harder as divergence between strains increases. This bias is likely to interfere materially with the arguments the authors want to make.

It is not true that the results in Figure 2 assume that the identification of recombined regions is ‘fully accurate’. We state very clearly that this method is used to estimate the recombined regions for each pair. We also completely agree with the reviewer that it becomes harder to estimate the fraction of clonally inherited regions as divergence increases. Indeed, one purpose of panels A, B, and C of Figure 2 was to show readers that the rare recombined regions in close pairs can almost be identified by eye whereas, as recombined regions get denser on the genome, it becomes harder to distinguish them from the clonally inherited regions.

In contrast to what the reviewer implies, the conclusions that we draw from Figure 2 are very robust to inaccuracies in these estimates. We are noting broad quantitative trends that are observed reproducibly across many independently analyzed pairs of strains. For example, *all* close pairs with divergence less than 0.001 have more than 90% clonally inherited regions and *all* pairs with divergence over 0.01 have a very low fraction of clonally inherited regions. In addition, in the transition regime of divergences between 0.001 and 0.01, the estimated fraction of clonally inherited regions drops systematically with divergence. That is, even though each pair is analyzed independently, the fraction of clonally inherited regions decreases in a very systematic and reproducible way as a function of the divergence between the strains. Our conclusions only depend on this very reproducible trend seen across all pairs of strains. We find it hard to imagine how inaccuracy in these estimates could artificially produce such a clear systematic dependence on divergence.

Nonetheless, to remove any doubts about the accuracy of this procedure, we now additionally apply the same pairwise analysis to data from simulations for which the ground truth is known. We show that the results of the pair statistics accurately reflect the known ground truth in those simulations (Figure 1—figure supplement 3 and Figure 2—figure supplement 2). In addition, in Figure 2—figure supplement 2 we directly show that the estimated fractions of clonally inherited loci are always very close to the true fractions of clonally inherited loci. This analysis shows that our method is in fact conservative in that it tends to slightly overestimate the fraction of clonally inherited loci.

1.4) Further, large recombination events are easier to detect than small ones. Also, multiple recombination events can easily be called as one, whether or not they actually overlap. This makes it very likely that the median size of recombination events is overestimated, probably substantially.

We agree that overlapping recombination events can lead to overestimation of the size of recombination events. Precisely for this reason, we estimated the sizes of recombination events using only comparison of very close strains that have only a handful of recombination events between them, i.e. we stated:”Using a Hidden Markov model on close pairs, we also estimated the distribution of lengths of recombined regions (see Materials and methods),” and in the Materials and methods it says:”To estimate the lengths of recombination events, we first extracted pairs that are sufficiently close (divergence less than 0.002) such that multiple overlapping recombination events are unlikely.” It appears the reviewer failed to note this. Nonetheless, to further confirm that our approach accurately estimates the size of recombination events, we applied the same approach to data from simulations for the revision. As shown in Figure 2—figure supplement 2, the sizes estimated from the simulated data are in fact very close to the true size of the recombined fragments used in the simulation.

Note also that, even if we did overestimate the size of the recombination events, it would leave virtually all the main conclusions of the paper unaffected, because almost none of our conclusions strongly depend it.

1.5) The authors nowhere acknowledge that the recombination detection method is less than perfect, attempt to quantify its errors or allow for those errors in interpreting the patterns they deduce from it.

It is not entirely clear which of our methods the reviewer refers to here. If this still refers to the pairwise analysis, then this comment repeats comment 1.3. We thus suspect that the reviewer must be referring to the method explained in Figure 3 and used in the results of Figures 4, 5, and 6.

First, as we pointed out in the paper, this use of bi-allelic SNPs to detect phylogenetic inconsistencies is in fact commonly used in the literature on sexually reproducing species. Second, it is not true that we pretended this method is ‘perfect’. We explicitly discuss that the method assumes that bi-allelic SNPs predominantly correspond to single substitution events, i.e. that homoplasies are rare. Indeed, to support this assumption, we included calculations in the Materials and methods to estimate the fraction of homoplasies and showed that those are very rare. Moreover, in the application in Figure 6 we explicitly discussed that the method gives only a lower bound on the number of recombination events.

However, we agree that we did not explicitly try to quantify the effects of a small fraction of homoplasies (or other errors) since we thought it was obvious that our interpretation of the results in Figures 4, 5, and 6 could not possibly be substantially affected by them. From the response of this reviewer and reviewer 2, we now see that this was apparently not obvious and that we *should* have done an explicit quantification. Thus, for the revision we have performed several new analyses to explicitly quantify the accuracy of this method:

1) Following questions from reviewer 2 we now quantify the rate of sequencing errors and show these are so rare that they cannot possible affect any of these analyses.

2) We have extended our model for estimating the homoplasy rate by including a 3-to-1 transition to transversion ratio in mutations (which raises the estimated homoplasy rate from 2.5% to 3.3%). In addition, we confirm the accuracy of the estimation of the homoplasy rate using data from simulations for which the exact homoplasy rate is known (Materials and methods).

3) We introduce a new method for correcting for homoplasies. In particular, we first quantify for each SNP column in the core genome to what extent it clashes with other SNP columns in its vicinity, and then remove the most inconsistent columns from the alignment. By recalculating the effect on all of our statistics (i.e. those of Figures 4, 5, and 6) of removing either 5% or 10% of potentially homoplasic columns, we show that our results are indeed not substantially affected by homoplasies (see Figure 4—figure supplement 2, Figure 5—figure supplement and Figure 6—figure supplement 1).

4) We also calculate all the relevant statistics from Figures 4, 5 and 6 for the data from the simulations and we do this both for the original data and for data from which 5% or 10% of potentially homoplasic columns have been removed. Comparisons with the known ground truth allow us to assess the accuracy and precise meaning of our statistics (see Figure 4—figure supplement 3, Figure 5—figure supplement 2, Figure 6—figure supplement 2, Figure 6—figure supplement 3 and Figure 6—figure supplement 4).

These very substantial additional analyses and validations all confirm the validity of our methods and results.

1.6) The LD plot is not new nor (more importantly) particularly well integrated with the rest of the manuscript.

We did not claim this analysis is new. On the contrary, we explicitly stated that this is a standard LD measure. We included it because this analysis has been used in other studies, e.g the recent analysis of thermophilic Cyanobacteria from Yellowstone, which we specifically refer to.

1.7) The section on the lower bound of recombination to substitution events fails to note that a single recombination event will introduce changes in phylogeny at both ends of the event. Therefore, even if the rest of the reasoning is correct, the estimate would be off by a factor of 2.

This is correct and we thank the reviewer for pointing this out. *C/M* is a lower bound on the number of phylogeny changes per SNP, not recombinations. Knowing that this lower bound likely strongly underestimates the true number, we sloppily failed to properly distinguish between phylogeny changes and recombination events. In the revision we now explicitly compare the lower bound *C/M* of phylogeny changes per SNP obtained for simulation data, with the ratio *ρ/µ* of recombination to mutation rate used in the simulations. This analysis shows that, at very low recombination rates, the ratio *C/M* is exactly twice *ρ/µ*, precisely for the reason that the reviewer mentions. Thus, for very low recombination rates, the ratio *C/*(2*M*) accurately estimates the recombination to mutation rate (Figure 6—figure supplement 1). At such low ratios *ρ/µ*, there are many SNPs columns that separate consecutive edges of recombination events on the alignment, so that each such edge leads to detectable phylogeny clashes. However, already when *ρ/µ* ≥ 0.05 or so (Figure 6—figure supplement 2), a substantial fraction of recombination events goes undetected because they do not result in phylogeny clashes, so that *C/M* is in fact substantially lower than *ρ/µ*. In the regime of the *E. coli* data, i.e. *C/M* ≈ 0.13, the ratio *C/M* likely underestimates the ratio of recombination to mutation events by more than a factor ten.

In the revision we have now paid attention to clearly distinguish the lower bound on the number of phylogeny changes *C*, from a lower bound on the number of recombination events, which is *C/*2. However, as just noted, this factor 2 correction is small compared to the factor by which the ratio *C/M* underestimates the true ratio of recombination to mutation.

1.8) The authors have also made no effort to quantify how large the effect of homoplasy is on these results. It may be larger than they imagine, since each homoplasy can potentially introduce two phylogeny changes. My intuition is that only a relatively small proportion of SNPs need to be affected to have quite substantial effects. Of course, this is not tested in any way.

As discussed already above, we have now explicitly tested this and show that, even using a very conservative procedure in which the most clashing columns are removed, and a larger fraction of columns is removed than the estimated homoplasy rate, the estimate of *C/M* only decreases by about 16% from 0.155 to 0.13 (see Figure 6—figure supplement 1).

1.9) Even if the estimate for C is accepted, the estimate for the number of times that each site has recombined is questionable for a few reasons. The estimate of average recombination size of 20kb is totally inappropriate to use for this purpose. As discussed above it is sure to be upwardly biased. We know in any case that there are multiple mechanisms of bacterial recombination and that they have very different length sizes. Long ones are disproportionately important in break up the clonal frame, while short ones break up the phylogeny. Dividing C by M effectively assumes, I think, that all recombinations are the mean estimated length.

We would like to again point out that the method for providing a lower bound *C* is a very standard method in the field of sexually reproducing species.

Regarding our order of magnitude estimate for the average number of times alignment positions have been overwritten by recombination, we do not make the assumption that all segments must have the same length. The argument is very simple and just based on the estimated average length *L_r_*of the recombined segments. We tried to clarify this in the revision.

Second, as we explained above, the reviewer’s claim that our estimate of the mean size of recombination segments is biased upward is based on a misunderstanding of how this estimate was done (and we now provide results from analysis of simulations to validate the accuracy of our estimation method).

Third, in the revision we use data from simulations to show that, even if we use an average length of recombined segments approximately half of what we estimate from the *E. coli* data, already at recombination to mutation rates as low as *ρ/µ* = 0.01, positions in the genome have been overwritten on average more than 12 times, and at *ρ/µ* = 1, positions have been overwritten on average more than 1500 times (Figure 6—figure supplement 3). In addition, using the simulation data we also show that our simple order of magnitude estimate *CL_r_/*(2*L*) in fact severely *underestimates* the true average number of times positions in the genome have been overwritten by recombination (Figure 6—figure supplement 4).

But independently of all this, we make very clear in the manuscript that this is only an order of magnitude estimate that we use to conclude that each position in the genome has been overwritten many times. The precise number is completely immaterial for the overall conclusions and message of the paper. Indeed, in the revision the number has been changed from 300 to 125 due to the factor 2 we overlooked (see response 1.7) and because we now use the 5% homoplasy corrected alignments. But, as suggested by the analysis of the simulated data (Figure 6—figure supplement 4), this estimate may well be a factor 10 below the true value. That is, it is clear that we are very far from having any loci in the genome that are not affected by recombination, and this is all that matters for our conclusions.

1.10) Furthermore, even if the estimate of C/M is taken at face value, it is difficult to interpret. There are 87 genomes in the sample, so it implies that each strain has recombined about 4 times on average at each site since the shared common ancestor. But the largest number of recombination events per site per strain that their method can detect is 1. So all that this “order of magnitude” estimate implies, I think, is that assumptions of the inference of C/M are wrong in some way, as I would expect based on the arguments above.

We frankly cannot follow the reasoning in this comment. There are 92 instead of 87 genomes in our sample, but this is a detail. Mutations and recombinations are not events that happen to ‘strains’, they are events that happen in the evolutionary history of a set of strains. Each mutation at a genomic position is an event that happened on a branch of the phylogenetic tree at that position. For recombinations the situation is even more complex, i.e to precisely describe the recombination events in the history of a set of strains one needs the more complex structure of a recombination graph.

The *C/M* value provides a lower bound on the ratio of the number of times the phylogeny changes along the alignment and the number of mutations along the alignment. Thus if the rate of SNPs is about 1*/*10 and *C/M* ≈ 1*/*7, it means that once every 10 alignment columns, a substitution occurred along some branch of the phylogeny at that position, and once every 7 SNPs (or 70 bps) a phylogeny change occurred, i.e. either the start or end of a recombination event that occurred somewhere in the evolutionary history of the strains.

We do agree that, per definition *C/M* cannot be larger than 1, and so in the limit that recombination rates are higher than mutation rates, i.e. *ρ/µ >* 1, the lower bound *C/*2 will hugely underestimate the true number of recombination events (see Figure 6—figure supplement 2). This is precisely why this method can only give a lower bound.

In any case, the comparisons with ground truth in the simulations show that the estimate *C/*(2*M*) accurately estimates the ratio of recombination-to-mutation rate *ρ/µ* when this ratio is very low, and gives a valid lower bound when it is large. Similarly, we also show that the estimate *L_r_C/*(2*L*) is a valid lower bound for the average number of times each position in the alignment was overwritten by recombination. These additional comparisons with data from simulations should remove any questions about the meaning of these estimates and confirms their validity.

1.11) In any case, the authors provide little guidance on what they think the value actually means. How much of the recombination do the authors think has taken place since the evolution of the phylogroups, for example? I have no idea.

Although we tried to be already quite specific in the original submission, for the revision we have made sure to spell out precisely in the text what the various lower bounds mean. In particular,

*C* gives a lower bound on the number of phylogeny changes in an alignment.*C/*2 gives a lower bound on the number of recombination events in an alignment.*C/M* gives a lower bound on the ratio of phylogeny changes to substitutions in the alignment.*C/*(2*M*) gives a lower bound on the number of recombination events per substitution in the alignment.*T*_est_ = *L_r_C/*(2*L*) gives a lower bound on the average number of times each position in the genome has been overwritten by recombination.

Note that we confirm the validity of all these lower bounds using data from simulations.

At the request of the reviewer, we have now also calculated these quantities for each of the phylogroups represented in our dataset (Appendix 1—table 2). In addition, we have added a supplementary analysis where we look in great detail at all statistics for the phylogroup B2, which is represented by 6 strains in our collection.

1.12) The n-SNP section is interesting, particular for 2-SNPs. I have not seen this in bacterial genomics and believe it can potentially be very informative. However, this section, based on weird reasoning, completely ignores the impact of clonal relationships. Pairs of strains that share a recent common ancestor will share many 2-SNPs. They have not undergone large amounts of recombination. This is obviously likely to account for many of the extremely high 2-SNP values. The so called “scale free” distribution, which I do not think is terribly helpful in any case – parameter estimation would be a better idea – is very dependent on these high values. Note also that individual large recombination events can give high 2-SNP values between pairs of less closely strains. The authors also do not discuss this at all.

The claims of the reviewer, i.e. that the pairs with highest 2-SNP counts are ‘obviously likely’ due to clonal pairs, and that the long-tailed distribution of 2-SNP counts is ‘very dependent’ on these clonal pairs, are both incorrect. One only needs to carefully look at Figure 9 to see that the first claim cannot be maintained. In Figure 9A strain A1 is shown to share the highest number of 2-SNPs with strains D8 (214 2-SNPS), A2 (196 2-SNPs) and A11 (194 2-SNPs). Figure 9C shows that these pairs are among the top 10 highest of all 2-SNP counts. However, only one of these three strains can share a most recent clonal ancestor with A1 so at least 2 of these three pairs do not correspond to clonal pairs.

In order to determine the exact effect of 2-SNPs from clonal pairs on the long-tailed distributions in Figure 9C we would of course have to know which pairs correspond to clonal pairs, and this is not always obvious, as the example in Figure 9A demonstrates. However, to make a conservative estimate of how much removal of clonal SNPs *could* have on Figures 9C and 9D we proceeded as follows: We first assumed that all SNPs that correspond to bi-partitions that occur in the clonal tree are clonal SNPs. We then removed all these potentially clonal SNPs and remade Figures 9C and D without the n-SNPs corresponding to clonal SNPs. As show in Figure 9—figure supplement 1, removal of these potentially clonal n-SNPs has very little effect on the distributions of n-SNPs.

1.13) Furthermore, there is an apparent confusion between lineages having high average rates of recombination, i.e. measured relative to time or to rates of mutation and lineages being very non-random in their patterns of recombination, i.e. which strains they recombine with. What the authors are actually measuring is much more related to the latter. These results gives no indication that strains differ in their absolute recombination rates, but this is the impression given, including in the headline conclusion that is made prominently in the Abstract. It’s really extremely unclear what the authors actually think they have shown.

As we already mentioned in our response 1.10, recombinations are not events that happen to strains. It therefore makes fundamentally no sense to talk about the recombination rate ‘of a strain’. We were in fact careful to talk in the Abstract about the ‘recombination rates between different *lineages*’ (emphasis added). Moreover, the statement in the Abstract is about the amount of *variation* in the recombination rates across different lineages and this is thus per definition a statement about the relative sizes of rates.

We thought that it was clear that, while the analysis in Figures 5 and 6 quantifies the absolute amount of recombination within alignments of subsets of strains, Figure 9 shows that the *relative* rates of recombination between different lineages vary along a long-tailed continuum that spans several orders of magnitude. To avoid any confusion, we now explicitly emphasize this point in the description of these results.

1.14) In the discussion the authors conclude (which is not novel), that the ancestral phylogeny is very difficult, or impossible to reconstruct. This is true for most bacterial species, but this is typically because there are branches at the base of the tree that are extremely uncertain. It is still likely possible to infer a great deal about bacterial evolution from inferring clonal relationships closer to the tips. For most species, this includes a fairly substantial proportion of the clonal tree. While there are the kernel of some good ideas presented, I am not sure, once the substantial issues underlined above are dealt with, that this manuscript ultimately changes our view of this in a substantive way.

It seems that this final comment is not so much a specific criticism of any of our analyses, but more an expression of the reviewer’s own views regarding the role of recombination in bacterial genome evolution. We understand the results of our analyses are at least partially at odds with these views, but we hope the reviewer can agree that this is not in itself a valid reason to reject our work.

Also, there is by no means consensus in the field on these points. For example, while this reviewer claims that it is ‘not novel’ to conclude that the ancestral phylogeny cannot be reconstructed, reviewer 2 seems to be of the opinion that it can. In fact, since reviewer 2 identified himself, we know that this reviewer is co-author on a well-known paper on *E. coli* genome evolution that makes this claim. A main aim of our manuscript is to provide new methods that can help clarify these issues.

We are of course aware that our results challenge views held by many in the field of bacterial genome evolution, and because of this we not only spend great time and effort to develop our analysis methods, extensively testing and redesigning them over more than 6 years, but we also made sure to draw conclusions based not on the results of one single method, but on the very consistent quantitative picture that emerges from multiple different and complementary methods. Thus, even if there were valid technical issues with some of our methods, we find it very hard to see how several different and complementary methods can all paint the same quantitative picture that is wrong. Unfortunately, it seems that the reviewer has completely ignored this point.

Nonetheless, for the revision we have now addressed *every single one* of the technical issues that were raised in the review, providing extensive additional analyses and validations to confirm that our methods were in fact all valid and support the conclusions we draw. We hope the reviewer can appreciate this and will agree with us that, having addressed all issues that were raised, there are no good reasons left to obstruct publication of this work, and allowing other researchers to learn about our analyses.

Reviewer 2 - Olivier Tenaillon:

Reviewer 2’s review was long and complex and there were multiple cases where the same or highly-related issues came up in different contexts. Therefore, instead of going through all comments one by one, we have attempted to bring some structure in both the issues raised and all our responses to these issues.

To provide structure to the reviewer’s comments and our responses, we start from points (i)-(vi) of the review, which provide a fairly accurate summary of the key points of the paper (maybe with exception of the interpretation of the results of Figures 7 and 9). To anticipate some of the responses to specific comments of the reviewer, we’ve added short comments to each of these points.

• Ad (i): phylogenies constructed from blocks of 3kb are distinct from whole-genome phylogeny (”core tree”), and distinct across blocks. However, phylogenies of 50% of the 3kb blocks are similar to the core tree.

Note that we also show that the block phylogenies are not just minor variants but that they substantially differ from the core tree. that is, we do not just do a same/different but look individually at each branch of the core tree. Half of the branches in the core tree are found in less than 25% of the block phylogenies and only a small fraction of the branches is supported by the large majority of the blocks.

• Ad (ii):when comparing pairs of *E. coli* genomes, SNP density is either low (interpreted as clonal inheritance of mutations) or high (interpreted as recombination). The fraction of recombination SNPs (high density regions) increases with divergence between strains. Half of the genome is recombined at a low divergence of 0.0032.

Note also that. even for close pairs, most SNPs derive from recombination. For example, for the green pair in Figure 2, at divergence 0.002, even though almost 90% of the alignment is clonally inherited, 90% of the SNPs lie in the recombined regions. That is, there are virtually no pairs where their divergence reflects the distance to the clonal ancestor of the pair and most of the divergence derives from recombinations.

• Ad (iii): only about half of the individual SNPs support the core tree

Although this is correct, the total fraction of SNPs consistent with the core tree is not very informative because a significant fraction of all SNPs correspond to a single branch (i.e. 36% of all SNPs are of the type where the outgroup has one allele and all other strains another allele). To get a more informative picture of the SNP support for the core tree we analyzed the SNP support separately for each branch of the core tree. We find that for half of the branches in the tree, the branch is rejected by 20 times as many SNPs as support it, and this applies to virtually all branches deeper in the tree (with the exception of the branch to the outgroup). The only branches that do not have a high fraction of rejecting SNPs are branches to clades of strains that are all extremely close to each other.

• Ad (iv): phylogeny changes every ∼ 10 bp along the genome, implying a ratio of recombination to mutation events of at least 0.15

Note that we calculated the ratio of phylogeny changes to substitution events (C/M) not only for the full alignment, but across a very large number of random subsets of strains (Figure 6). The results show that, the smaller the subset of strains, the more variable the C/M ratio, suggesting recombination rates vary across lineages. However, as soon as one looks at groups of 10 or more strains, the C/M ratio is never small (i.e. larger than 0.05 meaning a recombination occurs every 20 SNPs in the alignment of the subset).

• Ad (v): For a focal strain, they identify all strains that share a SNP with the focal strain and none of the others. The frequency distribution of the number of 2-SNPs (SNPs present in exactly 2 strains) shared by a focal strain with each other strain is scale free. This is taken as evidence that there is no group of highly recombining clades.

This is the one point where the reviewer’s summary does not quite match what we were intending to show. The n-SNP patterns were used for two complementary analyses. When we plot the distribution of 2-SNPs (Figure 9C), we are not looking at a focal strain but put all 2-SNPs are put’ in a pot’ and we simply plot their overall frequency distribution. That is, we are plotting the distribution of the thickness of all lines in Figure 9B (and similarly for 3-SNPs, 4-SNPs, etc). We interpret the fact that all these distributions are smooth and approximately scale-free as showing that the strains cannot be easily clustered into groups that have high within-group and low between-group recombination rates.

The second analysis, which does use focal strains, calculates entropy profiles. Whenever a focal strain occurs in different *n*-SNP types, all these *n*-SNP types are mutually inconsistent, so that each *n*-SNP type must correspond to a different phylogeny. For each focal strain and *n*, we can summarize the diversity of mutually exclusive *n*-SNP types by the entropy of *n*-SNP type frequencies (Materials and methods). This gives the entropy profiles of Figure 10 of the revision. The main result of this analysis is that, especially when *n* gets larger, all strains show high entropy indicating that they occur in a diverse set of phylogenies in which they belong to different subgroups of strains. In addition, the fact that the entropy profiles look different for most strains shows that the *n*-SNP distributions are distinct across strains.

**The main criticism separated into 2 parts**

The main criticism is that we have not provided sufficient validation for our two claims that 1. the structure of the phylogeny is generated by recombination patterns (as opposed to clonal ancestry) and that 2. there is currently no known model that can account for the patterns we observe. I would like to start by stressing that we are not contending that there is an unambiguous phylogenetic structure in the sense that when one reconstructs a phylogeny from sufficiently many genomic loci, one tends to obtain very similar phylogenies. Indeed we explicitly show this in Figure 1. The question is what this phylogenetic structure reflects. Second, for the purpose of the discussion, it is useful to separate our claims into 2 parts:

1) We show, using several complementary methods, that recombination is so dominant over clonal mutations that the clonal phylogeny cannot be reconstructed and it is unlikely that the core genome tree corresponds to the clonal phylogeny. Moreover, even if we could reconstruct the clonal phylogeny, there is such a large number of different phylogenies along the alignment that the observed mutational patterns cannot be meaningfully captured by a single phylogeny.

2) If the core genome phylogeny doesn’t reflect clonal structure, how does one explain there is a stable phylogeny? We argue that, since all genomic loci have been affected many times by recombination and thousands of different phylogenies occur along the alignment, the overall phylogenetic structure reflects the statistics of the *distribution* of phylogenies that occur along the alignment. We propose that these statistics reflect the relative frequencies with which recombination has occurred among different lineages. To support this we showed that, indeed, the relative frequencies with which different subsets of strains share SNPs varies over a wide continuum spanning several orders of magnitude and that almost every strain has its own distinct statistics of SNP sharing with other strains. No population genetics model that we are aware of can reproduce such statistics.

**Issues affecting part 1 of our claims.**

Let’s first discuss part 1 of our claims. The main criticism of our methods for quantifying recombination is that we fail to account for alternative explanations for phylogenetic inconsistencies and the following alternative explanations are put forward by the reviewer:

Tree inference errors.Sequencing errors.Homoplasies.

Let’s go through points (i)-(v) and examine how each of these 3 alternative explanations would affect the results we report.

**Tree inference errors**

Tree inference errors can only affect the analysis of (i) because none of the other results rely on tree inference. As an aside, the main point of (i) is not to present new results but to introduce the main problem that our study addresses using a well-established method (PhyML).

We of course agree that there can be tree inference errors but Figure 1 shows large statistical patterns of the phylogenies across all 3Kb blocks, i.e. the fraction of 3Kb block phylogenies that support each branch of the core tree. We find that the majority of the core tree branches occur only in a minority of the 3Kb block phylogenies. We cannot see how tree inference errors could substantially affect such broad statistics. Note also that each block significantly rejects the phylogeny of each other block (Figure 1—figure supplement 2), i.e the log-likelihood differences between different phylogenies on a given alignment block are never small. To argue that the apparent inconsistencies between the phylogenies of 3Kb blocks are artefacts of tree inference errors one would have to argue that the ML inference on alignment blocks is so error prone that even overall statistical patterns (i.e. fractions of blocks agreeing on the occurrence of particular bipartitions) cannot be trusted at all. This would essentially invalidate *any* analysis using ML phylogeny reconstruction and it seems unlikely to us that the reviewer really wants to claim that.

However, since Figure 1 is only used to introduce the problem we analyze, even if the reviewer wants to discount the results of Figure 1 altogether, this does not really affect any of the methods and conclusions we present.

**Sequencing errors**

We completely agree that we should have included more about the sequencing methods and the potential rate of sequencing errors. For the revision we have now included a section at the start of the methods describing the sequencing procedures and the core alignment construction using Realphy. Realphy is in fact very conservative in what alignment columns to include in the core genome alignment and we also show that, from the occurrence of multiple independent genomes that are *identical* in the core genome, we can bound the rate of sequencing errors to be less than 1.1 ∗ 10^−7^.

In the methods we now also provide a new section that shows, using a simple calculation, that sequencing errors can account for at most one in every hundred thousand informative SNPs, which obviously is not going to meaningfully affect any of the results.

**Homoplasies**

This leaves homoplasies. We agree with the reviewer that if the rate of homoplasy were very high (e.g. if the majority of SNPs were homoplasies), then this would be problematic. Indeed, to show that homoplasies must be rare we presented analysis in the methods to give a rough estimate of the rate of homoplasy. The most relevant estimate is for synonymous sites excluding the outgroup and gives that about 2.5% of all bi-allelic SNPs are homoplasies (i.e. more than 1 mutation occurred).

One criticism that the reviewer had is that we may be significantly underestimating the homoplasy rate because we ignore heterogeneity in mutation rates, in particular that transition mutations are more frequent than transversion mutations. For the revision we have now redone the analysis using a 3-to-1 transition to reversion ratio. We show that this only raises the estimated homoplasy rate from 2.5% to 3.3%, i.e. still well below 5%.

Note that, for the revision, we have developed new software for simulating populations evolving under drift, mutation, and recombination (of the type where a genomic segment from one individual is recombined in the genome of another) while at the same time keeping track of the clonal phylogeny, and the number of times each position on each branch of the clonal tree is affected by recombination. We then used this software to simulate populations evolving under a large range of recombination rates. Throughout the revision we use data from these simulations to validate our analysis methods and to compare results on *E. coli* with corresponding results from simulations for which the ground truth is known.

To confirm the accuracy of the estimation method for the rate of homoplasies, we used data from our simulations and compared the estimated rate of homoplasy with the true rate of homoplasy in the alignment, finding them to be virtually identical, i.e. 2.43% estimated versus 2.56% true rate.

We thought that it would be obvious that such a low rate of homoplasy, although of course affecting the precise values of the various statistics, could not meaningfully affect any of our conclusions. However, both this reviewer and reviewer 1 question this, suggesting that even a fairly low rate of homoplasy could have large effects on the results of the analysis. To dispel this concern we developed a method that estimates the maximal effect homoplasies can have on the results, by identifying those SNP columns in the alignment that lead to the largest number of clashes with other SNP columns, and removing either the 5% or 10% most clashing SNP columns. We then redid all the analyses with such ‘homoplasy-corrected’ alignments and included comparison of the original and homoplasy-corrected results in the revision. In addition, we also performed such comparisons for all the data from the simulations.

Note that our method is conservative because it does not just remove potentially homoplasic columns, but systematically removes those columns that lead to the largest number of inconsistencies. In addition, we remove up to 3-fold more columns than the estimated rate of homoplasies. We now discuss what this new analysis showed about the effect of homoplasies on each of our results.

Effect of homoplasies on the results (ii): The analysis in (ii) is not based on phylogenetic inconsistencies at all and just looks at the patterns of SNP density in pairwise alignments. Therefore, these analyses are not affected at all by homoplasies and neither are the conclusions from the pairwise analysis, i.e. that for the vast majority of pairs no clonally inherited DNA is left in their pairwise alignment, and that even for close pairs for which most genomic segments were clonally inherited, the large majority of their SNPs comes from recombined regions.

Note also that in the revision we extensively validate the accuracy of the pairwise analyses using data from simulations (Figure 1—figure supplement 3, Figure 2—figure supplement 1 and Figure 2—figure supplement 2).

Effect of homoplasies on the results (iii): Regarding (iii), our main observation in Figure 4 is that for half of the branches in the core tree, there are 20 times as many conflicting as supporting SNPs. As shown in Figure 4—figure supplement 2, these results are virtually unchanged for the homoplasy-corrected alignments for which 5% or 10% of the most clashing SNP columns have been removed.

To further explore the possible effect of homoplasies on this type of analyses, Figure 4—figure supplement 3 shows the same analysis on the simulated datasets. We see that homoplasies only significantly affect the results when there is no recombination at all. In this case all clashing SNPs result from homoplasies and the homoplasy removal has a substantial effect, i.e. only after 5% of homoplasies have been removed does the analysis confirm that there is essentially 100% SNP support for each branch.

The analysis of the simulation data also confirm what our pairwise analysis already indicated: that even in situations where most genomic loci are clonally inherited, the large majority of the SNPs can result from recombination events. In particular, for the low recombination rate of *ρ/µ* = 0.01, most genomic loci are clonally inherited along all but the long inner branches of the clonal tree (Figure 1—figure supplement 3). However, for most pairs of strains, including pairs for which almost all loci were clonally inherited, most of their divergence derives from recombination events (Figure 2—figure supplement 1), and this is confirmed by the analysis in Figure 4—figure supplement 3, which shows that for simulations with *ρ/µ* = 0.01, half of the branches have less than 20% support.

Effect of homoplasies on the results (iv): Homoplasies are most relevant for the analysis of (iv) and we agree with the reviewer that it is worthwhile to investigate how much homoplasies can affect the results shown in Figures 5 and 6. For the revision we have now explicitly done so by redoing all these analyses on the homoplasy-corrected alignments. In addition, to further validate these analyses, we have also performed all these analyses on the data from the simulations, including investigating the effect of removing 5% or 10% of potentially homoplasic sites from the alignments of the simulation data.

As shown in the new Figure 5—figure supplement 2 and Appendix 1—table 1, the distribution of the number of consecutive tree-compatible SNPs is only very modestly affected by the removal of potentially homoplasic SNPs, i.e. the average number of consecutive tree-compatible SNPs increases from 7.6 to 8.8 when 5% of potentially homoplasic sites are removed. For the simulation data, the removal of homoplasies only significantly affects the results for low recombination rates *ρ/µ* ≤ 0.01, i.e. exactly when the frequency of true recombination events is lower than the frequency of homoplasies. However, the *E. coli* data is clearly in the regime where homoplasies have little effect on this statistic.

As shown in the new Figure 6—figure supplement 1, the observed ratios *C/M* for random subsets of strains are about 20% higher for the full alignment than for the 5% homoplasy-corrected alignment, i.e. *C/M* increases from 0.13 to 0.155 for the full set of strains. For the revision, we now show the results from the 5% homoplasy-corrected alignment in the main paper. In addition, following a request by reviewer 1, we also show *C/M* statistics for the subsets of strains of different phylogroups represented in our data (Appendix 1—table 2).

The analysis of C/M on the simulation data allows us to compare the lower bound on the recombination to mutation ratio *C/*(2*M*) with the ground truth, i.e. the ratio *ρ/µ* in the simulations. This analysis shows (Figure 6—figure supplement 2) that for very low recombination rates, i.e. *ρ/µ* ≤ 0.01, the ratio *C/*(2*M*) actually estimates the true recombination to mutation rate very accurately. However, as *ρ* increases the ratio *C/*(2*M*) severely underestimates the true ratio of recombination to mutation rate. For example, for the *E. coli* data we have *C/*(2*M*) ≈ 0.065, which is very close to the observed ratio *C/*(2*M*) for simulations with *ρ/µ* = 1, i.e. almost 20-fold higher.

We also expanded the section on the lower bound for the average number of times each position in the genome has been overwritten by recombination. Using data from simulations for which the ground truth is known we show that only at *ρ/µ* = 0.001 are there any positions left that have *not* been overwritten by recombination, at *ρ/µ* = 0.1 positions have been overwritten 120 − 200 times, and at *ρ/µ* = 1 each position has been overwritten 1450 − 1700 times by recombination (Figure 6—figure supplement 3). We also show (Figure 6—figure supplement 4) that our lower bound on the number of times each position has been overwritten by recombination is strongly underestimating the true number as soon as *ρ/µ* ≥ 0.1. Thus, although the lower bound for the *E. coli* data is that each position has been overwritten at least 125 times by recombination, the true number is likely larger than 1000 times.

Effect of homoplasies on the results (v): For the revision, we show in Figure 9—figure supplement 1 that the *n*-SNP distributions of the 5% homoplasy-corrected alignment are virtually identical to *n*SNP distributions of the full alignment. Also, none of the 2-SNPs in Figure 9A were removed by the 5% homoplasy correction.

Even though homoplasies hardly affect the overall patterns we see in the *n*-SNP statistics, for the revision we nonetheless used the 5% homoplasy-corrected alignments for all the *E. coli* results shown on *n*-SNP distributions and entropy profiles.

**Response to miscellaneous comments regarding part 1 of our claim**

For completeness, we will also address the remaining other comments that reviewer 2 made regarding part 1 of our claims.

**Using a strict criterion of phylogenetic consistency**

First, there are these 3 related comments:

A lot of the issues come from the use of a strict criterion for phylogeny support, that with more strains added gives an evident discrepancy as soon as a bit of recombination or homoplasy is present. So the present manuscript strongly validates the idea that there is some recombination but falls short of telling significantly how strong.The larger the dataset, the most likely it is that each fragment will be affected to some extent by both of these processes, and the phylogeny will be different, but if one strain out of 100 is changing place in the phylogeny, the phylogeny will be said to be incompatible, though it still holds some strong information. Similarly, for the bi-partition analysis, the more strains there are below a branch, the more likely it is that one of the strains will not fit the bi-partition. So once again the analysis is not conclusive regarding the quantitative importance of recombination.Analyses based on comparison of phylogenies (i), (iii), (iv) would benefit from a more progressive metric of agreement between two phylogenies. Two phylogenies can conflict for very minor reasons (e.g. different structure of small terminal sub-trees) or major topological changes. The authors choose a binary measure to compare phylogenies based on the presence or not of a specific strain bi-partition. But one could use a progressive measure of distance between two bi-partitions, e.g. based on the transfer distance (number of taxa that must be transferred to make the two bi-partitions identical) Lemoine et al. Nature 2018. In principle changes in topologies along the genome should be due to minor differences (small transfer distances).

We of course agree that, the more strains are involved, the easier it becomes to detect phylogenetic inconsistencies. We also agree that more ‘soft’ or graded measures of inconsistency across phylogenies could be useful to measure the statistics of similarities of the different phylogenies that occur along the alignment. The problem is, however, that if there is a lot of population structure such that relative recombination rates are very different across lineages (as we indeed claim is the case) then even when there is a saturating amount of recombination, the phylogenies at different loci may still show significant similarities. Therefore, one cannot reliably take similarity of phylogenies as a measure for the amount of recombination that has occurred.

A strict measure of inconsistency is in fact a more reliable way to quantify the amount of recombination. In particular, whether ‘small’ or ‘large’, each inconsistency is still an inconsistency. For each such inconsistency, there either is another explanation (e.g. homoplasy), or else it indicates the occurrence of a recombination event. The methods of Figures 3-6 precisely use this fact to quantify how much recombination must have occurred. And as mentioned already, the methods we use here are variations of very standard approaches used in the field of sexually reproducing eukaryotes.

We cannot understand how the reviewer can claim that we did not provide quantitative estimates of the importance of recombination. In fact, we provide several very direct quantifications of the amount and importance of recombination using several different methods. These include:

Figure 2G quantifies what fraction of the core genome alignment of pairs of strains has been recombined as a function of their total divergence. This is a direct quantitative measure of the strength of recombination that does not rely on any phylogenetic consistency evaluations at all.Figure 2H quantifies, for each pair of strain, what fraction of nucleotide differences between them were derived ancestrally versus from recombination events. This directly quantifies how much of the pairwise divergence of each pair is due to recombination, and again this measure does not use phylogenetic consistency at all.Figure 4 shows the relative numbers of supporting vs. inconsistent SNPs, separately for each branch of the tree. Although this is a more indirect measure of the amount of recombination, it does directly quantify the amount of inconsistency separately for each branch of the core tree.Figure 6 provides a direct lower bound on the ratio *C/M* of the number of phylogeny changes to mutation events in the core alignments not only of the full set of strains, but for arbitrary subsets of any number of strains. We can hardly see what more direct quantitative bound on the amount of recombination one could give. Indeed, our simulation analyses confirm that *C/*(2*M*) is a direct lower bound on the overall recombination to mutation ratio.

**Heterogeneity of rates along the genome and effects of selection:**

The distribution of differences along the genome is also not supposed to be homogenous along the core genome. Some genes are much more conserved than others even at synonymous sites, and the mutation rate has been shown to be locally heterogenous. Selection on recombinant may also differ along the genome, and some regions like the O-antigene lead to strong selection of recombinant that affect the neighbouring core genes. These regions may increase the estimated contribution of recombination. Selection may also counter select recent recombinant and lead to an overestimation recombination: in a similar way based on recent phage insertion in genomes we could conjecture that ancestral genomes should be only composed of prophages.

We think it is important to distinguish between the actual rates at which mutations and recombinations are introduced into individuals of the evolving populations, and the number of substitutions and recombination events that have occurred in the evolutionary history of the sequences in the core genome alignment. We are nowhere in the article attempting to estimate the actual mutation and recombination rates precisely because it is obviously very hard to accurately estimate the size of the effects of selection. Instead, we aim to provide estimates and bounds for the number of substitutions and recombination events that have occurred in the history of the observed strains. We tried to make this point clear in the revision.

**Reconciliation with approaches that claim to remove recombination events:**

It is difficult to reconcile this set of analysis with how people deal with recombination in bacterial genomics. Softwares exist to remove recombination events, some of them similar to the approach used by the authors (Figure 2), e.g. Gubbins or others (Bratnextgen). These allow estimating ratios of SNPs introduced by recombination over mutation (r/m ratios). Phylogenies can then be inferred on alignments without recombination. How does this approach compare to the authors’?

Although we did not refer to the two specific methods mentioned here (i.e. Gubbins and Bratnextgen), we discussed this general issue in the discussion of the results in Figure 1, i.e. we wrote:

“How should we interpret this convergence of phylogenies to the core phylogeny as increasing numbers of genomic loci are included? One interpretation that has been proposed, is that once a large number of genomic segments is considered, effects of horizontal transfer are effectively averaged out, and the phylogeny that emerges corresponds to the clonal ancestry of the strains, e.g. [8,11]. Indeed, it has become quite common for researchers to detect and quantify recombination using methods that compare local phylogenetic patterns with an overall reference phylogeny constructed from the entire genome, e.g. [11–15]. However, the validity of such approaches rests on the assumption that this reference phylogeny really represents the clonal phylogeny, and it is currently unclear whether this is justified. Indeed, some recent studies have argued that recombination is so common in some bacterial species that it is impossible to meaningfully reconstruct the clonal ancestry from the genome sequences, and that these species should be considered freely recombining, e.g. [27].”

Like the methods we cite here, methods like Gubbins and Bratnextgen simply start from the assumption that recombination is sufficiently limited such that the clonal phylogeny can be inferred from the core genome alignment. In particular, these methods assume that there are sufficiently many genomic loci whose phylogeny has not been disturbed by recombination so that the clonal phylogeny can be inferred from those loci. However, we are not aware of any demonstration that real data of bacterial genomes are in the regime where this assumption holds and, as we point out, there is no agreement in the community on this fact. Indeed, reviewer 1 expresses the opinion that it is true and ‘not novel’ that the precise clonal phylogeny cannot be reconstructed for most bacterial species.

One of the aims of our work is to rigorously investigate this assumption and we feel that our results clearly show that recombination occurs so often that no loci evolve free from recombination, and the clonal phylogeny cannot be reconstructed at all. In other words, we maintain that methods like Gubbins and Bratnextgen start from assumptions that do not hold for the bacterial species that we analyzed. We made this point clear in the revision.

**Summary, status of part 1 of our claims**

In summary, all our additional analyses and validations using data from simulations show that *none* of the technical issues raised can substantially effect any of the results (i)-(v). But even if there were valid questions about the reliability of some of our statistics, we feel that the validity of our overall conclusions are supported by something far stronger. The results of (i)-(v) are based on very different and mutually complementary methods, and these analyses all independently paint the same quantitatively consistent picture, i.e. that recombination is completely dominating and has overwritten virtually all clonal structure. Even if there were a technical problem with one or other of these methods, we find it extremely hard to see how *all* of the methods could consistently create the wrong impression that recombination is dominating.

In order to maintain the belief that, in spite of all our results, the clonal structure is what is determining the core genome phylogeny, one would have to argue that all results (i)-(v) are not only all wrong, but that they somehow also all fortuitously paint the same quantitative picture, and that for some reason they also by sheer luck happen to paint a correct picture on simulation data. We submit that this is simply not believable.

**Part 2 of our claims**

To most constructively discuss part 2 of our claims, we think it useful to start with the following comment from the end of the review:

Overall, a lot of remaining questions are not solved by the present model: if recombination is so massive, we would expect each transferred fragment to be itself the result of multiple fragments and therefore to leave a somehow fragmented signature of recombination, how does this model allow diversification to occur and most importantly why is the phylogeny overall stable since the use of MLST two decades ago…

First, as already mentioned above, we do not at all dispute that a clear phylogenetic structure is present in the data. Indeed we start by explicitly showing in Figure 1 that, given sufficiently many genomic loci, the phylogeny reconstructed from them converges to the core genome phylogeny. As we pointed out in the discussion of this result, some in the field see no alternative but to conclude that this stable phylogeny must correspond to the clonal phylogeny. But it is not at all impossible for there to be massive recombination and a stable genome-wide phylogeny at the same time. Even if, through massive recombination, there is a large number of different phylogenies along the genome, an algorithm that attempts to fit the data assuming a single phylogeny (like PhyML), will generally still converge to a stable result when given sufficient data. To give an analogy, imagine that we fit data coming from a complex mixture of Gaussian distributions to a single Gaussian. Given sufficiently many samples from the complex distribution, the fit to a single Gaussian will become very stable, even though the underlying distribution is not a simple Gaussian at all.

Since there clearly was confusion on this point we have included new sections in the revision explaining these points in detail. In particular, we illustrate the point by constructing phylogenies (using PhyML) from autosomal sequences from the 1000 genomes project, i.e. human genomes. We show that, even though there is of course no such thing as a clonal phylogeny for human autosomal chromosomes, phylogenies reconstructed from sufficiently many loci robustly converge to the’ core’ phylogeny. Moreover, this phylogeny reflects known population structure, i.e. individuals from the same geographic regions consistently form clades in the phylogeny (Figure 8). That is, humans with ancestors who predominantly come from a common geographic area end up closer in this phylogeny because they are more likely to share recent ancestors *on average* across the different phylogenies along the genome. In exactly the same way, we propose that the whole genome phylogenies for many bacterial species reflect the statistics of recombination between their ancestors.

**Do 'close' strains recombine more often?**

It seems that strains share SNPs much more frequently with close strains. This information can be found for strain A1 from comparing Figure 7A and Figure 1—figure supplement 1, showing that strain A1 shares a lot of 2-SNPs with its neighbours H5, D8, A2. If these SNPs are indeed shared because of the action of recombination, this means that a good description of recombination patterns would be one where the rate of recombination between pairs of strains is a decreasing function of their divergence. This relationship must be systematically investigated and presented. This relationship has been known for a while, has been used in models of diversification (e.g. Fraser et al., 2007) and seems a natural way to describe recombination structure. Yet this is not mentioned at all in the text.

I think this comment is very valuable because it illustrates so clearly the confusion that results from continuing to think of pairwise divergence as reflecting the distance to a clonal ancestor. That is, the reviewer’s comment presupposes that the pairwise divergences and rates of recombination are separate independent entities which do not *a priori* need to correlate, but that the data just happen to suggest that they do.

However, in the regime where recombination dominates almost every SNP can be thought of as deriving from an independent phylogeny. In that limit, in order for two strains *x* and *y* to have a different nucleotide in a given SNP column, the mutation must be more recent than the common ancestor of *x* and *y* in the phylogeny *at that position*. Thus, the divergence of a pair of strains does not reflect the distance to their clonal ancestor, but the *average* distance to their common ancestors across the many different trees. And these distances depend precisely on how often the lineages of strains *x* and *y* have recombined. If their lineages recombine rarely, then on average their common ancestors will lie further in the past, and if they recombine a lot, their common ancestors will on average be recent. Thus, when recombination dominates the pairwise divergence of two strains is a direct reflection of the recombination rates of their lineages, and it becomes almost a tautology that the lineages of strains that have low pairwise divergence must have recombined frequently. That is, rather than a surprising observation this correlation is in fact predicted by our interpretation of the data.

In the revision we have added comments to make this point clear. Also, we have added an Appendix in which we perform an in-depth analysis of the recombination structure of phylogroup B2 (to which the strains the reviewer mentions belong) to further illustrate and confirm this interpretation.

**Entropy profiles of *n*-SNPs**

We are not convinced about the interpretation of the 2-SNP, 3-SNP, etc., analysis. This analysis is interpreted as implying that each strain has a unique recombination profile, that “there is no natural way of dividing the strains into groups of highly recombining clades” and that individuals are not exchangeable. How do the authors quantify the uniqueness of the’ entropy profiles’ (Figure 10)?

For the revision, we have significantly extended the analysis of the entropy profiles and this answers the question of the reviewer in several ways. Apart from showing the entropy profiles of all *E. coli* strains in the main text, and discussing several example profiles, we now also show the entropy profiles of the strains of each of the phylogroups that are represented in the dataset (Figure 10—figure supplement 1). This analysis shows that, when a group of strains has a very recent common ancestor, their entropy profiles *H_n_*(*s*) are not just similar, they are exactly identical. In contrast, entropy profiles *H_n_*(*s*) for strains of the same phylogroup look similar for large *n*, but they are never identical. Moreover, entropy profiles of strains from different phylogroups tend to look clearly different, even at large *n*.

Following the reviewer’s comments, we also developed a statistical test (based on the Fisher exact test) for testing the difference in *n*-SNP statistics for a pair of strains. This analysis shows that about 95% of all pairs of strains have significantly different entropy profiles (Figure 10—figure supplement 2). Finally, we also show that the entropy profiles of the data of the simulations look very different from those of the *E. coli* data. Instead of highly variable entropy profiles, entropy profiles from simulations show consistently low entropies at low recombination rates, and very similar looking entropy profiles for all strains at high recombination rates (Figure 10—figure supplement 3).

**What do we mean with a population of exchangeable individuals?**

Finally, the article starts by posing two cornerstones of bacterial epidemiology: a robust phylogeny and a population like structure. It is not clear to us what is the meaning of the second assertion and how recombination differs from mutation in that process. In a coalescent as much as in coalescent with recombination, the genealogy has fractal structure and any of the possible clustering relies on some arbitrary choice. Whether mutation or recombination contribute to that process, there is no clear definition of population. Only at the ecological level can one find within a population of hosts some discrete bacterial populations.

We are here referring specifically to the concept of a population of exchangeable individuals in *mathematical models* of evolutionary dynamics. In mathematical population genetics models members of the ‘same population’ are treated as exchangeable. For example, in coalescent models, each pair of lineages is equally likely to coalesce. In models with recombination, such as the one we use in the simulations, each pair of individuals is equally likely to recombine. Even in models with substructure, e.g. with different subpopulations with migration between them, the members within each sub-population are still treated as exchangeable.

We believe that, in order to successfully model the *n*-SNP statistics and entropy profiles that we observe, we would have to introduce almost as many subpopulations as there are strains in our collection. If one needs almost as many subpopulations as there are strains to successfully model the observations, it becomes reasonable to question whether models that start from subpopulations of exchangeable individuals are appropriate at all. In the revision we have attempted to clarify these points.

**What would *n*-SNP distributions look like for no or free recombination?**

Moreover, while the distribution of 2-SNP for one strain may reflect some recombination, it is not clear what is expected for the overall distribution in the absence of recombination. A simple coalescent process could also presumably lead to a long-tailed distribution. The arguments made in the text on the typical mode of 2-SNP in the case of free recombination is not substantiated and should be validated by simulations or an analytical derivation. The whole topology of the tree will affect how many sites are shared by two strains. If a pair of strains coalesced, what matter for 2SNP is the time of the coalescence of their ancestor. This is not a homogeneous distribution. We need to know how much the distribution is informative about the recombination process, but the manuscript in its present form is not addressing that question: what is expected in the absence of recombination, in the presence of homoplasy, how much a little bit of recombination is affecting the patterns… These are key question if one want as the author to challenge the whole idea of phylogeny.

These are valid points. We could have discussed much more the interpretation of these distributions of *n*-SNPs. In addition, we agree that it is useful to see what such distributions look like for simulations of either a simple Kingman coalescent without recombination, or a model with mutation and free recombination within a well-mixed population. As already discussed, for the revision we have developed simulation code and performed such simulations with a large range of recombination-to-mutation rate ratios.

In the revision we now use the 5% homoplasy-corrected alignments for the *n*-SNP analysis, and in Appendix 4 we present extensive comparison of the *n*-SNP distributions observed for *E. coli* with those observed for the simulations, either without recombination, or with different recombination rates. This analysis shows that *n*-SNP statistics observed for *E. coli* are fundamentally different from the *n*-SNP statistics observed for any of the simulations. In particular, we confirm that at recombination rates as high as we observe for the *E. coli* data, no long-tailed distributions of *n*-SNPs are observed in the simulations.

Finally, we agree that the number of *n*-SNPs of a given type does not only depend on the fraction of the phylogenies along the genome for which that group of *n* strains form a clade in the tree, but also on the average length of the branch from their ancestor to the next coalescence. In the revision we have now explicitly pointed this out.

**Is our set of strains special?**

Of note the strains were also isolated in some atypical reservoir of *E. coli*. In this ecological niche it is not impossible that recent transfer may be magnified due to higher action of phage and larger diversity than within a regular gut (colonised by a few strains maximum). However the fact that the pattern is similar to the one described by Dixit and Maslov suggest this may not be too much of an issue.

The fact that we observed very similar statistics for the other species already shows that our observations are not specific to our set of *E. coli* strains. However, we have also performed our analyses on a collection of *E. coli* genomes from public databases and confirmed that these show the same overall patterns. We could in principle also include these analyses but this would be quite some extra work and we don’t think this would bring any useful additional information.

Reviewer 3 - Daniel Weissman3.1) This is a very exciting manuscript that changed the way that I think about bacterial evolution. The key findings are:• For several different species of bacteria, there are essentially no parts of the genome that are inherited along the clonal phylogeny in samples of size 100.• Despite this, there is a clear preferred genome tree, indicating that there are patterns in which some strains preferentially recombine with each other. These patterns persist deep into the past, beyond the point at which every gene has recombined off the clonal phylogeny.

We thank the reviewer for these comments. It is very rewarding to see that some readers do appreciate the key points of our work.

3.2) These findings indicate that there is some hidden structure in the studied bacterial populations. I think the nature of this structure is a real mystery. It could be ecological, epistatic, spontaneously arising from genome divergence suppressing recombination, or something else entirely. I hope that this paper will spark a wave of interest in trying to figure out what population genetics should look like in most of the tree of life. The authors make a first attempt at this by looking at some interesting distributions of shared SNPs across strains. I found these distributions hard to interpret and wasn’t sure about all the details of the analysis, but I think it’s good to try out some new ways of looking at the data.

We completely agree with the reviewer. Indeed, the key question is now to understand what this ‘population structure’ setting these very different relative rates of recombination between different lineages represents. We also agree that our *n*-SNP distributions (and *n*-SNP entropy profiles) are only a first step toward solving this puzzle. To make clearer what these *n*-SNP distributions represent, we now also calculate these distributions for data from simulations of completely mixed populations evolving under (neutral) mutation, drift, and recombination and compare and contrast the patterns that are observed for the real data with the patterns that are observed from these simulations. We also show what the entropy profiles look like for the traditional *E. coli* phylogroups.

3.3) I would partially reorganize the paper so that it’s organized more by method/analysis than species. Concretely, I would move Figure 7 (non-*E. coli* bacteria) and the accompanying analysis up so that they come before Figure 9 (SNP-sharing distributions) and its accompanying analysis. This way, you would introduce the tree analysis in *E. coli*, then apply it to other species showing that you get similar results, then move on to SNP sharing.

Upon reflection, we agree with the reviewer that it makes sense to first show that the dominance of recombination is also observed for other species, and then turn to the question about what the phylogenetic structure in the data represents. Thus, for the revision we have reorganized the presentation as suggested by the reviewer.

3.4) I would try changing the analysis comparing SNPs to putative branches. Right now, it’s just consistent vs inconsistent, which is nice because it’s simple. But I don’t think it’s that informative. If you have a putative branch that splits the sample in half, and a SNP at 50% frequency, and they totally agree except for a single sample, that gets called “inconsistent”, whereas a random SNP that’s only found in a pair of individuals has a 50% chance of being consistent. Similarly, a putative branch that just lies above a pair of samples will be consistent with lots of SNPs just by chance. What about using mutual information, either instead of consistency or as an additional analysis? I think that Figure 8—figure supplement 1 indicates that you have lots of SNPs that are nearly or partly consistent, given that you recover something very close to the original tree using only inconsistent SNPs. Hopefully an MI analysis would pick these up as still being correlated with the branches. (Of course, MI can’t go negative, but you can still classify branch-SNP pairs as correlated or not depending on whether the MI exceeds what you would get if you drew them both at random conditioning on number of strains under the branch and SNP frequency.)

In Figures 4, 5, and 6 the aim is to show that the SNP inconsistencies imply certain lower bounds on the number of recombination events and show that very large fractions of SNPs do not correspond to mutations along the branches of the core tree, i.e. that one cannot approximate the dynamics as taking place along the branches of the tree. For that it is essential to categorize SNPs as simply consistent/inconsistent with particular tree branches or with each other.

At the same time, we of course agree that the distribution of SNP types is not random at all but that some lineages are much more likely to recombine with each other than others. That is, whereas recombination is so frequent that many thousands of phylogenies occur along the alignment, these phylogenies are not completely random but drawn from some biased distribution over phylogenies. How to efficiently capture or summarize this distribution over phylogenies is an entirely separate question. In the paper we have proposed the *n*-SNP distributions and *n-*SNP entropy profiles of individual strains. For the revision we now also provide a new statistic that quantifies the similarity of the *n*-SNP statistics of two strains. We agree that additional more graded measures of similarity between SNPs could be helpful but it is not entirely clear to us what precisely measures such as the mutual information that the reviewer proposes would capture. It is still the case that a single recombination event can either make a large or a small change to the SNP pattern.

Figure 8—figure supplement 1 simply shows that, if one is forced to summarize the patterns of SNP sharing across the strains (which really reflects the distribution of phylogenies along the alignment) by a single ‘average’ phylogeny, then this average is very similar whether one includes the SNPs that actually fall on this tree or not. This observation is analogous to observing that the median or average of a sample from a distribution is typically unchanged under removing data points near this median/average.

3.5) I don’t understand the argument that because Figure 9C shows broad distributions, that must mean that strains are not organized into subpopulations. Figure 11—figure supplement 3 shows a similar pattern for humans – are the authors claiming that subpopulations are not a good description of the human data? At the very least, couldn’t the sub-population sizes be broadly distributed? But more basically, do we know what these distributions should look like in a simple well-mixed, recombining Wright-Fisher population? I’m sorry, I really meant to simulate this, but I’m turning in this review late as it is. Maybe you get that the occurrence of a particular kind of n-SNP can be dominated by its occurrence on a single genome block?

We agree that interpreting what the *n*-SNP distributions indicate regarding the population structure is still challenging at this point. We presume that, if there were clear subpopulations with high recombination rates within them, and low recombination rates between them, then this would be visible as specific substructure in the *n*-SNP distributions. However, we agree we have no strong argument to support this and for the revision we have rewritten this section to make clear that the precise interpretation of the *n*-SNP distributions is currently unclear.

Following the suggestion of the reviewer, we now extensively compare the observed *n*-SNP statistics with statistics from simulations of a simple well-mixed Wright-Fisher model with the product of population size and mutation rate set to match the sequence diversity observed in our *E. coli* strains, and recombination-to-mutation rates ranging from *ρ/µ* = 0.001 to *ρ/µ* = 10 (Appendix 4). We show that the *n*-SNP distributions observed in these simulations differ significantly from what we observe for the real data. In the limit where recombination is very weak there are not many *n*-SNP types and they essentially capture the structure of the clonal tree. When recombination is high (like for the real data) the complete mixing causes different *n*-SNP patterns to occur with similar frequency, and the distributions are no longer long-tailed.

With regard to the evidence of the population separating into subpopulations, we have moved our discussion of this issue to the discussion of the entropy profiles, since these contain more relevant information about this question.

**Detailed questions/comments:**

3.6) What’s the allele frequency spectrum look like? I don’t think I saw this anywhere in the manuscript.

This was shown in Figure 4—figure supplement 2. We have extended this analysis to include data from the simulations and the allele frequency spectra are now shown in Appendix 4—figure 1.

3.7) How well does the consensus tree capture the pairwise distances among strains?

We now show this in Figure 4—figure supplement 1. Since the pairwise divergences vary over 4 orders of magnitude, there is a good correlation between the pairwise divergences and the pairwise distances in the core tree, but the distances along the tree are systematically lower than the true distance. Moreover, the branch lengths of the core tree and the observed frequencies of SNPs falling on each branch differ by up to a factor 100 (Figure 4—figure supplement 1, right panel).

3.8) Could you say more about the potential effects of the SNPs that correspond to multiple mutational events? If I understand the numbers correctly, in one of the 3kb blocks you expect about 5 SNPs to actually come from two independent mutations. Does PhyML take this into account? I don’t think it should be a big deal for most of the rest of the analysis, since Figure 5 indicates that you see inconsistencies after about 10 SNPs, and you don’t expect a multi-origin till you have about 50 SNPs, but I think it would be worth going through this explicitly.

We are glad to see that the reviewer appreciates that a back-of-the-envelope calculation already shows that multiple mutations (homoplasies) cannot substantially affect our analyses. However, we agree a explicit calculations are helpful. For the revision we have done extensive analyses of the potential effects of homoplasies. First, we updated our estimate of the rate of homoplasy taking transition-to-transversion bias into account. We now estimate about 3.3% of bi-allelic SNPs to correspond to homoplasies (this previously was 2.5%). Second, we developed a method for ranking SNP columns by the overall amount of phylogenetic clashes they generate with other SNPs in their neighborhood. To correct for homoplasies we then removed either 5% or 10% of the most clashing SNPs from the alignment and redid all our analysis with this homoplasy-corrected alignment. We also investigated the effects of homoplasy removal for all the simulation data. These analyses show that, indeed, none of our results and conclusions are significantly affected by homoplasies.

With regards to PhyML, this indeed implements a full likelihood model that does allow for multiple substitutions.

3.9) What do the wiggles in Figure 9C mean? They’re produced by the tree structure, right, and correspond to approximately exchangeable strains?

We’re sorry but we are not sure what the reviewer has in mind. We can see wiggles in Figure 7C (now 9C) but we do not know what causes them and cannot think of an easy way to figure this out. Frankly, we would already be happy to have a theory for the overall slope of these approximate power-laws.

3.10) For the human analysis, was it too much computationally to do more than just chromosome 21? Could you do all the autosomes for individuals from just one subpopulation at least?

For the revision we have now done chromosomes 1-12.

3.11) Did you ever think about just running STRUCTURE on the *E. coli* data and seeing what it said for different values of k? Maybe it would find “subpopulations” that would be clustered on the consensus tree.

We hope the reviewer can understand that, given very large number of additional analyses we already performed in response to reviewers 1 and 2, we feel that this rather exploratory additional analysis is beyond the scope of this manuscript.

3.12) I don’t find the power law fits to the SNP-sharing distributions very convincing, particularly the ones shown in Figure 11—figure supplement 1. I would cut the left panel of Figure 9, and I would qualify the sentence on p13 starting “Moreover…”

We do not agree with the reviewer here. For some reason, whereas one rarely hears complaints about fitting a linear trend to scatters that are far from straight, nor complaints about fitting an approximate exponential increase or decay to scatters on a semi-log plot, the moment one fits a line on a log-log plot there are immediately complaints that the data are ‘not really power laws’. We are not claiming that these are perfect power laws. However, it is a fact that these scatters look far more straight in log-log scales than on linear or semi-log scales. The fits just capture the overall trend of these long-tailed distributions. We have now explicitly pointed out that we are not claiming that these distributions are perfect power-laws.

3.13) What do the entropy curves look like for the human data?

For the revision we have performed this analysis, which is shown in Figure 11—figure supplement 3. Interestingly, the entropy profiles are virtually identical for all individuals, except for the individuals with African ancestry, which have clearly larger entropy at large *n*. Note that the similarity of the entropy profiles doesn’t imply that there aren’t any human subpopulations. It just implies that the diversity of different *n*-SNPs across the genome increases in a similar fashion for all individuals.

[Editors' note: we include below the second set of reviews that the authors received from another journal, along with the authors’ responses.]

Reviewer 1 - Daniel Falush:1.1) This paper is substantially improved, thanks especially to the introduction of simulations and extensive reworking of many sections and the fixing of logical and other errors. I still have a number of substantial problems with it, as I will explain below. I do find the response more confrontational and even highhanded than is necessary, to the point of rudeness. Within the cover letter this is exemplified by bringing out one clich´e of the self-proclaimed oppressed revolutionary scientist, Max Planck’s quote that science progresses one funeral at a time. Which comes perilously close to implying that the field would be in better shape if me and the other people who were not happy to accept the work in its current form were no longer in it. The counterpoint to that quote is the opinion that I first saw in a movie about the life of Sigmund Freud but I believe has another original source: “The work contains much that is new and much that is true. However that which is new is not true and that which is true is not new.” Which is what the old turks in most scientific fields think of the great majority attempts to foment revolution. And at least 90% of the time they are right.

We are sorry that our attempt at levity toward the end of the cover letter appears to have misfired. We hope that the reviewer noticed that we did not approvingly cite Max Planck, but instead pointed out that we are much more optimistic than Max Planck and expected that we *can* convince the ‘old turks’ that our work contains results that are both new and true. We’ll leave it up to the reviewer to decide whether this optimism is warranted.

1.2) The manuscript is greatly improved and some important errors have been fixed. I thank them for the extensive time and effort they have put into this; once they did it they made an effort to do it properly but I wish to remind them that reviewers and editors put their time in gratis. We could be doing our own research instead. The onus is on authors, not editors or reviewers to make the manuscript and its relationship to previous literature clear and unfortunately this still has not been achieved. Despite the many improvements, establishing what is new and what is true in this manuscript remains a substantial problem, firstly because the Introduction continues to set up a straw man for what is actually believed within the field. The authors correctly point out that neither a phylogenetic approach or a population genetic approach is fully applicable to bacteria. However this is common knowledge in the field. For example Didelot and Falush, 2007, when attempting to apply ClonalFrame to entire bacterial species state: A more ambitious use of the method is to attempt to define the clonal relationships and imports for a sample from an entire bacterial species. The assumptions of the model fit less well since in many instances recombination will reassort existing polymorphisms rather than introduce novel ones. In addition, the deeper branches in the genealogy might be difficult or impossible to resolve accurately, especially if there has been substantial recombination so that most of the genome of each strain has been exchanged since they shared a common ancestor. The authors imply that it is believed within the bacterial population genetics field that phylogenetic methods should recover the true genealogy but this is simply not true. ClonalFrame, Gubbins and other methods reconstruct genealogies while simultaneously trying to account for recombination events, in the latter case by removing recombined regions from the alignment and clonalframe attempts to use MCMC to represent the substantial uncertainty that exists about the genealogy while other methods (BURST, BAPS, PopPunk) forgo attempts to reconstruct an entire genealogy and instead try to identify relationships amongst closely related strains at the tip of the genealogy. Insofar as people in the field do use phylogenetic methods to infer relationships, this is a compromise, most usually due to computational issues. Unfortunately the MCMC approach to inferring genealogies used by ClonalFrame does not scale up, so ClonalframeML uses a maximum likelihood tree but this does not mean that the authors of that paper believe it to be the correct answer and I myself am currently involved in attempting to make a method that is based on inferred local genealogies and that attempts to indicate which branches cannot be inferred with any statistical confidence. The Introduction of the manuscripts states: First, we find that for most bacterial species recombination is so frequent that, within an alignment of strains, each genomic locus has been overwritten by recombination many times and the phylogeny typically changes tens of thousands of times along the genome. Moreover, for most pairs of strains, none of the loci in their pairwise alignment derives from their ancestor in the ancestral phylogeny, and the vast majority of genomic differences result from recombination events, even for very close pairs. Consequently, the ancestral phylogeny cannot be reconstructed from the genome sequences using currently available methods and, more generally, the strategy of modelling microbial genome evolution as occurring along the branches of an ancestral phylogeny breaks down.In my view, none of these claims are new, or indeed depart obviously from the informed consensus in the field.

In his previous review the reviewer characterized our paper as making ‘provocative claims that are not supported by the evidence provided’, and the review focused on pointing out perceived problems with our methodology, leading the reviewer to claim that our methods were sufficiently flawed that our conclusions could not be trusted. In response we thoroughly addressed all methodological issues raised and showed that our conclusions were in fact fully warranted. The reviewer appears to accept this because (with the exception of a discussion of potential underestimating of the average length of recombined regions that we respond to below) the reviewer no longer raises any technical issues. In light of this, we hope the reviewer can understand that we are quite surprised that he now takes the position that it is unclear to what extent any of our claims are novel. In response, we first of all note that this position is clearly at odds with those of reviewers 2 and 3.

The central novel insight of our work is that the robust phylogenetic structure that is evident in whole genome alignments of bacterial strains does not reflect clonal ancestry, but reflects population structure. We indicated this in the *title* of our manuscript, we explained it at length in the paper, and again in our responses to the reviews. In addition, to avoid any conceptual misunderstanding on this central point, we included a new section in the first revision showing how population structure can cause robust phylogenetic relationships to appear even in a manifestly non-clonal species such as human.

Unfortunately, the reviewer simply does not engage at all with this central result. It is unclear to us whether the reviewer has not understood this central claim, or that he simply rejects it without giving a justification for this rejection. It is clear from the reviewer’s comments, however, that he continues to assume that robust phylogenetic structures in whole genome alignments can only reflect ancestry. Although the reviewer quotes from Didelot and Falush, 2007, to point out that it has been acknowledged that approaches such as ClonalFrame are not perfect, and that it may be impossible to accurately reconstruct branches deep in the clonal tree, the entire premise of such methods is that much of the clonal tree *can* be inferred from the whole genome alignment. Indeed, in the section of Didelot and Falush, 2007 following the passage that the reviewer quotes, these authors apply ClonalFrame to a dataset of sequences from *multiple* Bacillus species, implying that they believe clonal relationships can be successfully inferred even beyond individual species. This is also consistent with statements of this reviewer in his first review: “In the discussion the authors conclude (which is not novel), that the ancestral phylogeny is very difficult, or impossible to reconstruct. This is true for most bacterial species, but this is typically because there are branches at the base of the tree that are extremely uncertain. It is still likely possible to infer a great deal about bacterial evolution from inferring clonal relationships closer to the tips. For most species, this includes a fairly substantial proportion of the clonal tree.” In summary, while the reviewer acknowledges that it may be difficult to reconstruct the entire clonal tree, it is clearly assumed that approaches such as ClonalFrame, Gubbins, and ClonalFrameML can be used to recover much of the clonal tree.

Our work fundamentally challenges this assumption and several passages in our manuscript explicitly stated this:

Subsection “Phylogenies of individual loci disagree with the phylogeny of the core genome” introduces the main assumption, explicitly cites methods such as clonalframeML as examples of it, and makes clear we are going to question this assumption: “How should we interpret this convergence of phylogenies to the core phylogeny as increasing numbers of genomic loci are included? One interpretation that has been proposed, is that once a large number of genomic segments is considered, effects of horizontal transfer are effectively averaged out, and the phylogeny that emerges corresponds to the clonal ancestry of the strains, e.g. [8,11]. Indeed, it has become quite common for researchers to detect and quantify recombination using methods that compare local phylogenetic patterns with an overall reference phylogeny constructed from the entire genome, e.g. [11–15,27]. However, the validity of such approaches rests on the assumption that this reference phylogeny really represents the clonal phylogeny, and it is currently unclear whether this is justified.”In subsection “Phylogenetic structures reflect population structure” after having shown that recombination is so common that hardly any clonally inherited DNA is left in the genome alignments, we explicitly point out that these results invalidate this assumption: “As we mentioned in the Introduction, some researchers interpret this convergence to the core tree to mean that the core tree must correspond to the clonal phylogeny of the strains. The interpretation is that the SNP patterns in the data are a combination ‘clonal SNPs’ that fall on the clonal phylogeny, plus a substantial number of ‘recombined SNPs’ that were affected by recombination, which act so as to introduce noise on the clonal phylogenetic signal. In this interpretation, trees build from individual loci can differ from the core tree because the ‘recombination noise’ can locally drown out the true clonal signal, but once sufficiently many loci are considered, the recombination noise ‘averages out’ and the true clonal structure emerges. However, this interpretation makes two assumptions that do not necessarily hold. First, assuming that the effects of recombination will ‘average out’ when sufficiently many loci are considered only holds if recombination is effectively random and does not have clear population structure itself. Second, for the clonal phylogeny to emerge one has to assume that there are sufficiently many genomic loci left that have not been affected by recombination, whereas our analyses above have shown that there is no clonally inherited DNA left for most pairs, and that each locus in the genome has been overwritten many times by recombination.”We then go on to show that the first assumption, i.e. that there is no population structure, also has to be rejected, and that instead of averaging out, this population structure can cause robust phylogenetic structures to appear.Finally, in the discussion we summarized our findings by stressing that methods which assume that much of the clonal tree can be recovered are inherently flawed: “Given this, there is no reason to assume that a phylogeny reconstructed using maximum likelihood from the core genome alignment corresponds to the clonal phylogeny of the strains. Moreover, methods that aim to reconstruct the clonal phylogeny from a subset of positions that have not been affected by recombination are also inherently problematic because our analysis shows that, for most species, such positions simply do not exist.”

As a general principle, we try to avoid directly attacking previous work of other researchers as fundamentally flawed in our paper. Instead, we assumed that the passages cited above would make it obvious to readers that approaches such as ClonalFrame, Gubbins, and ClonalFrameML are fundamentally flawed because they rely on assumptions that do not hold.

However, since this was apparently not clear to the reviewer, we can explain more explicitly what the problems are with an approach such as ClonalFrame. For a given set of sequences, ClonalFrame aims to reconstruct the clonal tree of these sequences using a model that assumes that

The clonal tree follows Kingman coalescent statistics.Along each branch of the tree, some genomic segments evolve clonally at a constant mutation rate, while other segments are overwritten by recombination events.The lengths of the recombined segments are exponentially distributed with a mean length fitted from the data.It is assumed these recombination events do not copy a DNA segment from another member of the population, but rather simply bring in random substitutions at a fixed rate.

There are several problems with these assumptions. First, there is no reason to assume that the clonal relationships of real sequences should follow Kingman coalescent statistics. Second, the average length of the recombined segments is typically fitted to something relative small, e.g. 1 kilobase. Given the assumed exponential distribution, this would make it virtually *impossible* to ever see recombined segments of 10Kb or more. In contrast, based on comparison of very close pairs of strains in *E. coli*, as much as half of the recombined segments range from 20 to 80 Kb. However, these are not our main concern. The main concern is the last assumption, i.e that recombination events simply bring in random substitutions at a fixed rate. It is not only unrealistic to assume that all recombined segments are always at a fixed divergence from the sequence they recombine into, more importantly, it assumes that there is no population structure at all. As a consequence, in this model recombination cannot introduce any phylogenetic structure *by construction* and any phylogenetic structure evident in the data will per definition be assumed to reflect clonal relationships. Thus, approaches such as ClonalFrame will mistake the phylogenetic structure introduced by population structure for clonal relationships by construction.

In the latest revision we have edited the text in a number of places to make these points clearer, including some comments in the Discussion.

1.3) There is a significant (and not cited) literature on estimating ratio of changes due to recombination and mutation (most notably an article by Vos and Didelot from 2008) which show that there are many species where R/M is substantially above one. Are the predictions of the authors actually substantially different from these either methodologically or in terms of the estimates produced. If so why? Even if they are, the qualitative conclusion is not different.

In order to determine R/M, the study of Vos and Didelot uses ClonalFrame to determine both the clonal tree, as well as the recombination events. For reasons just explained, we do not believe one can validly estimate the extent of recombination in this way. More importantly, however, we never estimated a value for R/M in the paper, because we believe this is conceptually misleading. That is, one of the key results of our paper is that *there is no such thing as a single ratio R/M for a bacterial species*. The ratio R/M depends both on the ratio of mutation and recombination *ρ/µ* as well as on the number of substitutions that are introduced per recombination event, which depends both on the lengths of the recombination events, and on the divergence between the donor and acceptor strains of the recombination events. Since our results show that there is a significant population structure, with rates of recombination varying dramatically between different lineages, it would be misleading to pretend that a species can be represented by a single ratio R/M.

We note that we explicitly pointed this out in subsection “lower bound on the ratio of phylogeny changes to substitution events” of the previous revision: “The observed ratio C/M for the *E. coli* alignments is very similar to the value of C/M observed in the simulations with ρ/µ = 1 (Figure 6—figure supplement 2). Although it is tempting to conclude from this that the ratio of recombination to mutation rate must be close to 1 for the *E. coli* strains, such a conclusion would be unwarranted. The evolutionary dynamics of the simulations makes several strong simplifying assumptions, i.e. that the clonal phylogeny is drawn from the Kingman coalescent process, and that the population is completely mixed so that all strains are equally likely to recombine. Both these assumptions may not apply to the evolution of *E. coli* in the wild. Indeed, we will see below that there is strong evidence that relative recombination rates vary highly across lineages so that instead of a single recombination rate, there is a wide distribution of recombination rates between different lineages. Therefore, it could be misleading to describe *E. coli*’s evolution by a single recombination rate ρ and we instead focus on providing a lower bound C/M on the number of phylogeny changes per SNP column, which is a meaningful quantity independent of the precise evolutionary dynamics that caused the substitutions and phylogeny changes, and can be calculated independently for any subset of strains.”

This point was reiterated in the Discussion section: “Given that recombination rates vary over orders of magnitude across different lineages, the idea of an effective single recombination rate for a species might be misleading, and it seems problematic to fit the data to models that assume a constant rate of recombination within a species [34].”For the latest revision we have added the citation to Vos and Didelot at this point.

We note that closest to estimates of R/M are the results shown in Figure 2H. These show, for each pair of strains, the fraction *f_r_*of the single nucleotide differences between the pair of genomes that is due to recombination. The ratio R/M thus roughly corresponds to *f_r_/*(1 − *f_r_*) in the limit of very close pairs. However, as can be seen from the figure, the ratio *f_r_/*(1 − *f_r_*) becomes highly variable for close pairs that are mostly clonal. This already suggests that *R/M* can vary significantly across different pairs of closely related strains, and our results on *n*-SNP statistics confirm this. In the latest revision we have pointed this out in the section on the pair statistics.

In summary, both our approach and our results are qualitatively different in that our results show there is no single ratio *R/M*.

1.4) The claim about most pairs of strains having no genome shared by recombination is also true for many species. See for example our preprint”why panmictic bacteria are rare”, which shows that for many species, the recombination scaled effective population size is 100 or more, which indeed means that for most pairs of strains the proportion of shared inherited clonal frame will be vanishingly small.

Like in the other studies that the reviewer cites, the cited paper uses a model that assumes that the clonal phylogeny obeys Kingman coalescent statistics, that there is no population structure (i.e. panmixia), and that recombination introduces segments of DNA at a fixed divergence from the donor strain. In particular, it uses such a model to estimate *N_e_*×*r*. The main result of this work is that the data *rejects* the assumption of panmixia and we in fact cited this paper in our Discussion section in this context.

However, given that the panmictic model was rejected by the data, we do not see how it can be used to accurately estimate what fraction of pairs have still clonally inherited DNA in their alignment. In fact, the methods used in the paper seem to *assume a priori* that pairwise divergences are dominated by recombination.

1.5) Phylogeny (I would prefer the more exact terminology local tree or local genealogy) changing tens of thousands of times is not a quantitively surprising prediction (and is very dependent on the size of the sample). We do not currently have the technology to make robust inferences of tens of thousands of distinct local trees but we know that the tree changes many times. This is just not new either. The second set of claims potentially has more novel elements but has a number of problems of coherence. Second, we show that the structure represented in whole genome phylogenies of microbial strains does not reflect ancestry, but instead the relative rates with which different lineages have recombined in the past.“ancestry” is normally used (e.g. in human genetics) to reflect the gene pools from which DNA comes. I think that here it is meant to mean “clonal inheritance”. But if this is what is meant it is an overstatement since the authors do analyse pairs of closely related strains in order to estimate relative ratio of recombination and mutation. In so far as it is a coherent statement it simply seems to be a restatement of the claims made in the paragraph above.

We agree with the reviewer that the term ancestry may be ambiguous and have rephrased in our latest revision to make clear that we are referring to the clonal phylogeny.

Apart from this, the reviewers comments just repeat what we already addressed in response 1.2, i.e. the reviewer does not engage with the central claim made in the title of our paper that the whole genome tree reflects population structure. We have rephrased this section of the Introduction in an attempt to be clearer.

1.6) Whereas almost every short genomic segment follows a different phylogeny, these phylogenies are not uniformly randomly sampled from all possible phylogenies, but sampled from highly biased distributions.Some parts of every local tree will follow the clonal frame and thus be a highly non-random subset of all possible local genealogies. So this is predicted even under a standard coalescent with gene conversion model. The bias can of course be made larger by non-random recombination.

This comment simply revisits the view that substantial parts of the local genealogies will reflect the clonal phylogeny. However, this is only true if a substantial fraction of the genome is unaffected by recombination along each branch of the clonal tree, and our results show that this is not the case. While the reviewer acknowledges that non-random recombination, i.e. population structure, can increase biases in the distribution of phylogenies along the genome, he somehow fails to acknowledge that this non-random recombination can itself induce phylogenetic structure. We expected that our analysis of the human data would have driven this point home, but for the revision we have rewritten this section in an attempt to further clarify this point.

1.7) In particular, the relative frequencies with which particular sub-clades of strains occur in the phylogenies at different loci follow roughly power-law distributions and each strain has a distinct distribution of co-occurrence frequency with the other strains.It took a great deal of effort before I was able to start to parse this key sentence and I am still left guessing what co-occurrence frequency refers exactly to.

Whenever a SNP occurs that is shared only by the triplet of strains (A,B,C), then the strains (A,B,C) form a clade in the phylogenetic tree of that particular locus. The statement simply means that we observe that the frequencies of different triplets (A,B,C) follows a power-law. For the revision we have rephrased so as to make this clear.

1.8) Since almost every strain has a unique ‘finger print’ of recombination rates with the lineages of other strains, the assumption that at some level the strains can be considered as exchangeable members of a population, also fundamentally breaks down.The idea that recombination is non-random is also extremely well known within the bacterial field. This is firstly due to laboratory experiments that have shown that in many species recombination rates are suppressed as a function of the number of mismatches between donor and recipient. Secondly, there have been studies done on *E. coli*, which show that recombination is highly non-random, with higher rates occurring within phylogroups than between them. Therefore this claim is not new either. See Didelot, Meric, Falush and Darling, BMC Microbiology 2012, and also the literature cited therein.Therefore, I have got to the end of the Introduction and still have little idea what the novel claims of the manuscript actually are.

We are well aware that biases in recombination have been shown previously, and in fact explicitly discussed the suppression by mismatches in the Discussion section, including citations. However, we are not aware of any studies showing or even suggesting that the phylogenetic trees that are reconstructed from whole genome alignments reflect the statistics of biases in recombination rates.

1.9) We can also provide a rough estimate for the typical number of times T that each position in the genome has been overwritten by recombination in its history.It needs to be made clear what “its history” means. This means anywhere across the clonal genealogy of these strains. It does not cover events before their common ancestor. It is the property of a sample of strains. It is not the property of any one strain for example, since this will also have undergone recombination events going back to LUCA.

We are referring to the history of a *particular position in the core genome alignment*. Note that, because of recombination, the nucleotides at different loci derive from different ancestors with different histories and a different common ancestor. It is thus not only a property of the set of strains but also of the specific position and *T* corresponds to the average across positions. We have rephrased in the revision to make this point clear.

1.10) I thank the authors for doing simulations. The reasons I am personally sceptical of these estimates of T is that I think 20,000 base pairs is a substantial over-estimate of the mean length of the recombination events that change the local trees. Some recombination events will be this long and these are the easiest events to detect. However there can also be many shorter events that still introduce 2 changes in local genealogies. It is known for example that individual events can lead to introduction of several fragments, many of which are short, within a long sequence block. For example see Golubchik et al. Nature genetics 2012. These events naturally get fused together in the kind of pairwise analysis that the authors do. It can be that the events of more than 20,000 bases are responsible for half of the Introduction of DNA but only responsible for a very small fraction of changes in local genealogy. If the authors estimated mean recombination fragment size by methods based on LD rather than by attempting to characterize individual events they would I think get much shorter estimates. I am not exactly sure what the authors should do to address this issue. One is to ask whether the pattern of LD as a function of genetic distance is really consistent with 20,000 bp imports. Another is simply to acknowledge the methodological difficulties.

The reviewer imagines that our identification of recombination segments in very close pairs may miss short segments, and that this may cause us to significantly overestimate the average length of recombined segments.

For this revision we have included a new Appendix 2 discussing why we believe it is unlikely that we are underestimating the average length of the recombination segments by a significant factor. To summarize: Our method does not have trouble detecting relatively short segments, i.e. fully 30% of the recombination segments we identify are only 1-2Kb long. Second, because typical pairs of *E. coli* strains are 1 − 3% diverged, even a recombined segment of only 250bp long would bring in 4-7 mutations, and this would easily be detectable in the alignment of a very close pairs of strains, which have at most 2 SNPs or less per kilobase in clonally inherited regions. In order for our method to plausibly miss a recombined segment it would have to be shorter than 100bp. Third, as we discuss in the Appendix 2, in order for our estimate of the average length of the recombination segments to be too large by a factor *n*, we would have to assume that for each recombination event of length 250 bp or more that we can detect, there are *n* recombination events of length 100 bp or less that we cannot detect.

The LD analyses that the reviewer mentions are based on explicit population genetics models that make a number of assumptions that, according to the results we present, simply do not hold. Consequently, we do not believe the results of such analyses can be trusted. To give just one example, such analyses typically assume that lengths of recombination segments are exponentially distributed and typically fit a mean length of 1 kilobase or less. Under such a model, recombination segments of length 10Kb or more would be essentially impossible. However, our pairwise analysis of very close pairs clearly shows that recombination segments of tens of kilobases long are in fact common (Figure 2A, B, and J).

The only statistic that we report that is affected by the estimated average length of the recombination segments is the average number of times *T* each position in the alignment has been overwritten by recombination. This latter estimate is based on the lower bound on the number of phylogeny changes in the alignment. Comparison with simulations suggests that this estimate may well be 10-fold below the true value. Thus, in order for our estimate to still be an overestimate one would have to assume that our 20Kb recombination segment length (which is already 50% less than the actual mean that we estimated) is an overestimate by 10-fold or more. We think it is very implausible that there are 10 times as many recombination events of length 100 bp or less as all recombination events of length 250bp or more.

In addition, in response to a suggestion by reviewer 3 we have included an Appendix 3 with a complementary estimate of the number of times each position has been overwritten by recombination, based on the statistics from the pair analysis (see response 3.5). The results of this analysis are entirely consistent with our estimate based on the minimal number of phylogeny changes across the genome alignment, i.e. suggesting that each position has been overwritten somewhere between 200 and 500 times by recombination on average.

Finally, as already pointed out in our previous response, *even if* our estimate were still a significant overestimate, this would still not affect any of the main conclusions of our paper.

For the revision we have further rephrased the discussion of this estimate to make clear that it could be affected if there were very large numbers of very short recombination events that we cannot detect.

1.11) The consensus in the field based on careful work of many scientists is that M. tuberculosis has undergone no recombination at all in its history back to its last common ancestor and that when authors detect it, it is due to methodological issues (either with the analysis method or with the data). If the authors wish to claim that they have detected positive evidence for recombination in the species, they need to spell out why. Otherwise, they need to comment on which the deviation from this established truth says about the practical limitations of their methods.

We agree that, while there are some reports in the literature of evidence for recombination with *M. tuberculosis*, the consensus is that, if there is any recombination, it must be very rare. In line with this, almost all of our statistics on *M. tuberculosis* confirm that, if there is any recombination, it is really rare. The pair statistics infers that all pairs are almost fully clonal, there is no diversity of *n*-SNP patterns, and the *n*-SNP entropy profiles all have low entropy. The only potential evidence of recombination is the occurrence of a (small) fraction of SNPs that clash with the core tree. Since these clashing SNPs are relatively rare, i.e. roughly one every 4.7Kb along the genome, we do not think it can excluded that they are due to problems with either the assembly of some strains, or their alignment.

Since the reviewer asked we have performed some more analysis of the *M. tuberculosis* SNPs for this revision. We discovered that the majority of clashing SNPs involved a particular pair of strains and found that these two strains have since been *removed* from the database because of evidence of contamination in these sequences. In total the genomes of 5 of the strains that we used have since been retracted. We removed these strains from the alignments and redid all analysis to find that the rate of clashing SNPs has now been reduced to once every 12.7Kb. Although there are still clashing SNPs left, we obviously cannot exclude that there are problems with the assemblies of some of the other strains as well.

For the revision we have edited the way we describe our *M. tuberculosis* results, and have added comments about the retracted strains and their effect on the frequency of SNPs that clash with the core genome phylogeny.

1.12) Figure 10—figure supplement 3 and Figure 11—figure supplement 2 are interesting because they show a deviation of the *E. coli* and other species data from the standard model of evolution with gene conversion model. It is nice to see that H. pylori looks like a freely recombining species, while M. tb looks like a non-recombining species according to this pattern (I think these plots should be given the same scale). This is the first part of the manuscript where I think I am seeing a new and interesting way of visualizing the data. However, frustratingly, the difference is not worked through and explained. My view is that these figures should be brought into the main text and the difference in pattern between simulated data with intermediate recombination rates and what is seen in *E. coli* should be explained carefully. The problem with the current main text figures is that they are very dry; it is far from obvious what they actually say about what is happening in *E. coli* and it takes a great deal of effort to fight through and understand them.As stated above, in our 2012 paper we showed that recombination was inhomogeneous within *E. coli*. We used ClonalOrigin to infer recombination events and found more events within clades than between them. The visualization in Figure 10—figure supplement 3 and Figure 11—figure supplement 2 is interesting because it shows that inhomogeneity is a direct feature of the data compared to simulated data and does not need complex inference methods to detect. This I believe should be the centre point of the manuscript.

We are very glad that the reviewer appreciates that the results in Figure 10—figure supplement 3 and Figure 11—figure supplement 2 directly show the heterogeneity in recombination without the need of postulating specific models. In the first revision we already moved some of these entropy profile results to the main text (Figure 10) and we do not think it is necessary to move these supplementary figures to the main text as well. We also note that reviewer 2 has the opposite opinion, i.e. that we should spend less space in the main text on these results. We believe that the current presentation strikes a reasonable balance.

At the suggestion of the reviewer, we have reworked the text describing these results to improve the explanation of what they show.

1.13) I asked a talented postdoc to look at the preprint of this paper and tell me what he thought. He spent 4 hours and was not able to decide whether anything could be concluded from all these new methods. The real task of the authors now is to focus down on making what is genuinely new accessible and with the time I can feasibly devote to it these are my comments to help the author to get there. I hope they are grateful!

We’re convinced that many of the comments of the reviewer have ultimately positively impacted the paper.

Reviewer 22.1) The manuscript has largely improved since last version and is making a clear point. It now combines multiple previous observations and some new statistics to discuss the impact of recombination on bacterial genetic diversity and the possibility to represent bacterial evolutionary history as a phylogeny.The addition of situations added clearly some idea for comparison with a standard population genetic background.Yet the picture is still not clear and can become quite technical. Here are a few propositions to improve the manuscript.

In light of this reviewer’s strong skepticism of the original version of our manuscript, we are very grateful to the reviewer for these gracious comments.

2.2) The main problem is that the n-SNPs statistics remain difficult to interpret. Specifically, the authors give an intuition for the n-SNP distribution in a freely recombining population. It should be peaked around a typical number of occurrence per type (Appendix 4—figure 3, brown line), In a clonal population, the n-SNP distribution can also resemble a power law (Appendix 4—figure 3), a fact that is somewhat buried in the Supplement of the manuscript. A first clear description of the expected n-SNPs distribution in these two extreme scenarios is warranted before going to more complex statistics (e.g. the entropy profiles).

We note that reviewer 3 also asked about the interpretation of the n-SNP distributions in the first round of review, and did feel that our revisions had cleared this up. Nonetheless, for this revision we have significantly edited the text to make as clear as possible what these *n*-SNPs represent, including a formal description of how we imagine relative rates of recombination determine the relative frequencies of different *n*-SNP patterns. We have also edited our descriptions of what is observed in the clonal and freely recombining limits, and how those observations fundamentally differ from what we see in the *E. coli* data.

2.3) At the 2 SNPs, it makes sense that having multiple strains with almost equivalent level of 2 SNPs links is a reflection of recombination. But it would be interesting to plot the genomic localisation of the 2SNPs of the Figure 9A. The initial part of the paper mentions that for closely related strains a single recombination event will bring most variable SNPs. As such a recombination between two distant strains may lead to a high 2SNPs score just based on one transfer. Conversely, consider that strain A and B are more closely related and C is their closest relative. All mutations that occurred between on the branch leading to clone ABC will appear as 2 SNPs between either A and B, B and C or A and C if they occurred in regions that have later been overwritten by a recombination in branch C, A or B respectively. Here, the topology will play a major role. The 2SNPs connection will reflect clonality perturbed by recombination, not preferential recombination between strains. The 2 SNPs statistics is both dependent on the topology (its asymmetry, long branches…) and on the recombination. The simulations using coalescent approach did not capture the highly biased sampling structure of the population of genomes studied. So it would be interesting to illustrate on the example of Figure 9A where are the 2SNPs to substantiate the underlying structure of these 2 SNPs.

For this revision we have followed the reviewer’s suggestion and included a supplementary figure showing the distribution along the genome of the 5 most common 2-SNPs from Figure 9A (Appendix 5—figure 2) within the section analyzing phylogroup B2 in detail (A1 belongs to phylogroup B2). Consistent with our observation (from the pair statistics) that little clonal signal is left for this set of strains, and that recombination has overwritten each locus already multiple times, we observe a fairly uniform distribution of these 2-SNPs along the genome. Apart from just visualizing where these SNPs fall along the alignment, we have also shown the distributions of distances between consecutive occurrences of the same 2-SNP, and the distribution of the length of runs of the same 2-SNP. This shows that, although there are some locations where a cluster of the same 2-SNP occurs, runs of identical 2-SNPs are mostly short. Finally, among the top 5 pair-SNP partners of A1 are strains D4 and H6, neither of which belong to phylogroup B2. Appendix 5—figure 2 also shows that 2-SNPs of A1 with D4 and H6 are dispersed all across the core genome alignment and in the latest revision we point out that this also underscores that phylogroups should not be thought of as reflecting clonal ancestry.

2.4) The trouble in the later part of the paper is that the idea of preferential recombination is not clearly supported by the data. The n-SNPs distribution in data do not perfectly follow the simulation model (although on Appendix 4—figure 3, the profiles are not that distinct either). But this does not imply that strains have to present “unique recombination profiles”. Moreover, the fact that the distribution is scale free does not necessarily imply that there are no groups of preferential recombination. All in all, we had trouble following the whole paragraph on n-SNPs distribution and even more that on the entropy profiles. All we see is that indeed the simulations do not perfectly match these profiles. There could be multiple processes behind this poor match, like population structure for example or sampling biases. We would therefore suggest to considerably shorten that part (which comes at the end of an already long paper) and to present the “unique recombination profile” as one hypothesis by which these patterns could possibly arise (although this is only based on a verbal argument).

Reviewer 2’s opinion differs from the opinions of both reviewers 1 and 3 on these points. Reviewer 3 agrees we have explained the interpretation of these n-SNP distributions and feels they clearly differ from the simulations and show evidence of population structure. Note also that, whereas reviewer 2 appears to contrast ‘population structure’ and ‘preferential recombination’, these two terms mean essentially the same in this context (as reviewer 3 has stressed). Regarding how long this section should be, we note that here the reviewer is also in disagreement with those of the other two reviewers, e.g. reviewer 1 feels we should expand the section on the entropy profiles. Overall we believe that the amount of space invested on the *n*-SNP statistics is appropriate.

For this revision we have rewritten the section in question once more to stress that our results show that the *n*-SNP distributions and entropy profiles show that different lineages recombine at significantly different rates, and that (apart from strains that had a common ancestor very recently) the lineages of each strain have a unique pattern of recombination with the lineages of other strains.

2.5) Heterogeneity in selection: One question that is not addressed is the heterogeneity in selective pressure along the genome? This heterogeneity could contribute to an over estimation of the fraction that is linked to recombination. Specifically, in the figure showing the accumulation or recombination with divergence, the last panel could be misleading because to the lack of resolution. Some genes are much more conserved than others and that intergenic regions. Synonymous are much more frequent (more than 10 fold) than non-synonymous…. All these factors could play a role in the variability observed at intermediate levels of divergence. Could a zoom on some regions reveal the heretogeneity of divergence and its decorrelation from the underlying genetic structure. Is this pattern different from simulations that lack these diversity on the level of constraints.

The reviewer is presumably referring to the panels of Figure 2A-C and wondering whether we may be over-estimating the amount of recombination because of heterogeneity in purifying selection. If we understand it correctly, the reviewer imagines that one could mistake a region with little purifying selection, and thus, increased mutations, for a recombined region. Or vice versa, mistake regions under strong purifying selection for ancestral regions. We agree with the reviewer that, for pairs that are close to fully recombined, it becomes harder to clearly distinguish ancestral from recombined regions, but this does not affect the overall statistics that we derive from the pair analysis, which are driven by closer pairs that contain substantial fractions of ancestrally inherited DNA, i.e. as in Figure 2A and 2B, where the recombined regions can be easily detected.

Moreover, we stress again that our quantification of recombination is not only based on the pair analysis, but also on the consistency of SNP splits, and those statistics are not affected by selection.

2.6) For the detection of homoplasy, the impact of synonymous and transitions transversion, the model proposed could be better explained and the supporting data (number of transitions transversions leading to bi, tri or quadri allelic data for different types of mutation (synonymous, intergenic...)) should be provided.

For the revision we have edited this section to improve clarity and expanded the explanation. In addition, we have added a table reporting the statistics the reviewer requested (Table 2).

2.7) Overall the article has improved but we fill the nSNP interpretation, could still be clarified and/or shortened.

As mentioned above, we have now further revised the section on the interpretation of the *n*-SNP statistics to improve clarity and expand the explanation.

2.8) Please specify the length of recombination fragments (20 kb) when you explain how recombination is implemented in the simulations.

In the simulations the lengths of the recombination segments were 12Kb, which we now also specify in the main text.

2.9) In paragraph “Recombination rates across lineages follow approx... scale-free distributions”, did not follow the sentence starting in “Formally,.…” are nearest neighbour in the phylogeny” - which phylogeny are you mentioning?

It is the different phylogenies along the genome alignment, i.e. each phylogeny at each locus. We have edited the text to make this clear.

2.10) Discussion “n-SNP types reflect the relative frequencies with which different subsets of n strains share a common ancestor across the phylogenies” - is that statement correct?

We have rewritten this section to make sure the statement is clear.

Reviewer 3 - Daniel B Weissman:3.1) I was very positive in my first review, and I remain so: all the things that I really liked about the manuscript are still there, with more support. The authors have addressed my main comment (reorganizing the manuscript) and one of my main comments (interpreting the n-SNP distributions), and partially addressed my other main comment (quantitative analysis of agreement between SNPs and the core tree), while explaining why they have not done more (it’s just a hard problem that will take some more serious analysis than my dashed-off idea). My main concern is just that the first submission was already quite a lot to take in (e.g., despite trying to read it carefully, I missed the site frequency spectrum) and now after addressing all the comments it’s really sprawling. I can’t think of magic fix to this, but a few of the minor suggestions below may help.

We are happy that reviewer 3 feels we have addressed his comments about the interpretation of the *n*-SNP distributions. We agree with the reviewer that, in response to the many additional controls that reviewers 1 and 2 requested, including simulations, there is now so much extra material that it will make it harder for readers to follow the main thread of the argument. In this second revision we have gone over the whole text again and tried to improve the presentation so that the overall thread is clear.

3.2) There’s a lot to the simulation analysis, and I got kind of lost at points. To me, there are two main points of the simulations. The first is just to provide a quick check on the all the statements the authors are making in the first part of the manuscript, where they show that recombination has erased the clonal tree. This part was pretty easy to follow.

We agree with the reviewer.

3.3) The second main point are the n-SNP distributions. The authors’ main claim about the *E. coli* n-SNP distributions is that they are roughly scale-free. The human ones exhibit this even more clearly. The simulations show a different pattern. This suggests that a more complex history, with a mix of population size changes, population structure, and gene flow, as in humans, is a better starting point than the simple simulation model.

Indeed, in the first round of reviews the reviewers questioned whether a simple model of neutral evolution plus fully mixed recombination would not lead to the same n-SNP distributions and the second point of the simulations is to show that it doesn’t. In this revision we also tried to make clearer that this is the main second point to take from these simulations.

3.4) I didn’t care so much about the fact that the simulations don’t match all the *E. coli* diversity statistics in detail – it would be weird if they did, given that they don’t even have the same sample size and that the core tree doesn’t look like a Kingman tree. If I’m correct about the two main points, it would be helpful if the authors could emphasize this to help guide the reader.

We agree and have tried to make this clearer in the revision.

3.5) Reviewer 1 mentioned the issue of biases in the estimation of the length distribution of the recombined segments. I don’t think the simulations fully address this, particularly the problem of missing small recombination events, since they assume that all recombination events are 12 kb. The authors mention that most of their conclusions don’t depend on the distribution of recombined segment lengths, which I agree with, but I think that the estimate of how many times each part of the genome is overwritten by recombination does depend on it. One could imagine a situation in which most of the phylogeny switches from alignment column are caused by very short recombined segments, while the segments detectable in close pairs are much longer. I don’t think that’s the case here, but the authors should make an argument why it’s not. What does ρ/µ look like if you just estimate from the close pairs? Is it roughly consistent with the ρ ≈ 0.3 − 1 that gives frequency of phylogeny changes in the alignment in simulations?

We agree with the reviewer that the estimate of the average size of recombination events is unimportant for almost all of our conclusions, and only affects the estimate of the number of times *T* each locus has been overwritten by recombination. Regarding the question of whether it is plausible that our lower bound estimate for *T* is still an overestimate, we refer the reviewer to our response 1.10. In short, for our estimate of the average length *L_r_*of the recombined segments to be an over-estimate by *n* fold one would need to have that for each recombined segment of length 250 or more, there are at least *n* recombined segments of length 100 or less. We feel that it is therefore not plausible that are estimate is off by a large factor.

The reviewer further asks whether we cannot get an estimate of *ρ/µ* using the analysis of close pairs. In response we have now added a Appendix 3 that estimates *L_r_ρ/µ* using statistics from partially recombined pairs for which the fraction of clonally inherited genome is between 0.25 and 0.75. This analysis suggests that *L_r_ρ/µ* ranges from 2100 to 4600 for these pairs.

Assuming that these rates also apply to the full evolutionary history of the strains in the genome alignment, the Appendix 3 also explains that, because a fraction 0.1 of columns in the alignment are polymorhic, this implies that the average number of times *T* each position in the alignment has been overwritten by recombination lies between *T* = 210 and *T* = 460. Note that this is consistent with our estimate based on the lower bound on the number of phylogeny changes *C*, which suggests *T* is at least 191 and may be as large as 1000.

However, we feel that this simple estimate based on the pair statistics should only be considered an order-of-magnitude check on the lower bound obtained from *C* for several reasons. First, the simple model assumes that, for a given pair, recombination has occured at a constant rate *ρ* since they diverged from a common ancestor. However, our results show that recombination rates vary strongly across lineages. More importantly, to estimate *T* we have to assume that the estimates of *L_r_ρ/µ* for the partially recombined pairs can be extended to the full core genome alignment, and it is not clear at all whether this is valid. We have noted this as well in Appendix 3.

3.6) I would replace “recombination rates between lineages” with “population structure” as much as possible. First, technically, with so much recombination, each sample corresponds to many lineages, not one, and the recombination will almost never actually occur between two lineages ancestral to sampled individuals. Rather, recombination will split one ancestral lineage, and then deeper in the past one of the resulting lineages will coalesce with a different ancestral lineage. Second and more importantly, population structure is a familiar notion in population genetics, so it will hopefully reduce cognitive load on readers. You can emphasize early on that this structure doesn’t need to be geographic. For “variation of relative recombination rates among lineages”, maybe “variation in proportions of shared ancestry among lineages”? Not sure, could also see leaving it as-is.

Precisely what terminology to use is something we have been thinking about quite a bit and it is very useful to have the reviewer give his views on this. We agree that ‘population structure’ is a familiar notion to some, but this may not include all readers of our paper and some may mistakenly think we are talking only about geographic distribution or something like that. Note that, in his last review, reviewer 2 refers to population structure and ‘preferential recombination’ as if they were two separate things. In our latest revision we have attempted to make sure that it is clear how we are using the terminology and we have in general increased the use of the term population structure.

The reviewer’s comments also made us realize that with ‘lineages’ we were not only thinking of the clonal ancestors of the sample, but the lineages of all ancestors that have contributed genetically to the genomes in the sample. In this revision we have also tried to make this point clearer by pointing how we use the term ‘lineages’.

3.7) In the human section: “few if any loci in the genome that follow this strictly maternal lineage”. I would explicitly mention the mitochondrion, and actually build a mitochondrial tree for the human sample that you can compare to the whole-genome tree. This would really help make things concrete, which I think would help people understand the bacterial results. If you want to, you could even include a cartoon human population tree, showing African and European admixture into Native Americans. Seeing the three trees (genomic consensus, mitochondrial/clonal, and model population structure) next to each other would drive the point home.

We agree that contrasting a mitochondrial tree with the whole-genome tree would help drive the point home. Unfortunately, the mitochondrial sequences are not available for these genome sequences.

3.8) I didn’t really understand the motivation behind the simulation setup. Why was it important that there be a single clonal tree shared by all the simulations? If anything, this seems to introduce a lot of noise, and I would be worried about unluckily drawing a weird tree. I would think it would be easier to use a pure coalescent simulation. If there’s an issue with the ARG getting too huge because recombination is too frequent, then I think that means that there is a corresponding issue in the forward-time simulations where multiple recombined segments are too frequently being drawn from the same lineages. The authors might even be able to use existing software. ms and I believe msprime can do gene conversion, and if the exploding ARG is a problem, FastSimBac uses a Sequential Markov approximation and also keeps track of the clonal frame. I don’t know if any of them record all the stats that the authors use in this manuscript though.

The reason we use the same clonal tree is to be able to more directly show the effect of different recombination rates on various statistics. For example, in Figure 1—figure supplement 3 we show, for different recombination rates, what the fraction is of the genome that has been overwritten by recombination along each branch of the clonal tree. If there had been different clonal trees for each simulation, it would be much harder to meaningfully compare these statistics for different recombination rates.

We have extensively looked into using existing software for doing our simulations but none allow us to gather all statistics we want to gather.

3.9) I would try to maintain a clear distinction between a locus being overwritten many times by recombination in the phylogeny as a whole, and being overwritten many times along the line of descent of any given individual. As I understand it, the authors are making the second, stronger claim for most of the species studied, but I wasn’t sure what the actual numbers quoted in, e.g., Figures 6—figure supplement 3 and 4 correspond to.

In Figures 6—figure supplement 3 and 4, we are referring to the number of times positions have been overwritten along the whole phylogeny. In this revision we have made sure to be explicit about this so that no confusion can occur.

3.10) “Indeed, if we remove…”: I’m not sure that the argument in this paragraph is quite correct as written. Suppose we had a large sample from a basic unstructured Wright-Fisher population with a low rate of recombination. Then even if we were to take only recombined loci, wouldn’t we still get back something like the clonal phylogeny? I think the key point is the one you make in the sentence before this paragraph, that every locus has recombined many times. So then yes, maybe those all those trees look like the clonal phylogeny, but there’s no particular reason to expect them to, any more than we expect all the loci in humans to have trees that look like the mitochondrion’s.

We don’t think that there is anything incorrect with the argument in this paragraph, but we clearly did not explain ourselves well enough. This section of the paper concerns the question of why there is a robust core phylogeny in the face of so much recombination. At the start of this section we introduced a viewpoint that many in the field take regarding this question: “some researchers interpret this convergence to the core tree to mean that the core tree must correspond to the clonal phylogeny of the strains. The interpretation is that the SNP patterns in the data are a combination ‘clonal SNPs’ that fall on the clonal phylogeny, plus a substantial number of ‘recombined SNPs’ that were affected by recombination, which act so as to introduce noise on the clonal phylogenetic signal. In this interpretation, trees build from individual loci can differ from the core tree because the ‘recombination noise’ can locally drown out the true clonal signal, but once sufficiently many loci are considered, the recombination noise ‘averages out’ and the true clonal structure emerges.”

If this interpretation were correct, one would expect that explicit removal of all SNPs that fall on the core tree would destabilize the phylogeny. However, this is not what we observe. For this revision we have rephrased this section with the aim of making this point clearer.

3.11) “Essentially all population genetics…”: I’m not sure that I see the contradiction in this paragraph. As I understand, say, STRUCTURE or ADMIXTURE, they have no problem with each sampled individual having a unique distribution of ancestry across populations, which would seem to correspond to the claim being made about the *E. coli* strains.

Software such as STRUCTURE fit the data to a general purpose mixture model which, at least as far as we understand it, is not based on any explicit model for the evolutionary dynamics. We suspect that it would not be easy to device an explicit dynamical model to reproduce a specific fit to the mixture model of STRUCTURE.

3.12) The SI would be easier to navigate if it had a table of contents and if Table S5 were a separate data file.

We agree with the reviewer and, for the revision, we have made these changes.

[Editors' note: we include below the third round of reviews that the authors received from another journal, there is no corresponding authors’ response, as the authors opted to submit to *eLife*.]

Thank you for the revisions and for the lengthy rebuttal.The problem with this paper continues to be that while the analyses are interesting, the description of the results and their interpretation remains garbled and confusing. Previous literature is mischaracterized but beyond that the manuscript fails to be clear in its own terms. These two problems need to be solved in parallel.It is quite tiring to spend a long time wading through unclear text and then to get an overly combative response yet again. For this reason, I have again not attempted a comprehensive review of the manuscript but tried to focus on the most important problem, which is that the manuscript needs a clear statement of what is actually new - using instructive examples.If correctly explained, the authors do have more than enough material for a solid and exciting contribution to bacterial population genetics that would be suitable for publication. Alas, unless the core issues of scientific clarity are addressed, the manuscript remains unsuitable for publication anywhere.The authors are correct to highlight the issue, we do not know how many well-resolved branches can be attributed to clonal structure and how many can be attributed to non-random patterns of recombination (and how many can be attributed e.g. to technical factors in phylogenetic reconstruction) and that we indeed have limited understanding of the effect of gene flow on overall patterns of variation. It is an interesting, deep and under-appreciated question in bacterial evolution. The best parts of the paper (still alas buried in the supplement) show that the null single gene-pool model, with homogeneous recombination rates between lineages (which is indeed used as the basis for many inference methods and simulations, especially due to the difficulty of implementing other models) is very far from being correct in many species.The paper examines some summary statistics that avoid several of the assumptions made by more parameterized inference approaches. But they do not provide meaningful quantitative estimates of, for example, how many well-resolved branches of the phylogeny in any given species are inconsistent with the clonal tree. Furthermore, they do not provide estimates of how non-random recombination actually is. These problems are not easy to address directly.The title, which describes “long-tailed distribution of recombination rates” is therefore extremely misleading. The authors do not provide estimates of recombination rates between pairs of lineages. The long-tailed distribution refer to n-SNP patterns, not directly to recombination patterns. This issue of jumping between one and the other is repeated throughout the manuscript and destroys its scientific clarity.“Each genomic locus has been overwritten so many times by recombination that it is impossible to reconstruct the clonal phylogeny and, instead of a consensus phylogeny, the phylogeny typically changes many thousands of times along the core genome alignment.”This section of the Abstract is entirely garbled. The question of whether it is possible to recover the clonal genealogy is a technical question rather than a biological one. We always know there is a clonal genealogy, because bacteria always reproduce by binary fission, the question is just whether it is recoverable from the data. For example, if the level of recombination between strains was determined entirely by how closely related they were (which is close to the truth if recombination is extremely homology dependent), then you could have infinite recombination rates and still be able to successfully reconstruct clonal relationships. I can only guess what is meant by the sentence “instead of a consensus phylogeny”. A consensus phylogeny is something that it is always possible to calculate but is often difficult to interpret. So in what sense is it “instead” that the phylogeny (which means actually local genealogies) changes many thousands of times. I cannot see what possible logic this sentence could be conveying.“Our findings show that bacterial populations are neither clonal nor freely recombining, but structured such that recombination rates between different lineages vary along a continuum spanning several orders of magnitude, with a unique pattern of rates for each lineage.”The first part is most definitely not new while the second part might be true but is not demonstrated. Recombination rates have not been measured directly, only n-SNP patterns.“The reviewer no longer raises any technical issues.”The reason I did not raise many technical issues in my review of Revision 1 is not because I had gone carefully through the entire manuscript and was unable to find any but because, as I stated in the review, I was already exhausted and spent far more time on the review than I wanted by the time I got to the end of the manuscript introduction. Now we are at revision 2 and there was a real danger of exhaustion by the time I get to the end of reading and responding to the cover letter. Do they actually want me to state every technical issue I have?“Our conclusions were in fact fully warranted.”I have pointed out several substantive and important mistakes, such as the factor of 2 misestimate of recombination in the previous version. The authors did actually make many scientifically substantive changes in response to my comments in this iteration as well on M. tb and on variation in tract length, which I thank them for.The main point here is that the conclusions were not, and still are not, clearly stated. So how can they possibly be fully warranted?I’m not going to respond line-by-line to the rebuttal of my reviewer points made in the cover letter, although it’s tempting, but the whole premise of the argument that is made is false in a key respect:“It is clear from the reviewer’s comments, however, that he continues to assume that robust phylogenetic structures in whole genome alignments can only reflect ancestry.”In 2003, my colleagues and I published a paper entitled “Traces of human migrations in Helicobacter pylori populations” in Science. Helicobacter pylori is a particularly recombinogenic species (as their analysis confirms) and recombination has abolished most obvious traces of clonal structure in the genomic data, at least if strains are sampled from unrelated individuals within large human populations. Nevertheless, in population level H. pylori data one recovers a tree, which has several well-supported deep branches. Indeed, figure 1 in that paper is a population tree, showing those branches. These robust phylogenetic structures clearly reflect patterns of gene flow (populations refers to population genetics which is the study of gene pools) and not clonal structure. (as I noted, their usage of the term ancestry is non-standard – it also seems to be extremely vague - and should be avoided).If strains are sampled from related individuals, one often finds clones and can infer some clonal structure, as I did for example in a paper published in 2015 in the journal Gut which found that the strains sampled from a single patient derived from two distinct clones but had recombined substantially.Moreover, in 2019 my colleagues and I published a paper in ISME journal showing that the marine pathogen Vibrio parahaemolyticus also has populations defined by gene flow, while showing near-to free recombination within each of those populations. Note however that in this species there is plenty of evidence for clonal structure. For example, if you sample strains that cause human disease, you find a handful of lineages of recent origin predominate. Within these lineages, there are nice time-resolved phylogenies that make total sense. It is easy to infer clonal structure and it is also known that recombination rates can differ substantially within these lineages.“The central novel insight of our work is that the robust phylogenetic structure that is evi- dent in whole genome alignments of bacterial strains does not reflect clonal ancestry, but reflects population structure.”This “central novel insight” is thus neither new nor true. In Helicobacter pylori we have already long ago and more recently in Vibrio parahaemolyticus shown that robust phylogentic structure in some parts of the tree results from clonal ancestry and in other parts of the tree from population structure. It all depends on where in the tree you look.“We are much more optimistic than Max Planck and expected that we can convince the ‘old turks’ that our work contains results that are both new and true. We’ll leave it up to the reviewer to decide whether this optimism is warranted.”I believe that there are new and true results in this work but unfortunately my doubts persist about whether they will ever be described accurately enough to allow people in the field to establish what they are.